# Hom-PGD+: Fast Reparameterized Optimization over Non-convex Ball-Homeomorphic Set

## Abstract

Optimization over general non-convex constraint sets poses significant computational challenges due to their inherent complexity. In this paper, we focus on optimization problems over non-convex constraint sets that are homeomorphic to a ball, which encompasses important problem classes such as star-shaped sets that frequently arise in machine learning and engineering applications. We propose **Hom-PGD+**, a fast, *learning-based* and *projection-efficient* first-order method that efficiently solves such optimization problems without requiring expensive projection or optimization oracles. Our approach leverages an invertible neural network (INN) to learn the homeomorphism between the non-convex constraint set and a unit ball, transforming the original problem into an equivalent ball-constrained optimization problem. This transformation enables fast projection-efficient optimization while preserving the fundamental structure of the original problem. We establish that Hom-PGD+ achieves an $\mathcal{O}(\epsilon^{-2})$ convergence rate to obtain an $\epsilon + \mathcal{O}(\sqrt{\epsilon_{\mathrm{inn}}})$-approximate stationary solution, where $\epsilon_{\mathrm{inn}}$ denotes the homeomorphism learning error. This convergence rate represents a significant improvement over existing methods for optimization over non-convex sets. Moreover, Hom-PGD+ maintains a per-iteration computational complexity of $\mathcal{O}(W)$, where $W$ is the number of INN parameters. Extensive numerical experiments, including chance-constrained optimization popular in power systems, demonstrate that Hom-PGD+ achieves convergence rates comparable to state-of-the-art methods while delivering speedups of up to one order of magnitude.

## 1 Introduction

We consider a class of non-convex constrained optimization problems where the constraint set is homeomorphic to a unit ball, also known as *ball-homeomorphic* (BH) sets. BH sets encompass any *compact convex set* and a class of *simply-connected non-convex* sets, such as star-shaped and geodesic-convex sets. This problem is fairly general and covers numerous optimization classes, including standard convex programming (Boyd et al., 2004), chance-constrained programming (Nemirovski & Shapiro, 2006; Pagnoncelli et al., 2009), and $\ell_p$-constrained regression (Xu et al., 2010; Jiang et al., 2016). These optimization problems naturally arise in real-world applications in machine learning and engineering, such as chance-constrained power grid optimization (Pagnoncelli et al., 2009) and $\ell_p$-constrained adversarial attacks in neural networks (Erdemir et al., 2021). While convex constrained optimization has been extensively studied and can be solved efficiently, this paper focuses on optimization over non-convex constraint sets, which present significant additional challenges.

Optimization over non-convex sets is highly challenging. Even establishing the feasibility of a general non-convex set can be *NP-hard* (Park & Boyd, 2017). Furthermore, in many real-time operational scenarios, one must repeatedly solve the same class of problems with varying parameters, introducing uncertainty and variability in a setting known as *parametric optimization* (Grancharova & Johansen, 2012). This scenario poses significant computational challenges. Traditional approaches include convex relaxation (Low, 2014a;b; Diamond et al., 2018; Anstreicher, 2012), reformulation-linearization (Sherali & Adams, 2013), and sequential convex approximation (Marks & Wright, 1978; Beck et al., 2010; Tran et al., 2013; Scutari et al., 2014). However, these methods are computationally expensive and do not provide tight guarantees on feasibility or optimality. Recent state-of-the-art works (Lin et al., 2022; Kume & Yamada, 2024; Ma et al., 2019) have proposed more efficient methods under different structural conditions and established convergence guarantees. Nevertheless,

Table 1: Summary of parameterization or iterative methods for (non)-convex constrained optimization.

| Reference | Settings Obj. | Settings Ctr. | Key Assumption | Parameterization Techniques | Algorithm | Per-iteration Complexity | Convergence Rate |
|---|---|---|---|---|---|---|---|
| (Li et al., 2023) | NC | Simplex | – | *Hadamard Transformation* | Perturbed RGD | $\mathcal{O}(n)$ | $\mathcal{O}(\epsilon^{-2})$ |
| (Chok & Vasil, 2025) | C | Simplex | – | | Cauchy-Simplex | $\mathcal{O}(n)$ | $\mathcal{O}(\epsilon^{-1})$ |
| (Tang & Toh, 2024) | (N)C | Polyhedra | Full-rank constraints. | | RGD + PGD | RO + PO | N/A |
| Liu et al. (2025a) | C SC NC | Convex | Non-degeneracy. – – | *Gauge Mapping* | PGD over ball | $\mathcal{O}(n^2)$ + MO | $\mathcal{O}(\epsilon^{-1})$ $\mathcal{O}(\log \epsilon^{-1})$ $\mathcal{O}(\epsilon^{-2})$ |
| (Barber & Ha, 2018) | SC | NC | Small local concavity coefficients of constraints. | – | PGD | PO | $\mathcal{O}(\log \epsilon^{-1})$ |
| (Lin et al., 2022) | WC | WC | Certain non-singularity. Initial feasible points. | – | Proximal-point penalty method | SCOO | $\tilde{\mathcal{O}}(\epsilon^{-3})$ $\tilde{\mathcal{O}}(\epsilon^{-4})$ |
| (Barik et al., 2023) | IV SIV | IV | Contraction and triangle inequality w.r.t. invexity. | – | Invex PGD | Invex PO | $\mathcal{O}(\epsilon^{-1})$ $\mathcal{O}(\log \epsilon^{-1})$ |
| Theorem 1 | NC | NC | Ball-homeomorphic. | *Invertible Neural Network* | Bisected-PGD | $\mathcal{O}(W)$+MO | $\mathcal{O}(\epsilon^{-2})$ |

[1] **Abbreviations**: C = "convex", NC = "non-convex", WC = "weakly convex ", SC = "strongly convex", IV = " invex ", SIV = " strongly invex " Obj = "objective", Ctr = "constraint", GD = "gradient descent", PGD = "projected gradient descent", RGD = " Riemannian gradient descent", SCOO = " strongly convex optimization oracle", MO = "membership oracle", PO = "Projection oracle", RO = "Retraction oracle".

[2] **Convergence rate**: number of iterations for finding an $\epsilon$-approximate stationary point for non-convex optimizations or an $\epsilon$-approximate optimum for convex optimizations.

[3] **Complexity**: Here $W$ denotes the size of the neural network we use to learn a homeomorphic mapping, referring to Sec. 3. In practice, we choose $W = \mathcal{O}(n^2)$ where $n$ is the problem size. Notably, Membership oracle (MO) enjoys the lowest complexity compared with other optimization-based oracles in general settings (Mhammedi, 2022).

several issues remain, including slower convergence rates, expensive per-iteration oracles, and the necessity for strong convergence assumptions.

In recent years, *reparameterization* has emerged as a powerful technique for solving challenging optimization problems by transforming them into simpler, more tractable forms. The core idea involves applying invertible/smooth transformations that preserve optimal solutions while mitigating difficulties such as non-smoothness or complex constraints. This approach has been successfully applied in semidefinite programming (Cifuentes, 2021), low-rank optimization (Mishra et al., 2014; Ha et al., 2020), and risk minimization (Bah et al., 2022). Recent works have extended this concept to optimization over simplices (Li et al., 2023), polyhedra (Tang & Toh, 2024), and general compact convex sets Liu et al. (2025a), as well as smoothing non-smooth objectives (Poon & Peyré, 2023) and modeling discrete data (Davis et al., 2024). However, most applications remain confined to convex settings (see Table 1) and require well-designed transformations. For more complex non-convex constraints, recent works (Liang et al., 2023; 2024) propose to use invertible neural networks (INNs) (Papamakarios et al., 2021; Dinh et al., 2014) for reparameterization. However, they focus on projection in the transformation space for the infeasible neural network predictions, rather than solving the optimization problems from initial points. We refer readers to Appendix A for a more detailed discussion on reparameterization and non-convex constrained optimization.

Despite the progress made for (non)-convex constrained optimization, a research gap still remains: *"Can we design an efficient approach for optimization over non-convex ball-homeomorphic sets with fast convergence and low per-iteration cost?"*

In this work, we propose a fast *first-order*, *learning-driven* and *projection-efficient* method for solving *parametric* optimization over *non-convex BH* sets. One could refer to Table 1 for a summary and comparison of existing work and our method. Specifically, we make the following contributions:

▷ In Sec. 3, we propose **Hom-PGD$^+$**: (i) it first exploits the BH structure of the constraints by employing an INN to parameterize the homeomorphism; (ii) it then reformulates the optimization over BH sets as an equivalent ball-constrained optimization via the learned INN; and (iii) it applies projection gradient descent to solve the ball-constrained problem and transforms the converged solution back to obtain the solution for the original problem.

▷ In Sec. 4, we establish convergence and complexity analysis for **Hom-PGD$^+$**: (i) it finds an $\epsilon + \mathcal{O}(\sqrt{\epsilon_{\text{inn}}})$-stationary point in $\mathcal{O}(\epsilon^{-2})$ iterations, where $\epsilon_{\text{inn}}$ is the INN learning error. This convergence rate outperforms existing first-order methods for optimization over non-convex sets (see Table 1). (ii) it achieves a per-iteration complexity of $\mathcal{O}(W)$, where $W$ is the number of INN parameters and setting $W = \mathcal{O}(n^2)$ is sufficient to achieve strong performance in practice. It demonstrates the scalability of our method compared to other methods requiring expensive optimization oracles.

▷ In Sec. 5, through extensive numerical experiments on non-convex problems, including applications to non-convex *quadratic-constrained* and *chance-constrained* optimization with applications in power grid operation, we demonstrate that **Hom-PGD**$^+$ outperforms existing approaches in computational efficiency, achieving both faster convergence and lower per-iteration cost.

## 2 PROBLEM STATEMENT

We consider the following *parametric* constrained optimization problem:

$$\min_{\mathbf{x}} \ f_{\boldsymbol{\theta}}(\mathbf{x}), \quad \text{s.t. } \mathbf{x} \in \mathcal{K}_{\boldsymbol{\theta}}, \tag{P}$$

where $\mathbf{x} \in \mathbb{R}^n$ is the decision variable and $\boldsymbol{\theta} \in \Theta \subseteq \mathbb{R}^d$ is the input parameters. The objective function $f_{\boldsymbol{\theta}}(\cdot)$ is continuous and smooth, and the constraint set $\mathcal{K}_{\boldsymbol{\theta}} \subset \mathbb{R}^n$ is compact. For ease of analysis and without loss of generality, we assume the constraint set $\mathcal{K}_{\boldsymbol{\theta}}$ is defined by inequalities[1] as $\mathcal{K}_{\boldsymbol{\theta}} = \{\mathbf{x} \in \mathbb{R}^n \mid \mathbf{g}_{\boldsymbol{\theta}}(\mathbf{x}) \leq \mathbf{0}\}$ with $\mathbf{g}_{\boldsymbol{\theta}} = (g_{1,\boldsymbol{\theta}}, \cdots, g_{m,\boldsymbol{\theta}})$, where $g_{i,\boldsymbol{\theta}} : \mathbb{R}^n \to \mathbb{R}$ are continuous functions. We further impose the following topological assumption on the constraint set $\mathcal{K}_{\boldsymbol{\theta}}$.

**Assumption 1.** The set $\mathcal{K}_{\boldsymbol{\theta}}$ is homeomorphic to a unit ball $\mathcal{B}^2$, denoted as $\mathcal{K}_{\boldsymbol{\theta}} \cong \mathcal{B}, \forall \boldsymbol{\theta} \in \Theta$.

Homeomorphism (or homeomorphic mapping) is a bi-continuous bijection from two topological spaces, guaranteeing the topological equivalence. The non-convex BH constraint is fairly general, covering *a broad class of compact and simply-connected non-convex sets* [3], and many real-world applications in machine learning and engineering as discussed in Sec.1.

**Open Issues:** While constrained optimization has been extensively studied, approaches for non-convex sets typically suffer from strong assumptions for convergence, slow convergence rates, or high per-iteration computational complexity. The central challenge is to develop efficient algorithms that not only preserve fast convergence but also maintain computational efficiency across both general convex and a broader range of non-convex programs.

## 3 HOMEOMORPHIC OPTIMIZATION APPROACH

Motivated by projection-free and reparameterization frameworks to speed up optimization problems over *convex* sets, (Li et al., 2023; Liu et al., 2025a), we propose to transform the original *non-convex* problem through a homeomorphic mapping between the constraint set $\mathcal{K}_{\boldsymbol{\theta}}$ and a unit ball $\mathcal{B}$, which preserves the problem structure while simplifying the constrained set.

**Definition 3.1** (Homeomorphic Constrained Optimization). Given a homeomorphism $\boldsymbol{\psi}_{\boldsymbol{\theta}} : \mathcal{B} \to \mathcal{K}_{\boldsymbol{\theta}}$, we define the transformed parametric optimization problem with objective function $h_{\boldsymbol{\theta}}(\mathbf{z}) = f_{\boldsymbol{\theta}}(\boldsymbol{\psi}_{\boldsymbol{\theta}}(\mathbf{z}))$ and constraint set as a unit ball $\mathcal{B} = \boldsymbol{\psi}_{\boldsymbol{\theta}}^{-1}(\mathcal{K}_{\boldsymbol{\theta}})$ as:

$$\min_{\mathbf{z}} \ h_{\boldsymbol{\theta}}(\mathbf{z}), \quad \text{s.t. } \mathbf{z} \in \mathcal{B}. \tag{H}$$

Under Assumption 1, we can transform any optimization problem **P** over a BH set into a ball-constrained program **H**. Notably, under the homeomorphic transformation, the original problem and its homeomorphic counterpart are equivalent, i.e., there exists a bijective correspondence between their optimal solution sets $\mathbf{P}^*$ and $\mathbf{H}^*$, where $\mathbf{P}^* = \{\mathbf{x} \mid \mathbf{x} \in \arg\min\{\mathbf{P}\}\}$ and similarly for $\mathbf{H}^*$. Specifically, for any $\mathbf{x} \in \mathbf{P}^*$, there exists a unique $\mathbf{z} \in \mathbf{H}^*$ such that $\mathbf{x} = \boldsymbol{\psi}(\mathbf{z})$, and vice versa. Thus, we can solve the reparameterized problem **H** without expensive projection to obtain the corresponding optimal solution of the original problem **P**.

However, finding homeomorphic transformations for general BH constraints remains non-trivial. Many existing *reparameterization* methods for optimization problems rely on explicitly constructed parameterized transformations. For instance, the *Hadamard transformation* (Li et al., 2023) enables mapping from a simplex to a sphere, while the *Gauge mapping* (Liu et al., 2025a) facilitates transformation from a compact convex set to a unit ball. Although these methods successfully construct specific homeomorphisms, they face several fundamental limitations: (i) Explicit or analytical forms for homeomorphisms do not exist for more general non-convex BH sets. (ii) The computational

---

[1] Equality constraints can be removed without loss of generality, see Appendix B.1 for discussions.

[2] In this work, we refer a unit ball $\mathcal{B}$ to a Euclidean norm ball, i.e., $\mathcal{B} = \{\mathbf{z} \in \mathbb{R}^n : \|\mathbf{z}\|_2 \leq 1\}$.

[3] For example, simply connected compact sets with Jordan curve boundary over $\mathbb{R}^2$ (Garnett & Marshall, 2005) and contractible manifold with simply connected boundary over $\mathbb{R}^n$ for $n \geq 6$ (Smale, 1962).

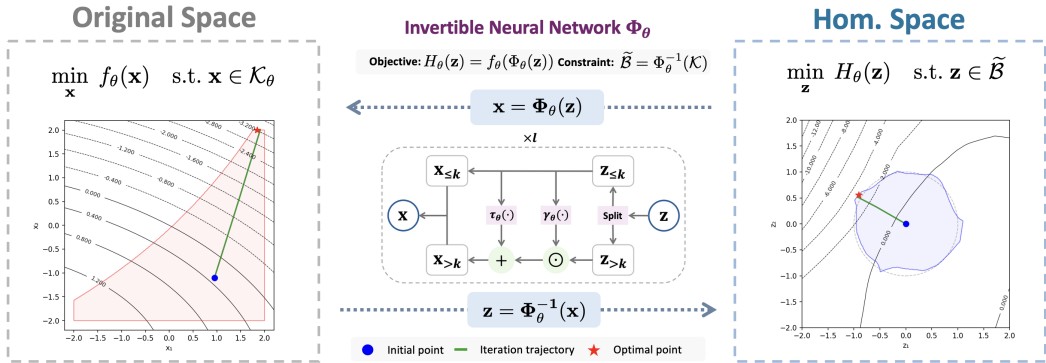

Figure 1: **Hom-PGD$^+$ framework:** It applies projection-based GD methods in a transformed space via an INN-learned homeomorphism $\Phi_{\theta}(\cdot)$, where the transformed constraint set $\tilde{\mathcal{B}}$ is an approximated ball and $h_{\theta}$ is the transformed objective. The iterative trajectory is visualized in the transformed homeomorphic space and also mapped back to the original space for comparison.

overhead required to construct different homeomorphisms for varying constraint sets becomes prohibitive when input parameters change frequently, thereby limiting the practical applicability of these approaches in real-time operational settings.

To address these limitations, we propose **Hom-PGD$^+$**, as illustrated in Figure 1. Our method leverages an invertible neural network (INN), a universal approximator of homeomorphisms, to transform the original non-convex constrained problem under different input parameters into a simple ball-constrained problem (Sec. 3.1 and 3.2). We then apply projected gradient descent (PGD) on the reformulated ball-constrained problem (**H**). Ideally, under an exact homeomorphism, projection is performed onto a unit ball with a closed-form expression. In practice, however, the INN-based homeomorphism provides only an approximation. We then propose a bisection scheme to compute a non-orthogonal projection onto this approximate ball. Complete algorithmic descriptions are provided in Algorithms 1 and 2.

### 3.1 HOMEOMORPHIC PARAMETERIZATION USING INVERTIBLE NEURAL NETWORK

We utilize an invertible neural network (INN)[4] to learn the homeomorphic mapping for general BH sets. An INN is a neural network $\Phi : \mathbb{R}^n \to \mathbb{R}^n$ that is invertible, meaning its inverse $\Phi^{-1}$ is well-defined and computationally tractable. Typically, an INN comprises multiple invertible layers, such as invertible linear layers (Kingma & Dhariwal, 2018), Lipschitz residual layers (Chen et al., 2019; Behrmann et al., 2019), and coupling layers (Papamakarios et al., 2021; Dinh et al., 2014). Furthermore, to parameterize the input-dependent homeomorphic mapping $\psi_{\theta}$, we adopt the conditional INN (Winkler et al., 2019; Lyu et al., 2022). Given changing input parameters $\theta$, we treat them as additional inputs and learn augmented homeomorphisms $\Phi_{\theta} : \mathcal{B} \to \mathcal{K}_{\theta}$, where $\mathcal{K}_{\theta} = \Phi_{\theta}(\mathcal{B})$ denotes the homeomorphic image under specific input parameters $\theta$.

In this work, we select coupling-layer INNs to learn the homeomorphic mapping due to their computational efficiency and universal approximation capability. Specifically, the coupling layer first randomly splits the input into two parts as $\mathbf{x} = [\mathbf{x}_{\leq k}, \mathbf{x}_{>k}]$. Then the forward/inverse mapping is as:

$$\text{Forward} : \mathbf{x}' = [\mathbf{x}_{\leq k}, \; \boldsymbol{\gamma}_{\boldsymbol{\theta}}\left(\mathbf{x}_{\leq k}\right) \odot \mathbf{x}_{>k} + \boldsymbol{\tau}_{\boldsymbol{\theta}}\left(\mathbf{x}_{\leq k}\right)],$$
$$\text{Inverse} : \mathbf{x} = [\mathbf{x}'_{\leq k}, \; \left(\mathbf{x}'_{>k} - \boldsymbol{\tau}_{\boldsymbol{\theta}}\left(\mathbf{x}'_{\leq k}\right)\right) / \boldsymbol{\gamma}_{\boldsymbol{\theta}}\left(\mathbf{x}'_{\leq k}\right)]$$

where $\boldsymbol{\gamma}_{\boldsymbol{\theta}}, \boldsymbol{\tau}_{\boldsymbol{\theta}} : \mathbb{R}^k \to \mathbb{R}^{n-k}$ are regular NNs (e.g., fully-connected), which take input parameter $\boldsymbol{\theta}$ and variables $\mathbf{x}_{\leq k}$ and output weight and bias for element-wise transformation of $\mathbf{x}_{>k}$. Notably, coupling-layer INN can *universally approximate* any target (differentiable) homeomorphism given sufficient layers (Jin et al., 2024; Ishikawa et al., 2022; Lyu et al., 2022), making it theoretically grounded for learning the homeomorphic mapping between constraints and a unit ball in our framework.

---

[4]For a more comprehensive introduction to INNs, we refer the reader to Appendix B.2.

## 3.2 INN Training for Obtaining the Homeomorphism

Next, we introduce the approach for training an INN to approximate the homeomorphism between the BH constraint and the unit ball. Specifically, we employ the following loss function and *maximize* it to train an INN $\Phi_{\boldsymbol{\theta}}$ following (Liang et al., 2024):

$$\mathcal{L}\left(\Phi_{\boldsymbol{\theta}}\right) = \widehat{V}\left(\Phi_{\boldsymbol{\theta}}(\mathcal{B})\right) - \lambda_1 P\left(\Phi_{\boldsymbol{\theta}}(\mathcal{B})\right) - \lambda_2 \widehat{L}\left(\Phi_{\boldsymbol{\theta}}\right) \qquad (1)$$

where $\lambda_1$ and $\lambda_2$ are positive coefficients to balance among the three terms, including:

▷ **Volume term**: $\widehat{V}\left(\Phi_{\boldsymbol{\theta}}(\mathcal{B})\right)$ is a computable approximation of the log-volume term $\log V\left(\Phi_{\boldsymbol{\theta}}(\mathcal{B})\right)$.
▷ **Penalty term**: $P\left(\Phi_{\boldsymbol{\theta}}(\mathcal{B})\right)$ is the penalty term for the constraint violation of $\Phi_{\boldsymbol{\theta}}(\mathcal{B}) \subseteq \mathcal{K}_{\boldsymbol{\theta}}$.
▷ **Lipschitz term**: $\widehat{L}\left(\Phi_{\boldsymbol{\theta}}\right)$ is a computable approximation of the log-Lipschitz term $\log L\left(\Phi_{\boldsymbol{\theta}}\right)$.

For details of computing the three terms and their analysis, we refer readers to Appendix B.4. Intuitively, the first two terms encourage the transformed set to maximize volume while remaining within the BH constraint set; achieving this yields a target homeomorphism. The third term regularizes the Lipschitz constant of the homeomorphism, improving optimization performance in the next stage (with formal convergence analysis in Sec. 4.2).

We then uniformly sample from a unit ball to prepare the training data for the loss function. Further, to train the INN for learning the homeomorphism under different $\boldsymbol{\theta}$, we uniformly sample input parameters $\{\boldsymbol{\theta}_i\}_{i=1}^{N}$ and train the INN following $\frac{1}{N}\sum_{i=1}^{N}\mathcal{L}(\Phi_{\boldsymbol{\theta}_i})$. After finite-sample training, the trained INN only approximates the homeomorphism, i.e., they do not perfectly map the constrained set to the unit ball, or vice versa. However, for our purposes, it suffices that the following validity condition holds to ensure the correctness of the transformed optimization and the projection-based algorithm introduced in the next section.

**Definition 3.2** (Valid INN). The INN approximated mapping $\Phi_{\boldsymbol{\theta}}$ is valid for $\mathcal{K}_{\boldsymbol{\theta}}$ if $\Phi_{\boldsymbol{\theta}}(\mathbf{0}) \in \mathcal{K}_{\boldsymbol{\theta}}$, i.e., it maps the origin in the unit ball to a feasible point in $\mathcal{K}_{\boldsymbol{\theta}}$.

Theoretically, such valid conditions hold for all $\boldsymbol{\theta} \in \Theta$ in the input parameter space, given that it holds for finite covering training data $\{\boldsymbol{\theta}_i\}_{i=1}^{N}$ (Liang et al., 2023; Liang & Chen, 2025). Empirically, we observe that the validity condition is consistently satisfied across both training and test inputs in the experimental section, which is not surprising since we try to keep the entire set within the constraint $\Phi_{\boldsymbol{\theta}}(\mathcal{B}) \subseteq \mathcal{K}_{\boldsymbol{\theta}}$ in loss design, while we only need the center to be feasible to satisfy the validity conditions. Furthermore, if $\Phi_{\boldsymbol{\theta}}(\mathbf{0}) \notin \mathcal{K}_{\boldsymbol{\theta}}$, we can enforce validity by defining a shifted INN as $\Phi_{\boldsymbol{\theta}}'(\cdot) = \Phi_{\boldsymbol{\theta}}(\cdot) - \Phi_{\boldsymbol{\theta}}(\mathbf{0}) + \mathbf{x}^{\circ}$ given an interior point $\mathbf{x}^{\circ} \in \mathcal{K}_{\boldsymbol{\theta}}$. Such an interior/feasible point requirement for worst-case feasibility guarantees aligns with existing works on non-convex constrained optimization (Barber & Ha, 2018; Lin et al., 2022).

## 3.3 Hom-PGD$^+$: Projected Gradient Descent with INN

| **Algorithm 1** Hom-PGD$^+$ | **Algorithm 2** BP Operator |
|---|---|
| **Input:** initial point $\mathbf{z}_0$, valid INN $\Phi_{\boldsymbol{\theta}}$, reformulated optimization problem, and total number of iterations $K$ | **Input:** input point $\mathbf{z}$, lower bound $\beta_l = 0$, upper bound $\beta_u = 1$, and max iterations $B$ |
| **for** $k = 0$ to $K$ **do** | **for** $t = 0$ to $B$ **do** |
|     Compute stepsize $\alpha_k$ |     Bisection $\beta_m = (\beta_l + \beta_u)/2$ |
|     **Update:** $\mathbf{z}_{k+1} = \mathrm{BP}_{\tilde{\mathcal{B}}}\left(\mathbf{z}_k - \alpha_k \nabla H_{\boldsymbol{\theta}}(\mathbf{z}_k)\right)$ |     **Update:** if $\Phi_{\boldsymbol{\theta}}(\beta_m \cdot \mathbf{z}) \in \mathcal{K}_{\boldsymbol{\theta}}$ then $\beta_l \leftarrow \beta_m$ |
| **end for** |     **else** $\beta_u \leftarrow \beta_m$ |
| **Output:** $\mathbf{x}_K = \Phi_{\boldsymbol{\theta}}(\mathbf{z}_K)$ | **end for** |
| | **Output:** $\hat{\mathbf{z}} = \beta_l \cdot \mathbf{z}$ |

In the ideal setting with perfect homeomorphism, we perform standard projected gradient descent (PGD) to problem (**H**) where the constrained set is a unit ball. However, in practice, due to the non-perfect training, the INN homeomorphic mapping is inexact, i.e., $\Phi_{\boldsymbol{\theta}} \neq \psi_{\boldsymbol{\theta}}$, thereby transforming $\mathcal{K}_{\boldsymbol{\theta}}$ into a non-perfect (and a non-convex)[5] ball $\tilde{\mathcal{B}} = \Phi_{\boldsymbol{\theta}}^{-1}(\mathcal{K}_{\boldsymbol{\theta}})$. To clarify the reformulated optimization problem we address, we denote the reformulated version induced by the INN as follows:

$$\min_{\mathbf{z}} H_{\boldsymbol{\theta}}(\mathbf{z}), \quad \text{s.t. } \mathbf{z} \in \tilde{\mathcal{B}}. \qquad (\mathbf{H}_{\mathrm{inn}})$$

---

[5]Here "non-perfect ball" means the learned ball $\tilde{B}$ is just an approximate ball, i.e., the shape is close to a unit ball, thus might exhibit non-convexities (e.g., see Fig. 1).

where $H_{\boldsymbol{\theta}} = f_{\boldsymbol{\theta}} \circ \Phi_{\boldsymbol{\theta}}$. It is worth noting that the orthogonal projection onto the approximate ball $\tilde{\mathcal{B}}$ is computationally challenging. To mitigate this, we employ a bisection-based projection operator to approximate the orthogonal projection in each iteration, formally defined below.

**Definition 3.3** (Bisected projection)**.** The bisected projection operator $\mathrm{BP}_{\tilde{\mathcal{B}}}(\mathbf{z})$ for $\mathbf{z} \in \mathbb{R}^n$ is as $\mathrm{BP}_{\tilde{\mathcal{B}}}(\mathbf{z}) \in \mathrm{segment}(\mathbf{oz}) \cap \partial\tilde{\mathcal{B}}$ for $\mathbf{z} \notin \tilde{\mathcal{B}}$ and $\mathrm{BP}_{\tilde{\mathcal{B}}}(\mathbf{z}) = \mathbf{z}$ for $\mathbf{z} \in \tilde{\mathcal{B}}$, where $\mathbf{o}$ is the origin.

We note the following properties of the bisected projection operator: **(i)** The bisected projection may have multiple solutions when the line segment intersects the boundary $\partial\tilde{\mathcal{B}}$ at multiple points; in such cases, the operator returns one of the valid solutions. **(ii)** The projected solution can be computed efficiently using bisection methods (Alg. 2) with linear convergence rate (Liang et al., 2023). Importantly, each bisection iteration requires a simple feasibility check (i.e., membership oracle queries). **(iii)** When the trained INN satisfies validity conditions (Def. 3.2), the composition $\Phi_{\boldsymbol{\theta}}(\mathrm{BP}_{\tilde{\mathcal{B}}}(\mathbf{z}))$ guarantees feasibility in $\mathcal{K}_{\boldsymbol{\theta}}$ for any $\mathbf{z} \in \mathbb{R}^n$.

We then apply the PGD with the bisection projection operator for the INN-transformed problem $\mathbf{H}_{\mathrm{inn}}$ (shown in Alg. 1). Finally, we map the obtained converged solution back to the original space to recover the corresponding solution for the original problem.

## 4 PERFORMANCE ANALYSIS

In this section, we present a comprehensive performance analysis for Hom-PGD$^+$, including the landscape analysis, convergence rate, and run-time complexity.

**General Assumptions and Notations** (with details in Appendix C.2): *For notational simplicity, we fix the input parameter $\boldsymbol{\theta}$ and omit it, writing $f$ in place of $f_{\boldsymbol{\theta}}(\cdot)$, and similarly for other functions and mappings.*

- The objective $f$ and each constraint function $g_i$ ($i \in [m]$) are $L_{f,0}$-Lipschitz ($L_{g_i,0}$ resp.) continuous, and $L_f$-smooth ($L_{g_i}$ resp.).
- The homeomorphic mapping $\psi$ is invertible, bi-Lipschitz continuous, and has a non-singular, Lipschitz continuous Jacobian matrix, denoted by $\mathrm{J}_{\psi}$.

Given a compact constrained set $\mathcal{K}$, these global conditions can be relaxed to hold on a compact domain. See Appendix C.2 for detailed explanations. *We remark that the learned INN $\Phi$ inherently satisfies the same assumptions as $\psi$, including bi-Lipschitz continuity and the existence of the Jacobian, by design of the INN architecture (refer to Appendix B.3).* Moreover, the composited function $H = f \circ \Phi$ and $G_i = g_i \circ \Phi$ for $i \in [m]$ inherit the same regularization properties as $f$ and $g_i$ from Lemma D.1. Specifically, we make further assumptions in the following.

- The learned INN is $(l_{\Phi}, u_{\Phi})$-bi-Lipschitz continuous and $L_{\Phi}$-smooth.
- The composited functions $H = f \circ \Phi$ and $G_i = g_i \circ \Phi$ ($i \in [m]$) are $L_{H,0}$-Lipschitz ($L_{G_i,0}$ resp.) continuous, and $L_H$-smooth ($L_{G_i}$ resp.).

In addition, we make the following assumption related to the learned INN.

**Assumption 2** (INN Approximation Error Bound)**.** We assume the INN-approximated homeomorphic mapping $\Phi : \mathbb{R}^n \to \mathbb{R}^n$ has (i) a bounded approximation error:

$$\mathcal{B}(0, 1 - \epsilon_{\mathrm{inn}}) \subseteq \Phi^{-1}(\mathcal{K}) \subseteq \mathcal{B}(0, 1 + \epsilon_{\mathrm{inn}}), \|\psi - \Phi\| \leq \epsilon_{\mathrm{inn}},$$

and (ii) a bounded Jacobian approximation error: $\|\mathrm{J}_{\psi} - \mathrm{J}_{\Phi}\| \leq \epsilon_{\mathrm{inn}}$.

The bounded INN approximation error could be made due to the training manner. Specifically, we design the INN $\Phi$ to map the ball $\mathcal{B}$ closely onto the constraint set $\mathcal{K}$, a behavior enforced by the loss function in Eq. (1). When $\Phi(\mathcal{B})$ approximates $\mathcal{K}$ well, it closely mimics the true homeomorphism $\psi$. However, controlling the Jacobian approximation error is a stronger condition, but this assumption is pivotal in our analysis to bound the KKT solution gap. In practice, since the ground truth mapping $\psi$ is unavailable, we incorporate Lipschitz regularization (i.e., spectral norm of INN Jacobian) into the training loss to reduce local sensitivities of $\Phi$.

### 4.1 LANDSCAPE ANALYSIS

In this subsection, we analyze the landscape of $\mathbf{H}$ under the homeomorphic transformation. The following lemma establishes a one-to-one correspondence between KKT stationary points (Def. D.2) of $\mathbf{P}$ and $\mathbf{H}$, where the relevant definitions and the proofs are provided in Appendix D.3.

**Proposition 4.1.** *Suppose the strict complementary condition holds for both problem* **P** *and* **H***. Then* $\mathbf{x}^*$ *is a first-order, second-order and non-degenerate KKT stationary point of* **P** *if and only if* $\mathbf{z}^*$ *is a corresponding KKT stationary point of* **H** *where* $\mathbf{z}^* = \boldsymbol{\psi}(\mathbf{x}^*)$.

The significance of this proposition lies in its ability to establish a fundamental equivalence between the solution properties of two distinct formulations of an optimization problem. Specifically, it guarantees that optimality conditions under the Karush-Kuhn-Tucker framework are preserved under a homeomorphic transformation.

### 4.2 CONVERGENCE ANALYSIS

**Definition 4.2** (Approximate KKT stationary point). A point $\mathbf{x}^*$ is said to be an $\epsilon$-approximate KKT stationary point of **P** if there exists $\boldsymbol{\lambda}^* \in \mathbb{R}^n_{\geq 0}$ such that

$$\left\|\nabla f(\mathbf{x}^*) + \sum_{i=1}^m \lambda_i^* \nabla g_i(\mathbf{x}^*)\right\| \leq \epsilon, \quad \|[\mathbf{g}(\mathbf{x}^*)]_+\| \leq \epsilon, \quad \sum_{i=1}^m |\lambda_i^* g_i(\mathbf{x}^*)| \leq \epsilon, \tag{2}$$

where we denote $[a]_+ := \max\{a, 0\}$ for a scalar $a \in \mathbb{R}$ and $[\mathbf{a}]_+ := ([a_i]_+)_i$ for a vector $\mathbf{a}$.

The convergence analysis of Hom-PGD$^+$ is as follows, where the proof is deferred to Appendix E.

**Theorem 1** (Convergence of Hom-PGD$^+$). *Let INN* $\Phi$ *satisfy Assumption 2. Then Hom-PGD$^+$ with constant step-size* $\alpha \in (0, \frac{1}{L_H}]$ *can find an* $\epsilon + \mathcal{O}\left(\sqrt{L_H \epsilon_{inn}}\right)$*-approximate KKT stationary point for* **P** *in* $\mathcal{O}\left(L_H \epsilon^{-2}\right)$ *iterations.*

To understand this result's significance, we examine it within the broader context of optimization theory, which presents fundamental difference for convex versus non-convex constraint sets.

For non-convex optimization over **convex constraints**, established methods like PGD and augmented-Lagrangian approaches (Beck, 2014; Zhang et al., 2022; Liu et al., 2025a) achieve $\mathcal{O}(\epsilon^{-2})$ rates. Under perfect INN training ($\epsilon_{\text{inn}} = 0$), our result recovers their result. The additional $\mathcal{O}\left(\sqrt{L_H \epsilon_{\text{inn}}}\right)$ term reflecting INN approximation error, is consistent with optimization under inexact information (Devolder et al., 2014; Barber & Ha, 2018; Liu et al., 2025b).

However, optimization over **non-convex constraints** is significantly more challenging. Existing PGD-like methods require restrictive assumptions such as small local concavity (Barber & Ha, 2018), hidden convexity (Barik et al., 2023; Fatkhullin et al., 2023), or specialized manifold structures (Balashov et al., 2020). Proximal-point-based algorithms have been proposed and analyzed in recent works (Boob et al., 2019; Ma et al., 2019; Lin et al., 2022), demonstrating complexity bounds of $\tilde{\mathcal{O}}(\epsilon^{-3})$ to find a stationary point under non-singular assumptions, and $\tilde{\mathcal{O}}(\epsilon^{-4})$ without them.

Our key insight is that the ball-homeomorphic structure bridges this complexity gap. While $\mathcal{K}$ may be highly non-convex, the homeomorphic mapping enables convex optimization techniques in the transformed space. This assumption is more natural than existing restrictive conditions and broadly applicable across machine learning and engineering domains, as discussed in Sec. 1.

Consequently, Theorem 1 achieves convex-like $\mathcal{O}(\epsilon^{-2})$ rates for non-convex constrained problems—a significant theoretical advance. Additionally, the dependence on $L_H = u_\Phi^2 L_f + L_\Phi L_{f,0}$ is related to the forward Lipschitz $u_\Phi$ (22) of the INN (Lemma D.1). Thus, the Lipschitz-regularized INN training scheme in Sec. 3.2 can accelerate the convergence rate by a constant factor.

### 4.3 RUN-TIME COMPLEXITY

We analyze the total runtime complexity of the Hom-PGD$^+$ method. The INN training process incurs a one-time computational cost that is performed offline and does not impact real-time performance. During the online phase, when a specific parameter $\boldsymbol{\theta}$ is provided, the pre-trained mapping $\Phi_{\boldsymbol{\theta}}$ can be directly utilized. Detailed discussion on the offline complexity of INN training is included in Appendix B.5. The following discussion focuses on the online complexity of the Hom-PGD$^+$ method.

**Oracles.** In Hom-PGD$^+$, we will use the following oracles. (i) *Zeroth-order and first-order oracle:* Given a point, a zeroth-order oracle returns the value of a function $f$, whereas a first-order oracle provides the gradient of $f$. (ii) *Membership oracle:* Given a point $\mathbf{x} \in \mathbb{R}^n$, this oracle $\mathcal{M}_{\mathcal{K}}(\mathbf{x}) := \mathbb{I}(\mathbf{x} \in \mathcal{K}) : \mathbb{R}^n \to \{0, 1\}$ returns 1 if and only if $\mathbf{x} \in \mathcal{K}$. Generally, the membership oracle is more efficient than the optimization oracle (Mhammedi, 2022), particularly for non-convex constraint sets.

**Basic operations in Hom-PGD$^+$.** Next, we provide the complexity of computing basic operators where we denote $W$ as the size of the trained INN (with details in Appendix B.3).

- *Computing* $\mathrm{BP}_{\mathcal{U}}(\cdot)$ *:* $\tilde{\mathcal{O}}(W \log 1/\epsilon)$. The bisected projection can be computed using Alg. 2. As shown in (Liang et al., 2023), the method enjoys a linear convergence rate. In each iteration, it requires one forward pass through the INN and $\tilde{\mathcal{O}}(1)$ query to the membership oracle for $\mathcal{K}_{\boldsymbol{\theta}}$.

- *Computing gradient of* $h$: $\mathcal{O}(W)$. The gradient can be computed by chain rule $\nabla h(\mathbf{z}) = \mathrm{J}_{\Phi}(\mathbf{z})^{\top} \nabla f(\mathbf{x})$. The Jacobian of $\Phi$ can be obtained through back propagation with cost $\mathcal{O}(W)$.

**Total run-time complexity of Hom-PGD$^+$.** Given a trained INN $\Phi$, the complexity includes:

- *Per-iteration complexity*. Each iteration requires gradient computation as $\nabla h(\mathbf{z}) = \mathrm{J}_{\Phi}(\mathbf{z})^{\top} \nabla f(\mathbf{x})$ and computation of homeomorphic bisected projection both with complexity $\tilde{\mathcal{O}}(W)$.

- *Last-step complexity*. The final converged solution in the transformed space is mapped back to the original space via $\Phi$ with complexity $\mathcal{O}(W)$ for a forward propagation.

- *Number of iterations* (I). Refer to Sec. 4.2 for the convergence analysis.

In conclusion, the total complexity of Hom-PGD$^+$ equals $\mathcal{O}(W \cdot \mathrm{I})$. Empirically, we choose a 3-layer INN with $\mathcal{O}(n)$ width, which exhibits strong performance and efficiency, and leads to complexity of $W = \mathcal{O}(n^2)$. This practical complexity is lower than that of second-order methods (with $\mathcal{O}(n^3)$ per-iteration cost), highlighting the scalability of Hom-PGD$^+$ to high-dimensional problems.

### 4.4 Extending beyond Ball-homeomorphic Constraint

While this work assumes that the constraint set is homeomorphic to a ball, our framework can, in principle, be extended to general compact non-convex sets, albeit with a potentially large optimality gap. **(i)** For non-BH constrained sets, one can still train an INN to learn an invertible mapping from the unit ball to a **subset** of the constraint set that is itself ball-homeomorphic (ideally, the largest subset via volume maximization) following the loss function in Sec 3.2. **(ii)** The Hom-PGD$^+$ (Alg. 1) can be directly applied to the reformulated problem without any modification under the valid INN condition. **(iii)** The convergence rate of Theorem 1 still holds, but the stationary point corresponds to the restricted problem over the subset. Consequently, the optimality gap with respect to the original problem cannot be directly quantified.

## 5 Empirical Study

We conduct extensive experiments to demonstrate the efficiency of Hom-PGD$^+$. **(i)** We evaluate Hom-PGD$^+$ on quadratically constrained quadratic programming (QCQP) problems. **(ii)**, we scaling the QCQP problem dimension and compare Hom-PGD$^+$ with industrial solver on scalability. **(iii)** We consider real-world power grid optimization under uncertainty with joint chance constraints (JCC). **(iv)** We conduct ablation studies including INN complexity and optimality gaps. Detailed experimental settings, problem formulation, data generation, baseline description, and supplementary results are provided in Appendices F and G.

**Baselines**: For non-convex constrained optimization problems, we consider the following baselines following the state-of-the-art work considering optimization over non-convex constrained sets (Lin et al., 2022). (i) **EPM** (Cartis et al., 2011): *exact penalty methods* iteratively solve subproblems by adding a penalty for constraint violations to the objective. (ii) **ALM** (Sahin et al., 2019; Xie & Wright, 2019; Birgin et al., 2003): *augmented Lagrangian methods* for problem **P** that alternately update primal and dual variables for an unconstrained Lagrangian formulation. (iii) **PPP** (Lin et al., 2022): *proximal-point penalty method* iteratively solves subproblems by augmenting the objective with a proximal term and quadratic penalty terms. (iv) **Hom-PGD$^+$** shown in Sec. 3.

### 5.1 Illustrative Examples of Hom-PGD$^+$ for Non-convex QCQP

As shown in Fig. 2, in the randomly generated non-convex QCQP instances, our Hom-PGD$^+$ method achieves fast convergence compared to other first-order algorithms. In terms of running time, compared to methods requiring expensive inner minimization problems such as Lagrangian or proximal-point methods, we only need bisection to project infeasible solutions back to the transformed constraint set, reaching linear convergence with low complexity through membership oracle queries.

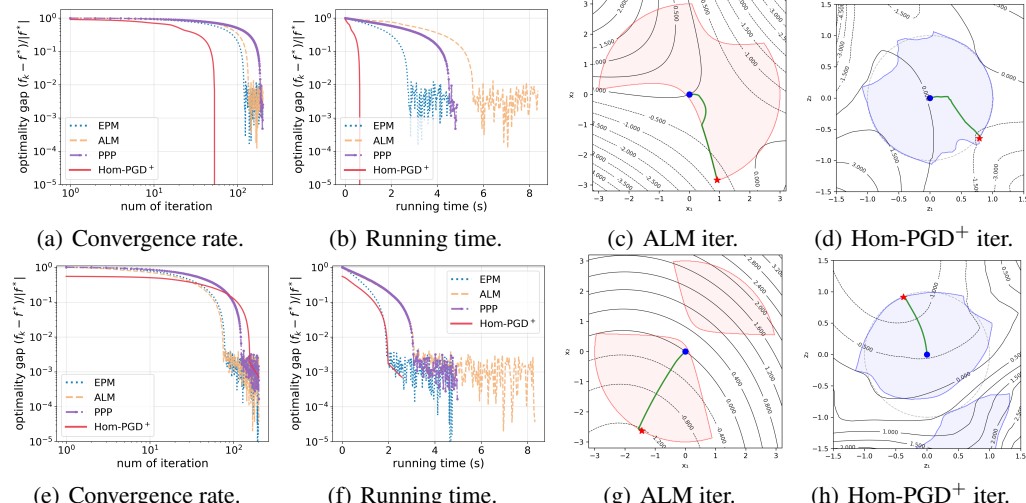

(a) Convergence rate.    (b) Running time.    (c) ALM iter.    (d) Hom-PGD$^+$ iter.

(e) Convergence rate.    (f) Running time.    (g) ALM iter.    (h) Hom-PGD$^+$ iter.

Figure 2: Illustrative examples of Hom-PGD$^+$ for solving QCQP, including non-convex BH and non-BH constraints. The optimality gap is evaluated over the IPOPT solver. One INN is trained to map the unit ball to the constraint set under different input parameters (with details in Appendix. G.1). Hom-PGD$^+$ convergence under various inputs is included in the Appendix. G.3.

We train one INN to transform the constraint set under different input parameters and deploy it for optimization, *amortizing* the homeomorphism construction complexity across different constraints and reducing online complexity. Furthermore, our method empirically works for non-BH constraint settings as long as the valid INN conditions hold, despite lacking tight theoretical bounds.

## 5.2 HOM-PGD$^+$ *vs* IPOPT IN HIGH-DIMENSIONAL NON-CONVEX QCQP

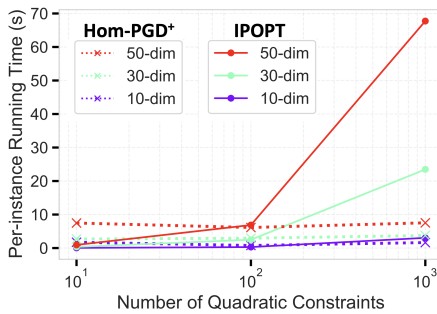

(a) Solution-Time Scaling in $m$ and $n$.

(b) Objective Value Comparison.

| $(n, m)$ | $(10, 10)$ | $(10, 100)$ | $(10, 1000)$ |
|---|---|---|---|
| IPOPT | -1.481 | -0.941 | -1.377 |
| **Hom-PGD$^+$** | -1.446 | -0.927 | -1.295 |
| $(n, m)$ | $(30, 10)$ | $(30, 100)$ | $(30, 1000)$ |
| IPOPT | -0.751 | -0.829 | -0.699 |
| **Hom-PGD$^+$** | -0.737 | -0.811 | -0.682 |
| $(n, m)$ | $(50, 10)$ | $(50, 100)$ | $(50, 1000)$ |
| IPOPT | -0.665 | -0.635 | -0.602 |
| **Hom-PGD$^+$** | -0.634 | -0.620 | -0.590 |

Figure 3: Scalability analysis of INN-PGD$^+$ with respect to problem dimensions-number of constraints $m \in \{10, 100, 1000\}$ and number of variables $n \in \{10, 30, 50\}$. The problem dimensions scale with as $\mathcal{O}(m \cdot n^2)$. (a) shows average per-instance solving time when scaling $m$ and $n$, while (b) shows the average converged objective values.

We scale our method to high-dimensional QCQP problems (which may be non-homeomorphic) along two axes: the number of decision variables $n$ and the number of quadratic constraints $m$, yielding $\mathcal{O}(n^2 \cdot m)$ problem parameters. Hom-PGD$^+$ demonstrates superior scaling compared to the well-optimized second-order industrial solver IPOPT. As $m$ increases by two orders of magnitude ($10 \rightarrow 1000$), IPOPT's per-instance time grows steeply—most notably for $n = 50$, where runtime jumps from 3 to 70 seconds. In contrast, Hom-PGD$^+$ exhibits near-constant runtime as $m$ scales and only mild growth with $n$, owing to efficient GPU-accelerated INN computation and batched constraint verification. Solution quality remains competitive: Hom-PGD$^+$ achieves an average objective gap of 2.9% on average with zero constraint violations. These results demonstrate that Hom-PGD$^+$ maintains efficiency as problem size grows, while IPOPT's computational cost escalates rapidly, particularly for large $n$ and $m$.

Table 2: Performance comparison over JCC optimal power flow on PGLIB `200`- and `500`-bus systems with `100` and `1000` uncertainty scenarios. (Obj., Vio., Time) denote the objective value, constraint violation, and inference time (in seconds), respectively. GUROBI is applied to compute the optimum with equivalent mixed-integer formulations in `3,600 seconds`. All baseline methods are executed in a maximum of `100` iterations.

| Power Gird | 200-bus | | | | | | 500-bus | | | | | |
|---|---|---|---|---|---|---|---|---|---|---|---|---|
| Scenarios | 100 | | | 1000 | | | 100 | | | 1000 | | |
| Metrics | Obj. | Vio. | Time | Obj. | Vio. | Time | Obj. | Vio. | Time | Obj. | Vio. | Time |
| GUROBI | 0.679 | 0 | 95 | *failed* | | | 7.43 | 0 | 1259 | *failed* | | |
| EPM | 0.690 | 0.9 | 76 | 0.933 | 1 | 801 | 8.63 | 1 | 109 | 8.65 | 1 | 1107 |
| ALM | 0.693 | 0.9 | 141 | 0.927 | 1 | 1452 | 8.66 | 1 | 205 | 8.67 | 1 | 2061 |
| PPP | 0.698 | 0.9 | 75 | 0.927 | 1 | 799 | 8.62 | 1 | 108 | 8.66 | 1 | 1102 |
| **Hom-PGD$^+$** | 0.688 | 0 | **44** | 0.768 | 0 | **246** | 7.66 | 0 | **103** | 8.56 | 0 | **396** |

## 5.3 NON-CONVEX JCC-OPTIMIZATION FOR POWER GRID OPERATION

Modern power grids face uncertainties from renewable generation and load fluctuations, requiring operators to determine generator settings that ensure safe operation with high probability. This problem can be modeled as non-convex joint chance constraints (JCC), which are computationally prohibitive for large-scale grids when solved exactly with mixed-integer formulations (Pagnoncelli et al., 2009). The computational challenge arises from integer variables scaling with scenarios and numerous operational constraints per scenario (exceeding 2,000 for the 500-bus grid).

Our method demonstrates strong performance on this challenging problem. As shown in Table 2, we significantly outperform baselines in running time while maintaining approximately 3% optimality gap compared to GUROBI and achieving exact chance constraint satisfaction. This efficiency stems from our bisection-based projection algorithm requiring only function evaluation (membership oracle) without gradient calculations for constraints, unlike other first-order methods that require both evaluations at each iteration, with computational burden growing linearly with scenarios.

## 5.4 ABLATION STUDY AND SENSITIVITY ANALYSIS

With details in Appendix G.2, we conduct the following analysis: **(i)** *INN Complexity and Performance*, showing the impact of INN complexity (e.g., 1/3/5-layer INN) on approximation error (2) and its Lipschitz constants, as well as the impacts on the downstream optimization task, showing that the 3-layer INN balances the approximation capability and parameter complexity. **(ii)** *Bisection Complexity and Performance*, showing that reducing the iterations of the bisection algorithm can further reduce the per-iteration cost, while it may incur a large optimality gap.

## 6 CONCLUSION AND LIMITATIONS

In this work, we proposed Hom-PGD$^+$, a fast projection-efficient, learning-based method for optimizing over non-convex constraint sets homeomorphic to a ball. Exploiting the constraint topological structure, we leverage INN to transform the problem and achieve efficient convergence with low per-iteration cost, outperforming existing methods both theoretically and empirically across various benchmarks. Despite the efficiency of Hom-PGD$^+$, several **limitations** remain for future work: (i) Learning homeomorphic mappings via INNs introduces significant worst-case theoretical complexity. Developing tighter approximation bounds for learning homeomorphisms could improve practical efficiency. (ii) Our convergence guarantee yields an $\epsilon + \mathcal{O}(\sqrt{\epsilon_{\text{inn}}})$-approximate stationary point. This square-root dependence for homeomorphism approximation error $\epsilon_{\text{inn}}$ may be suboptimal, and achieving a tighter relationship remains an open question. (iii) While designed for Euclidean ball-homeomorphic constraints, our framework may extend to manifold-constrained problems with favorable topology, though formalizing such extensions remains non-trivial.

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

# Contents

LLM USAGE

Large Language Models (LLMs) were used to aid in the writing and polishing of the manuscript.

## A    RELATED WORK

Non-convex optimization is notoriously challenging and is NP-hard in general. To better understand its structure and design more efficient algorithms, researchers have explored strong structural assumptions that enable convergence, sometimes even to global optima, as well as advanced techniques such as reparameterization and hidden convexity. We review these developments in the following sections.

### A.1    CONDITIONS FOR GLOBAL CONVERGENCE IN NON-CONVEX OPTIMIZATION

**Invexity.** Invexity (Hanson, 1981) is a generalization of convexity, with a property that stationary points are global optima (Martin, 1985; Ben-Israel & Mond, 1986). The classical theory of invexity is detailed in (Mishra & Giorgi, 2008). Recent work (Barik et al., 2023) develops projected invex gradient descent algorithms that find global optima for invex programs under certain assumptions. Additionally, the invex structure has been applied to learning tasks, such as image reconstruction (Pinilla et al., 2022; Pinilla & Thiyagalingam, 2024), to achieve global optima instead of merely critical points.

**PL/KL conditions.** Kurdyka-Łojasiewicz (KL) condition (Lojasiewicz, 1963a; Kurdyka, 1998) is widely used to analyze local convergence in non-convex minimization. The Polyak-Łojasiewicz (PL) condition (Polyak, 1963; Lojasiewicz, 1963b), a global variant of the KL condition, ensures that stationarity implies optimality and serves as a sufficient condition for global linear convergence in non-convex problems. This condition has been applied to non-convex, non-smooth optimization (Bento et al., 2024) and learning tasks such as training neural networks (Reddi et al., 2016; Lei et al., 2019) and stochastic risk minimization (Foster et al., 2018). Theoretical studies have explored the relationship between (generalized) PL and other conditions (Karimi et al., 2016), the calculus of KL functions (Li & Pong, 2018), and convergence rates for functions satisfying the KL condition with varying exponents (Frankel et al., 2015).

**Quasar-convexity.** Quasar-convexity (Hardt et al., 2018) is a relaxation of convexity parameterized by $\gamma \in (0, 1]$, with $\gamma = 1$ implying star-convexity. This property arises in various optimization and learning tasks such as the objectives in, learning linear dynamical systems (Hardt et al., 2018), positive semidefinite matrix completion (Ge et al., 2016), and neural network training tasks (Zhou et al., 2019; Kleinberg et al., 2018). For quasar-convex objectives, gradient-based methods can achieve a comparable convergence rate as convex objectives to a global optimum, with convergence analyses available for standard algorithms (Gower et al., 2021; Guminov et al., 2017) and accelerated methods (Guminov et al., 2017; Hinder et al., 2020; Nesterov et al., 2018a; Fu et al., 2023).

### A.2    NON-CONVEX CONSTRAINED OPTIMIZATION

For optimization problems with non-convex constraints, convergence guarantees for standard PGD algorithms are rarely provided. The existing literature often imposes extremely stringent conditions, such as assumptions on local concavity coefficients (Barber & Ha, 2018) or adopts a manifold optimization framework (Balashov et al., 2020; Balashov, 2021; Boumal, 2023).

In fact, convergence analysis for non-convex constrained optimization is generally scarce and frequently relies on inconsistent or overly restrictive assumptions, not just for projection-based algorithms but across other approaches as well. To address these challenges, several works have proposed alternative methodologies, including regularized subgradient methods (Ma et al., 2020), inexact Lagrangian augmented methods (Sahin et al., 2019; Xie & Wright, 2019; Birgin et al., 2003) and proximal-point-based algorithms (Boob et al., 2019; Ma et al., 2019; Lin et al., 2022). Among these works, the state-of-the-art work Lin et al. (2022) achieves the fastest convergence rate $\mathcal{O}(\epsilon^{-3})$ for non-convex optimization problems with weakly convex constraints, under some regularization assumption. We refer readers to this paper for a comprehensive discussion of the assumptions and convergence analysis in related work.

## A.3 Recent Advances for Non-Convex Optimization

To reduce the cost and accelerate the convergence for solving (non-)convex constrained optimization, recent novel projection-free methods and other advanced techniques involve inexact projection, radial dual formulation, reparameterizing optimization problems, and uncovering hidden convexity.

**Inexact projection**. In many cases, the projection operator lacks an analytic solution or is computationally expensive to compute exactly, motivating the analysis of inexact projected methods. For convex optimization, such methods achieve the same convergence rate as PGD if the cumulative projection error is bounded (Schmidt et al., 2011; Patrascu & Necoara, 2018), with new results derived under specific settings (Patrascu & Irofti, 2021). For nonconvex objectives with convex constraints, their convergence has been analyzed in (Birgin et al., 2003; Wang & Liu, 2006; Zhang et al., 2020). Recent advances further generalize inexact projection operators to broader settings (Ferreira et al., 2022; Aguiar et al., 2023).

**Radial duality.** Beyond classical projection-free methods, recent advancements have introduced novel approaches based on gauge and radial duality theory. Radial duality theory for nonnegative optimization problems (Grimmer, 2024a;b) demonstrates that constrained optimization problems can be reformulated as unconstrained problems using the gauge of their constraints. This framework has led to the development of new families of projection-free methods with optimal convergence guarantees (Liu & Grimmer, 2023), as well as relaxed conditions (Samakhoana & Grimmer, 2024) that enable more efficient line search operators for the reformulated unconstrained problems.

**Reparameterization.** Reparameterizing optimization problems aims to mitigate challenging properties, such as non-smoothness or non-convexity, via invertible transformations while preserving equivalent optima. Parameterization is widely used in optimization and learning tasks, including semi-definite programming (Cifuentes, 2021), low-rank optimization (Mishra et al., 2014; Ha et al., 2020), and risk minimization (Bah et al., 2022). Recent advancements include parameterizing simplex (Li et al., 2023) and polyhedron (Tang & Toh, 2024) optimization via Hadamard transformation to reduce projection complexity, smooth over-parameterization to accelerate non-smooth optimization algorithms (Poon & Peyré, 2023), parameterizing discrete data as continuous for generative learning (Davis et al., 2024), and analyzing the optimization landscape under parameterization transformations in non-convex settings (Levin et al., 2024).

**Hidden convexity.** Hidden convexity refers to transformations that reveal the convex structure of non-convex sets or functions, which has been exploited in problems such as rotation matrix optimization (Ramachandran et al., 2024), non-linear least squares (Drusvyatskiy & Paquette, 2019), revenue management and inventory control (Chen et al., 2022), and quadratically constrained quadratic programming (QCQP) with Toeplitz-Hermitian quadratics (Konar & Sidiropoulos, 2015). For non-convex stochastic optimization with hidden structure, projected gradient-based algorithms can achieve the same convergence rate as in convex optimization for both strongly convex (Fatkhullin et al., 2023) and convex objectives (Chen et al., 2022) under certain assumptions. Furthermore, QCQP, which is generally NP-hard, can be solved in polynomial time when hidden convexity is present (Konar & Sidiropoulos, 2015).

# B  LEARNING HOMEOMORPHISM VIA INVERTIBLE NEURAL NETWORKS

In this section, we provide the omitted details in Sec. 2 and 3.

## B.1  HANDLING CONSTRAINT SET WITH EQUALITY

We first explain how to handle equality constraints as mentioned in Sec. 2. Consider the constrained set $\mathcal{K}_{\boldsymbol{\theta}}$ as follows

$$\mathcal{K}_{\boldsymbol{\theta}} = \{\mathbf{x} \mid \mathbf{q}_{\boldsymbol{\theta}}(\mathbf{x}) = 0, g_{1,\boldsymbol{\theta}}(\mathbf{x}) \leq 0, \cdots, g_{m,\boldsymbol{\theta}}(\mathbf{x}) \leq 0\}$$

where $\mathbf{q} = (q_1, q_2, \cdots, q_{m_{\text{eq}}})$ with continuous functions $q_{i,\boldsymbol{\theta}}(\mathbf{x}) : \mathbb{R}^n \to \mathbb{R}$ with respect to $\mathbf{x}$ and $\boldsymbol{\theta}$.

Suppose the rank of the equality constrained function is constant for all $\mathbf{x} \in \mathcal{K}_{\boldsymbol{\theta}}$, i.e.,

$$\text{rank}\left(\mathbf{J}_{\mathbf{q}}(\mathbf{x})\right) = r, \quad \forall \mathbf{x} \in \mathcal{K}_{\boldsymbol{\theta}}.$$

Then $\{\mathbf{q}_{\boldsymbol{\theta}}(\mathbf{x}) = \mathbf{0}\}$ is of dimension $n-r$ by the Constant-Rank Level Set Theorem (Lee & Lee, 2012). In other words, we can use a subset of decision variables $\mathbf{x}_1 \in \mathbb{R}^{n-r}$ and reconstruct full decision variable $[\mathbf{x}_1, \mathbf{x}_2] \in \mathbb{R}^n$ via the equality constraint, where $\mathbf{x}_2 = \boldsymbol{\phi}_{\boldsymbol{\theta}}(\mathbf{x}_1)$ and $\mathbf{q}_{\boldsymbol{\theta}}([\mathbf{x}_1, \boldsymbol{\phi}_{\boldsymbol{\theta}}(\mathbf{x}_2)]) = \mathbf{0}$. Such a reconstruction process ensures the feasibility of the equality constraint. Hence, the constraint $\mathcal{K}_{\boldsymbol{\theta}}$ can be reformulated as

$$\mathcal{K}_{\boldsymbol{\theta}}^s = \{\mathbf{x}_1 \in \mathbb{R}^{n-r} \mid g_{1,\boldsymbol{\theta}}(\mathbf{x}_1, \boldsymbol{\phi}_{\boldsymbol{\theta}}(\mathbf{x}_1)) \leq 0, \cdots, g_{m,\boldsymbol{\theta}}(\mathbf{x}_1, \boldsymbol{\phi}_{\boldsymbol{\theta}}(\mathbf{x}_1)) \leq 0\}.$$

It follows from the reconstruction that

$$(\mathbf{x}_1, \mathbf{x}_2 = \boldsymbol{\phi}_{\boldsymbol{\theta}}(\mathbf{x}_2)) \in \mathcal{K}_{\boldsymbol{\theta}} \Leftrightarrow \mathbf{x}_1 \in \mathcal{K}_{\boldsymbol{\theta}}^s.$$

It is noteworthy that the constant rank assumption for $\mathbf{q}_{\boldsymbol{\theta}}(\cdot)$ holds globally for linear equalities and locally for nonlinear manifold equalities (see, e.g., (Lee, 2010; Boumal, 2023)), which encompasses a majority of practical optimization applications. Based on the foregoing analysis, this paper assumes that the constrained set $\mathcal{K}_{\boldsymbol{\theta}}$ includes only equality constraints. For a detailed discussion on managing linear equalities, nonlinear inequalities, and manifold equalities, the reader is referred to Appendix A and Appendix B in (Liang et al., 2023).

## B.2  INTRODUCTION OF INVERTIBLE NEURAL NETWORKS

The INN $\Phi : \mathbb{R}^n \to \mathbb{R}^n$ is a class of neural networks that is a continuous bijection. It is a finite composition of invertible layers, where each layer is also a homeomorphic mapping with tunable parameters. In the following, we introduce several commonly used invertible layers for INN, and refer readers to (Papamakarios et al., 2021) for a more comprehensive introduction. *Moreover, denote $\mathcal{H}^n$ the set of homeomorphisms from $\mathbb{R}^n$ to $\mathbb{R}^n$.*

- **Linear layer** (Kingma & Dhariwal, 2018). The invertible linear layer is defined as

$$\text{Forward}: \quad \mathbf{x}' = \mathbf{W}\mathbf{x} + \mathbf{b}, \quad \text{Inverse}: \quad \mathbf{x} = \mathbf{W}^{-1}(\mathbf{x}' - \mathbf{b})$$

  where $\mathbf{W} \in \mathbb{R}^{n \times n}, \mathbf{b} \in \mathbb{R}^n$ are matrices with tunable entries. Further, by the LU decomposition, the invertible matrix is designed as $\mathbf{W} = \mathbf{W}_P \mathbf{W}_L (\mathbf{W}_U + \text{diag}(\mathbf{s}))$, where $\mathbf{W}_P$ is a fixed permutation matrix, $\mathbf{W}_L$ is a lower triangular matrix, $\mathbf{W}_U$ is an upper triangular matrix, and $\mathbf{s} \in \mathbb{R}^n$ is the diagonal elements. The singular values of the invertible matrix are $|\mathbf{s}|$.

- **Coupling layer**. The coupling layer first randomly splits the input into two parts as $\mathbf{x} = \left(\mathbf{x}_{\leq k} \in \mathbb{R}^k, \mathbf{x}_{>k} \in \mathbb{R}^{n-k}\right)$ and the transformation is defined as

$$\text{Forward}: \quad \mathbf{x}'_{\leq k} = \mathbf{x}_{\leq k}, \mathbf{x}'_{>k} = \mathbf{s}\left(\mathbf{x}_{>k}; \mathbf{t}\left(\mathbf{x}_{\leq k}\right)\right),$$

$$\text{Inverse}: \quad \mathbf{x}_{\leq k} = \mathbf{x}'_{\leq k}, \mathbf{x}_{>k} = \mathbf{s}^{-1}\left(\mathbf{x}'_{>k}; \mathbf{t}\left(\mathbf{x}'_{\leq k}\right)\right)$$

  where $\mathbf{t} : \mathbb{R}^k \to \mathbb{R}^k$ is an arbitrary DNN and $\mathbf{s} : \mathbb{R}^{n-k} \times \mathbb{R}^k \to \mathbb{R}^{n-k}$ is an invertible map w.r.t. its first argument given the second, i.e., $\mathbf{s}(\cdot, \mathbf{y})$ is invertible for fixed $\mathbf{y}$. One particular choice is the affine coupling layer (Dinh et al., 2014) if $\mathbf{t} : \mathbb{R}^k \to \mathbb{R}^{n-k} \times \mathbb{R}^{n-k}$:

$$\mathbf{s}(\mathbf{a}; \mathbf{b}) = \mathbf{a} \odot \mathbf{b}_1 + \mathbf{b}_2, \text{ for } \mathbf{b}_1 \neq 0, \quad \text{and} \quad \mathbf{b} = \mathbf{t}(\mathbf{y}) = (\boldsymbol{\gamma}(\mathbf{y}), \boldsymbol{\tau}(\mathbf{y}))$$

  where $\boldsymbol{\gamma} > 0, \boldsymbol{\tau} : \mathbb{R}^k \to \mathbb{R}^{n-k}$ are two learnable NNs, $\odot$ denotes the element-wise product. To keep $\boldsymbol{\gamma} > 0$, one selection is $\boldsymbol{\gamma}(\mathbf{y}) = \exp \boldsymbol{\phi}(\mathbf{y})$ where $\boldsymbol{\phi} : \mathbb{R}^k \to \mathbb{R}^{n-k}$ is a regular NN and the operation $\exp$ is applied element-wise.

- **Residual layer** (Behrmann et al., 2019; Chen et al., 2019). The invertible residual layer is defined as
$$\text{Forward}: \quad \mathbf{x}' = \mathbf{x} + \mathbf{r}(\mathbf{x}) \quad \text{with} \quad \text{Lip}(\mathbf{r}) < 1,$$
$$\text{Inverse}: \quad \text{via the iteration} \quad \mathbf{x}^{(i+1)} = \mathbf{x}' - \mathbf{r}(\mathbf{x}^{(i)}) \quad \text{with} \quad \mathbf{x}^{(0)} = \mathbf{x}',$$
where $\mathbf{r} : \mathbb{R}^n \to \mathbb{R}^n$ is an arbitrary NN. The inverse process is computed iteratively through a fixed-point iteration scheme. Owing to the Lipschitz constraint, the fixed-point iteration is guaranteed to converge when $t \to \infty$, thus ensuring the invertibility of the residual layer. The log-determinant of this layer can be approximated by the power series (Behrmann et al., 2019).

- **Neural ODE layer** (Chen et al., 2018; Grathwohl et al., 2018). The ODE invertible layer is defined as
$$\text{Forward}: \quad \mathbf{x}' = \mathbf{x} + \int_0^1 \boldsymbol{\varphi}(\mathbf{x}, t)\mathrm{d}t, \quad \text{Inverse}: \quad \mathbf{x} = \mathbf{x}' + \int_0^{-1} \boldsymbol{\varphi}(\mathbf{x}', t)\mathrm{d}t,$$
where $\boldsymbol{\varphi}(\cdot, \cdot) : \mathbb{R}^n \times \mathbb{R} \to \mathbb{R}^n$ represents a time-dependent vector field. The forward and inverse processes are both computed based on integration, ensuring that the system is invertible.

- **Convex potential layer** (Huang et al., 2020).
$$\text{Forward}: \quad \mathbf{x}' = \nabla F(\mathbf{x}), \quad \text{Inverse}: \quad \mathbf{x} = \arg\min_{\mathbf{y}}\{F(\mathbf{y}) - \mathbf{y}^\top \mathbf{x}'\},$$
where $F : \mathbb{R}^n \to \mathbb{R}$ denotes a strongly convex function. The inverse process is computed by iteratively solving the optimization problem. Because of the strictly convex property of $F$, the solution for the inverse process is unique.

**Remark.** In this work, we follow the GLOW architecture (Kingma & Dhariwal, 2018) for INN design, which consists of a composition of finite affine coupling layers and invertible linear layers. Specifically, an $l$-layer INN is defined as
$$\Phi = \Phi^l \circ \Phi^{l-1} \cdots \circ \Phi^1$$
where each layer $\Phi^j = f_{\text{coup}}^j \circ \mathcal{L}^j$ ($j \in [l]$) consists of an invertible linear transformation $\mathcal{L}^j(\mathbf{x}) = \mathbf{Q}_j\mathbf{x}$ for some rotation matrix $\mathbf{Q}_j$ and a coupling layer $f_{\text{coup}}$ of fixed splitting strategy $k = \lfloor n/2 \rfloor$.

This structure offers several key advantages: (i) it admits closed-form forward and inverse computations through neural network propagation, (ii) it enables closed-form calculation of Jacobian singular values, which are essential for computing the log-determinant and Lipschitz constant required in our INN loss function, and (iii) affine coupling layers are universal approximators for any differentiable homeomorphism (Teshima et al., 2020). Given these theoretical and computational advantages, we adopt the coupling layer-based INN architecture for our framework.

### B.3 Computational Issues of Invertible Neural Networks

In this section, we analyze the computational issues of INNs $\Phi$. There are several requirements for the Invertible Neural Network (INN):

- (i) The forward and inverse mappings of the INN must be efficiently computable, as they are required to map solutions between the original space and the transformed space within Hom-PGD$^+$.
- (ii) The Jacobian of the INN must be computable, as it is essential for evaluating the gradient of the composite function $H = f \circ \Phi$ in the Hom-PGD$^+$ algorithm.
- (iii) The singular values of the Jacobian matrix must be accessible, as they are necessary for estimating terms in the loss function defined in Eq. (7) during the INN training process.
- (iv) The INN should have bounded distortion to ensure the worst-case performance for homeomorphic projection. Furthermore, the INN should be a universal approximator of homeomorphic mappings. This enables it to handle complex transformations involving a broad range of constraints.

*Since this paper adopts the coupling-layer-based INN architecture, we focus our analysis specifically on this type of INN.* For conciseness of notations, we fix $\boldsymbol{\theta}$ and omit it. For an $l$-layer INN denoted as $\Phi = \Phi^l \circ \cdots \circ \Phi^j \circ \cdots \circ \Phi^1$, we denote $\mathbf{x}^j = \Phi^{j-1}(\mathbf{x}^{j-1})$ for $j = 2, \cdots, l$ and $\mathbf{x}^1 = \mathbf{x}$. Moreover, we denote $W$ as the size (number of parameters) of an INN.

(i) In each affine coupling layer, the forward and inverse could be computed directly by the definition, i.e., for $\mathbf{x} = (\mathbf{x}_1 \in \mathbb{R}^{n_1}, \mathbf{x}_2 \in \mathbb{R}^{n_2})$ with $n_1 + n_2 = n$ and two arbitrary NNs $\boldsymbol{\gamma} > 0, \boldsymbol{\tau} : \mathbb{R}^{n_1} \to \mathbb{R}^{n_2}$, we have

$$\begin{aligned} \text{Forward}: \quad & (\mathbf{y}_1, \mathbf{y}_2) = (\mathbf{x}_1, \mathbf{x}_2 \odot \boldsymbol{\gamma}(\mathbf{x}_1) + \boldsymbol{\tau}(\mathbf{x}_1)), \\ \text{Inverse}: \quad & (\mathbf{x}_1, \mathbf{x}_2) = (\mathbf{y}_1, (\mathbf{y}_2 - \boldsymbol{\tau}(\mathbf{y}_1))/\boldsymbol{\gamma}(\mathbf{y}_1)) \end{aligned} \quad (3)$$

where $/$ is applied element-wise to vector computation. For the conditional layer, we augment the input parameters $\boldsymbol{\theta}$ as, $\boldsymbol{\gamma}_{\boldsymbol{\theta}}(\cdot)$ and $\boldsymbol{\tau}_{\boldsymbol{\theta}}(\cdot)$. *Therefore, the complexity of computing $\Phi$ and $\Phi^{-1}$ is $\mathcal{O}(W)$.*

(ii) The Jacobian of such a composited mapping and its determinant can be expressed as

$$\mathrm{J}_{\Phi}(\mathbf{x}) = \prod_{j=1}^{l} \mathrm{J}_{\Phi^j}\left(\mathbf{x}^j\right), \quad |\det \mathrm{J}_{\Phi}(\mathbf{x})| = \prod_{j=1}^{l} \left|\det \mathrm{J}_{\Phi^j}\left(\mathbf{x}^j\right)\right|.$$

For each affine coupling layer, the Jacobian can be expressed as

$$\frac{\partial \mathbf{y}}{\partial \mathbf{x}} = \left[ \begin{array}{cc} \mathbf{I}_{n_1} & 0 \\ \frac{\partial \mathbf{y}_2}{\partial \mathbf{x}_1} & \mathrm{diag}\left(\boldsymbol{\gamma}(\mathbf{x}_1)\right) \end{array} \right],$$

where $\mathrm{diag}(\mathbf{v})$ returns a diagonal matrix whose diagonal elements are given by the vector $\mathbf{v}$. *It follows that the complexity of computing $\mathrm{J}_{\Phi}(\mathbf{x})$ is $\mathcal{O}(W)$.*

(iii) For each layer, the Jacobian determinant can be expressed as the product of singular values:

$$\left|\det \mathrm{J}_{\Phi^j}\left(\mathbf{x}^j\right)\right| = \prod_{i=1}^{n} \sigma_i\left(\mathrm{J}_{\Phi^j}\left(\mathbf{x}^j\right)\right)$$

where $\sigma_1(\cdot) \geq \ldots \geq \sigma_n(\cdot) > 0$ are the sorted singular values of the Jacobian matrix of the mapping $\Phi^j(\cdot)$ at $\mathbf{x}$. By the design of each affine coupling layer, such an invertible transformation has a closed-form expression of singular values, which is $1$ or elements of $\boldsymbol{\gamma}(\mathbf{x}_1)$. *Therefore, the complexity to compute the determinant or singular values of an coupling layer INN is still $\mathcal{O}(W)$.*

(iv) The bounded distortion property of an INN constructed with affine coupling layers is inherently guaranteed by its architectural design. Moreover, its universal approximation capability for homeomorphic mappings over compact domains has been established in the existing literature. These two properties are formally stated below.

**Proposition B.1.** *Suppose $\Phi$ is an INN composed of affine coupling layers. Then:*

*(i) $\Phi$ is capable of approximating any $n$-dimensional differentiable homeomorphism over a compact domain, given a sufficiently large number of layers (Jin et al., 2024; Liang et al., 2024; Ishikawa et al., 2022).*

*(ii) $\Phi$ exhibits bounded distortion, where the bound depends on the number of layers (Liang et al., 2024).*

### B.4 Unsupervised INN Training

We denote

$$\mathcal{H}^n := \{\boldsymbol{\phi} : \mathbb{R}^n \to \mathbb{R}^n \mid \boldsymbol{\phi} \text{ is a homeomorphism}\}, \mathcal{H}^n(\mathcal{K}_{\boldsymbol{\theta}}, \mathcal{B}) := \{\boldsymbol{\psi} \in \mathcal{H}^n \mid \boldsymbol{\psi}(\mathcal{B}) = \mathcal{K}_{\boldsymbol{\theta}}\}.$$

Moreover, the feasible set $\mathcal{H}^n(\mathcal{K}_{\boldsymbol{\theta}}, \mathcal{B})$ is equivalent to the set of optimal solutions to the problem (Liang et al., 2023; 2024):

$$\max_{\boldsymbol{\psi}_{\boldsymbol{\theta}} \in \mathcal{H}^n} \log \mathrm{V}\left(\boldsymbol{\psi}_{\boldsymbol{\theta}}(\mathcal{B})\right) \quad \text{s.t. } \boldsymbol{\psi}_{\boldsymbol{\theta}}(\mathcal{B}) \subseteq \mathcal{K}_{\boldsymbol{\theta}} \quad (4)$$

where $\mathrm{V}\left(\boldsymbol{\psi}_{\boldsymbol{\theta}}(\mathcal{B})\right)$ computes the volume of set $\boldsymbol{\psi}_{\boldsymbol{\theta}}(\mathcal{B})$ and the constraint means that the set $\boldsymbol{\psi}_{\boldsymbol{\theta}}(\mathcal{B})$ is a subset of $\mathcal{K}_{\boldsymbol{\theta}}$. While there might be multiple homeomorphisms in the set $\mathcal{H}^n(\mathcal{K}_{\boldsymbol{\theta}}, \mathcal{B})$ (e.g., through composition with rotations over the ball, we get an additional such homeomorphism), we wish to learn one with minimum Lipschitz constant. To this end, we define the Lipschitz constant of a mapping $\boldsymbol{\psi}$ over a set $\mathcal{K}$ as

$$\mathrm{L}(\boldsymbol{\psi}) = \sup_{\mathbf{z} \neq \mathbf{u} \in \mathcal{K}} \frac{\|\boldsymbol{\psi}(\mathbf{z}) - \boldsymbol{\psi}(\mathbf{u})\|}{\|\mathbf{z} - \mathbf{u}\|}. \quad (5)$$

Intuitively, the minimum Lipschitz homeomorphical (MLH) mapping problem can be reformulated to the following bi-level problem:

$$\min_{\boldsymbol{\psi}_{\boldsymbol{\theta}} \in \mathcal{H}^n} \log \mathrm{L}\left(\boldsymbol{\psi}_{\boldsymbol{\theta}}\right) \text{ s.t. } \boldsymbol{\psi}_{\boldsymbol{\theta}} \in \arg\max\{ \text{ Problem in (4) }\}. \tag{6}$$

We employ the following loss function and maximize it to train an INN $\Phi_{\boldsymbol{\theta}}$ with $l$ layers for learning the homeomorphic mapping $\boldsymbol{\theta}$ in an unsupervised manner:

$$\mathcal{L}\left(\Phi_{\boldsymbol{\theta}}\right) = \widehat{\mathrm{V}}\left(\Phi_{\boldsymbol{\theta}}(\mathcal{B})\right) - \lambda_1 \mathrm{P}\left(\Phi_{\boldsymbol{\theta}}(\mathcal{B})\right) - \lambda_2 \widehat{\mathrm{L}}\left(\Phi_{\boldsymbol{\theta}}\right) \tag{7}$$

where $\lambda_1$ and $\lambda_2$ are positive coefficients to balance among the three terms. For ease of analysis of how to compute the three terms, we denote an $l$-layer INN as $\Phi_{\boldsymbol{\theta}} = \Phi_{\boldsymbol{\theta}}^l \circ \ldots \circ \Phi_{\boldsymbol{\theta}}^2 \circ \Phi_{\boldsymbol{\theta}}^1$, where each layer is either a bi-Lip affine coupling layer or an invertible linear layer.

(i) $\widehat{\mathrm{V}}\left(\Phi_{\boldsymbol{\theta}}(\mathcal{B})\right)$ is a computable approximation of the log-volume term $\log \mathrm{V}\left(\Phi_{\boldsymbol{\theta}}(\mathcal{B})\right)$ in (4) as:

$$\widehat{\mathrm{V}}\left(\Phi_{\boldsymbol{\theta}}(\mathcal{B})\right) = \frac{1}{\mathrm{V}(\mathcal{B})} \int_{\mathcal{B}} \sum_{i=1}^{n} \sum_{j=1}^{l} \log \sigma_i \left( \mathrm{J}_{\Phi_{\boldsymbol{\theta}}^j}\left(\mathbf{z}^j\right) \right) \mathrm{d}\mathbf{z} + \log \mathrm{V}(\mathcal{B}) \tag{8}$$

where $\mathbf{z}^j = \Phi_{\boldsymbol{\theta}}^{j-1}\left(\mathbf{z}^{j-1}\right)$ for $j = 2, \cdots, l$, and $\mathbf{z}^1 \in \mathcal{B}$, $\mathrm{J}_{\Phi_{\boldsymbol{\theta}}^j}\left(\mathbf{z}^j\right)$ denotes the Jacobian matrix of $\Phi_{\boldsymbol{\theta}}^j(\cdot)$ at $\mathbf{z}^j$.

(ii) $\mathrm{P}\left(\Phi_{\boldsymbol{\theta}}(\mathcal{B})\right)$ is the penalty term for the constraint violation of $\Phi_{\boldsymbol{\theta}}(\mathcal{B}) \subseteq \mathcal{K}_{\boldsymbol{\theta}}$ in (4) as:

$$\mathrm{P}\left(\Phi_{\boldsymbol{\theta}}(\mathcal{B})\right) = \int_{\mathcal{B}} \|\mathrm{ReLU}\left(\mathbf{g}\left(\Phi_{\boldsymbol{\theta}}(\mathbf{z}), \boldsymbol{\theta}\right)\right)\|_1 \ \mathrm{d}\mathbf{z}, \tag{9}$$

where $\mathrm{ReLU}(\cdot) = \max\{0, \cdot\}$ and $\mathbf{g}\left(\Phi_{\boldsymbol{\theta}}(\mathbf{z}), \boldsymbol{\theta}\right)$ calculates the residual for each inequality constraint as $[g_1\left(\Phi_{\boldsymbol{\theta}}(\mathbf{z}), \boldsymbol{\theta}\right), \ldots, g_m\left(\Phi_{\boldsymbol{\theta}}(\mathbf{z}), \boldsymbol{\theta}\right)]$.

(iii) $\widehat{\mathrm{L}}\left(\Phi_{\boldsymbol{\theta}}^{-1}, \mathcal{K}_{\boldsymbol{\theta}}\right)$ is a computable approximation of the log-Lipschitz term $\log \mathrm{L}\left(\Phi_{\boldsymbol{\theta}}^{-1}, \mathcal{K}_{\boldsymbol{\theta}}\right)$ as:

$$\widehat{\mathrm{L}}\left(\Phi_{\boldsymbol{\theta}}\right) = \sup_{\mathbf{z}^1 \in \mathcal{Z}_{\boldsymbol{\theta}}} \left\{ \sum_{j=1}^{l} \log \sigma_1 \left( \mathrm{J}_{\Phi_{\boldsymbol{\theta}}^j}\left(\mathbf{z}^j\right) \right) \right\} \tag{10}$$

where $\mathbf{z}^j = \Phi_{\boldsymbol{\theta}}^{j-1}\left(\mathbf{z}^{j-1}\right)$ for $j = 2, \cdots, l$, and $\mathbf{z}^1 \in \mathcal{Z}_{\boldsymbol{\theta}} = \Phi_{\boldsymbol{\theta}}^{-1}\left(\mathcal{K}_{\boldsymbol{\theta}}\right)$.

We have the following bounds for the approximations (Liang et al., 2023; 2024). The two approximation terms in (8) and (10) satisfy $\log \mathrm{V}\left(\Phi_{\boldsymbol{\theta}}(\mathcal{B})\right) \geq \widehat{\mathrm{V}}\left(\Phi_{\boldsymbol{\theta}}(\mathcal{B})\right)$ and $\log \mathrm{L}\left(\Phi_{\boldsymbol{\theta}}\right) \leq \widehat{\mathrm{L}}\left(\Phi_{\boldsymbol{\theta}}\right)$.

The above proposition implies that the loss function in (7) is actually a lower bound to the Lagrangian of the problem in (6). Therefore, we can maximize the loss function in (7) to approximate the MLH mapping under the equivalent reformulation in (6). Further, to train one conditional INN $\Phi \in \mathcal{H}^{n+d}$ to learn the $\boldsymbol{\theta}$-dependent MLH mappings for any $\boldsymbol{\theta} \in \Theta$, we generalize the loss in (7) to

$$\mathcal{L}(\Phi) = \mathbb{E}_{\boldsymbol{\theta}}\left[\mathcal{L}\left(\Phi_{\boldsymbol{\theta}}\right)\right]$$

where $\boldsymbol{\theta} \in \Theta$ is uniformly sampled. For the INN training, we prepare quasi Monte Carlo (QMC) samples $\{\mathbf{z}_i\}_{i=1}^{N} \subset \mathcal{B}$ to approximate the integration in (8) and (9). When evaluating the distortion in (10), since we may not know $\mathcal{Z}_{\boldsymbol{\theta}}$ in advance, we sample from $\mathcal{Z}_{\boldsymbol{\theta}} = \Phi_{\boldsymbol{\theta}}^{-1}\left(\mathcal{K}_{\boldsymbol{\theta}}\right) \subset \mathcal{B}$ over a unit ball as $\{\mathbf{z}_i\}_{i=1}^{N}$. In each iteration, we sample a batch of collected data and employ the Adam optimizer to maximize the loss function $\mathcal{L}(\Phi)$, similar to training standard NNs (Kingma & Ba, 2014).

## B.5 Offline Complexity to Obtain a Trained Valid INN

In this section, we will discuss the theoretical complexity of obtaining a trained, valid INN $\Phi_{\boldsymbol{\theta}}$ which approximates $\boldsymbol{\psi}_{\boldsymbol{\theta}}$ where $\boldsymbol{\psi}_{\boldsymbol{\theta}}(\mathcal{B}) = \mathcal{K}_{\boldsymbol{\theta}}$ for the optimization $\mathbf{P}$.

**Complexity of obtaining a valid INN.** To obtain a valid invertible neural network (INN) $\Phi_{\boldsymbol{\theta}} \approx \boldsymbol{\psi}_{\boldsymbol{\theta}}$ *given a ball-homeomorphic constrained set* $\mathcal{K}_{\boldsymbol{\theta}}$, one must incur the following cost.

- Training. Training a neural network is an unconstrained non-convex optimization, which is NP-hard to find a global optimum in general. In practice, we use Adam optimizer to maximize the loss function, similar to the process of training regular NNs (Kingma & Ba, 2014). Typically, the run-time is poly($\epsilon^{-1}$) to find an approximate stationary solution.

- #Samples of $\mathcal{B}$. As discussed in Sec. B.4, one will prepare samples $\{\mathbf{z}_i\} \subset \mathcal{B}$ to approximate the integration (8), (9) and (10) using QMC. The integration error for the QMC approach is $\mathcal{O}\left((\log N)^{n-1}/N\right)$ where $N$ is the number of samples, which is faster in the rate of convergence than Monte Carlo using a pseudorandom sequence Dick & Pillichshammer (2010).

- INN size. For the INN size to approximate a bi-continuous $n$-dimensional homeomorphism to an error $\epsilon$, the theoretical upper bound $\mathcal{O}(\epsilon^{-n})$ derived from (Jin et al., 2024) is high due to the worst-case analysis. Meanwhile, the lower bound is an open question so far. Note that the theoretical bound of INN size is high and grows exponentially with the input dimension $n$ due to a worst-case analysis. However, in practice, the target homeomorphism may be much simpler, requiring significantly fewer parameters for the INN to approximate it effectively. For instance, in our empirical study, we found that approximately three coupling layers with width $\mathcal{O}(n)$ are sufficient to learn the homeomorphic mapping from a non-convex set to a ball.

**Remark.** Although training the INN offline incurs additional computational cost, this expense is only one-time and can be amortized over numerous online problem instances. Moreover, modern deep learning frameworks, such as PyTorch coupled with GPU acceleration, render the training process efficient (e.g., less than 10 minutes for high-dimensional chance-constrained problems). Once the INN is appropriately trained, the framework achieves a convergence rate comparable to optimization over convex constraint sets $\left(\mathcal{O}\left(\epsilon^{-2}\right)\right)$ with a low per-iteration cost, significantly improving on state-of-the-art rates of $\mathcal{O}\left(\epsilon^{-4}\right)$ or $\mathcal{O}\left(\epsilon^{-3}\right)$ under regularity conditions (see Table 1 for details).

In practice, it is often necessary to verify whether a constrained set is homeomorphic to a ball. This question can generally be divided into two cases:

(i) *Special cases with known topological properties*. Certain sets are naturally homeomorphic to a ball, such as compact convex sets (Geschke, 2012; Bredon, 2013) and star-shaped sets (Appendix B.6). In particular, for compact convex sets, an explicit ball-homeomorphic mapping can be directly constructed using the gauge mapping, as discussed in Liu et al. (2025a). For star-shaped sets, a ball-homeomorphic mapping can also be constructed; however, it may depend on certain unknown parameters specific to the star-shaped set. As a result, it is often more practical to use an INN to approximate the homeomorphic mapping. Further details are provided in Appendix B.6.

(ii) *General non-convex sets*. For general compact non-convex constrained sets, we may apply topological data analysis (TDA) (Chazal & Michel, 2021; Otter et al., 2017) to determine whether the set satisfies the ball-homeomorphic property. The method is described below.

**Verify whether $\mathcal{K}_{\boldsymbol{\theta}} \cong \mathcal{B}$?** It is a classical result that a compact, contractible set of dimension $n \geq 6$ with a simply connected boundary is homeomorphic to a ball (Smale, 1962). Therefore, to verify whether $\mathcal{K}_{\boldsymbol{\theta}} \cong \mathcal{B}$, one can examine the presence of any "holes" in $\mathcal{K}_{\boldsymbol{\theta}}$ for $\boldsymbol{\theta} \in \Theta$. In practice, persistent homology (Chazal & Michel, 2021; Otter et al., 2017), a widely used technique in topological data analysis, provides an effective means of performing this verification.

- Sample complexity (#samples of $\mathcal{K}_{\boldsymbol{\theta}}$). To detect the absence of holes in the set $\mathcal{K}_{\boldsymbol{\theta}}$ (for a fixed $\boldsymbol{\theta}$) with diameters smaller than $\epsilon$, the number of required samples is given by the $\epsilon$-covering number of $\mathcal{K}$ Chazal & Michel (2021), which is of order $\mathcal{O}(\exp(n))$.

- Run-time. Given the samples of $\mathcal{K}_{\boldsymbol{\theta}}$, the run-time of persistent homology methods is of order poly(#Samples) (Otter et al., 2017).

**Remark.** While verifying the ball-homeomorphism property through sampling and topological data analysis can be computationally expensive, explicit verification is often unnecessary in practice. Many common constraint sets—including convex and star-shaped sets—possess known topological properties that naturally guarantee ball-homeomorphism.

More generally, our method can be applied whenever the *valid INN condition* (Definition 3.2) is satisfied, which requires only that *the INN maps the center of the unit ball to a feasible point in the*

*constraint set*. As discussed in Section 4.4, our theoretical guarantees (feasibility preservation and convergence rate) hold under this valid INN condition alone.

This makes ball-homeomorphism verification a *sufficient but not necessary* prerequisite—the valid INN condition provides a more practical and verifiable criterion that can be easily checked without expensive topological analysis. In essence, practitioners need only verify that their trained INN satisfies the valid INN condition, which is straightforward to evaluate through simple feasibility checking.

### B.6 HOMEOMORPHISMS FROM A STAR-SHAPED SET TO A BALL

**Definition B.2** (Star-shaped set). A set is called a *star-shaped* set if it has the property that all interior and boundary points are visible from a point $\mathbf{x}^\circ$ (called *star center*) in the set. Note that the set of star centers of a star-shaped set might have multiple and even infinite elements.

For the geometric, analytical, combinatorial and topological properties of star-shaped sets, and their broad applicability in many mathematical fields, we refer readers to (Hansen et al., 2020) for a comprehensive discussion and review.

Importantly, a star-shaped set is homeomorphic to a unit ball. The formal statement is given below, where one could refer to, e.g., Page 60 (Gonnord & Tosel, 1998) and Theorem 237 of the handbook *Analysis III* by Dirk Ferus, for its proof.

**Proposition B.3.** *Open star-shaped sets are diffeomorphic to open balls, where a diffeomorphism is a smooth homeomorphism.*

For a star-shaped set $\mathcal{S}$, using $\mathbf{x}^\circ$ as the center, one can construct an explicit homeomorphism $\psi$ that continuously and bijectively sends points in $\mathcal{S}$ to points in a unit ball $\mathcal{B}$. Such a homeomorphism is termed a gauge mapping (Tabas & Zhang, 2022) defined below.

**Definition B.4** (Gauge mapping). Suppose $\mathcal{S}$ is a star-shaped set with star center $\mathbf{x}^\circ$. Let $\gamma_\mathcal{S}(\mathbf{x}, \mathbf{x}^\circ) = \inf\{\lambda \geq 0 \mid \mathbf{x} \in \lambda(\mathcal{S} - \mathbf{x}^\circ)\}$ be the Gauge/Minkowski function (Blanchini & Miani, 2008) given a star center $\mathbf{x}^\circ \in \mathrm{int}(\mathcal{S})$. The gauge mapping $\psi : \mathcal{B} \to \mathcal{S}$ is defined between a unit ball and a compact star-shaped set:

$$\psi(\mathbf{z}) = \frac{\|\mathbf{z}\|}{\gamma_\mathcal{S}(\mathbf{z}, \mathbf{x}^\circ)}\mathbf{z} + \mathbf{x}^\circ, \ \forall \mathbf{z} \in \mathcal{B}; \qquad \psi^{-1}(\mathbf{x}) = \frac{\gamma_\mathcal{S}(\mathbf{x} - \mathbf{x}^\circ, \mathbf{x}^\circ)}{\|\mathbf{x} - \mathbf{x}^\circ\|}(\mathbf{x} - \mathbf{x}^\circ), \ \forall \mathbf{x} \in \mathcal{S}. \quad (11)$$

*Remark* B.5. In Liu et al. (2025a), the gauge mapping is constructed as a homeomorphism between the unit ball and a compact convex set. A key distinction in this setting is that, for compact convex sets, the gauge mapping consistently maps boundary points of the set to boundary points of the unit ball. In contrast, when the gauge mapping is applied to a star-shaped set, boundary points of the set may be mapped to interior points of the unit ball. A visualization of this behavior is provided in Fig. 4. Nevertheless, the gauge mapping remains a well-defined homeomorphism between the star-shaped set and the unit ball.

Based on the explicit construction of homeomorphisms between the unit ball and a star-shaped set, the gauge mapping can be efficiently computed by evaluating the gauge function using a bisection-based algorithm [Hom-PGD]. Moreover, it is important to note that the above construction depends on the center of a star-shaped set. However, in general, finding a star center of a star-shaped set is very challenging and can be NP-hard (O'Rourke & Supowit, 1983; Lee & Lin, 1986). In such cases, one can utilize an INN to learn the ball-homeomorphic mapping directly as discussed in Sec. 3, avoiding the need to verify whether the star-shaped set is ball-homeomorphic.

## C PRELIMINARIES FOR TECHNICAL PROOF

In this section, we summarize the related basic concepts, notations, assumptions, and fundamental propositions and lemmas.

### C.1 BASIC CONCEPTS

We list the basic concepts used in this paper below.

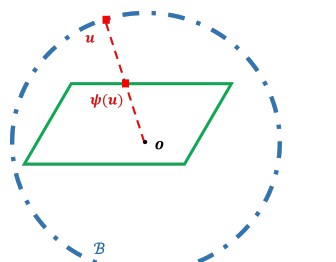 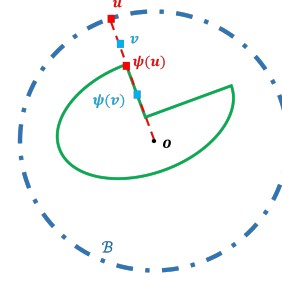

Figure 4: *Illustration of the gauge mapping between the unit ball and a convex set (left) versus a star-shaped set (right).* In the left figure, where the target set is convex, the gauge mapping consistently maps boundary points (resp. interior points) of the unit ball to boundary points (resp. interior points) of the convex set. In contrast, the right figure shows a star-shaped set with star center $\mathbf{o}$; here, the gauge mapping may map an interior point $\mathbf{v} \in \mathcal{B}$ to a boundary point $\psi(\mathbf{v})$ of the star-shaped set.

- Distance between a point and a set. For a closed set $\mathcal{X} \in \mathbb{R}^n$ and any $\mathbf{x} \in \mathbb{R}^n$, the distance between $\mathbf{x}$ and $\mathcal{X}$ is defined as $\mathrm{dist}(\mathbf{x}, \mathcal{X}) = \inf_{\mathbf{y} \in \mathcal{X}} \|\mathbf{x} - \mathbf{y}\|$.

- Orthogonal projection. For a closed set $\mathcal{X}$, the orthogonal projection of a point $\mathbf{x} \in \mathbb{R}^n$ onto $\mathcal{X}$ is defined as $\Pi_{\mathcal{X}}(\mathbf{x}) \in \arg\min_{\mathbf{y} \in \mathcal{X}} \|\mathbf{x} - \mathbf{y}\|$.

- Function convexity. For a differentiable function $f : \mathcal{X} \subseteq \mathbb{R}^n \to \mathbb{R}$, it is said to be convex if one of the following holds:

  1) Jensen's inequality. For $\theta$ with $0 \le \theta \le 1$, we have $f(\theta\mathbf{x} + (1-\theta)\mathbf{y}) \le \theta f(\mathbf{x}) + (1-\theta)f(\mathbf{y})$ for all $\mathbf{x}, \mathbf{y} \in \mathcal{X}$.
  2) first-order condition. $f(\mathbf{y}) \ge f(\mathbf{x}) + \langle \nabla f(\mathbf{x}), \mathbf{y} - \mathbf{x} \rangle, \forall \mathbf{x}, \mathbf{y} \in \mathcal{X}$.
  3) monotone gradient. $(\nabla f(\mathbf{x}) - \nabla f(\mathbf{y}))^T (\mathbf{x} - \mathbf{y}) \ge 0$ for all $\mathbf{x}, \mathbf{y} \in \mathcal{X}$.

- $L$-Smoothness. A differentiable function $f : \mathcal{X} \subseteq \mathbb{R}^n \to \mathbb{R}$ is said be $L$-smooth if one of the following holds:

  1) zeroth-order condition. $f(\lambda\mathbf{x} + (1-\lambda)\mathbf{y}) \ge \lambda f(\mathbf{x}) + (1-\lambda)f(\mathbf{y}) - \frac{L}{2}\lambda(1-\lambda)\|\mathbf{y} - \mathbf{x}\|^2$, for all $\mathbf{x}, \mathbf{y} \in \mathcal{X}, \lambda \in [0, 1]$.
  2) first-order condition. $f(\mathbf{y}) \le f(\mathbf{x}) + \langle \nabla f(\mathbf{x}), \mathbf{y} - \mathbf{x} \rangle + \frac{L}{2}\|\mathbf{y} - \mathbf{x}\|^2$, for all $\mathbf{x}, \mathbf{y} \in \mathcal{X}$.
  3) Lipschitz gradient. $\|\nabla f(\mathbf{y}) - \nabla f(\mathbf{x})\| \le L\|\mathbf{y} - \mathbf{x}\|$, for all $\mathbf{x}, \mathbf{y}$.

- Weak convexity. A function $f : \mathbb{R}^d \to \mathbb{R}$ is said to be weakly convex with constant $\ell_f > 0$ if the function $f(\mathbf{x}) + (\ell_f/2)\|\mathbf{x}\|^2$ is convex.

- Jacobian matrix. Suppose $\mathbf{f} : \mathbb{R}^n \to \mathbb{R}^m$ is a function such that each of its first-order partial derivatives exists on $\mathbb{R}^n$. Then the Jacobian matrix of $\mathbf{f}$, denoted $J_{\mathbf{f}} \in \mathbb{R}^{m \times n}$, is defined as $J_{\mathbf{f}} = (\frac{\partial f_i}{\partial x_j})_{ij}$.

- A Hessian of a function $f : \mathbb{R}^n \to \mathbb{R}$ is defined as $\nabla^2 f = (\frac{\partial^2 f}{\partial x_i \partial x_j})_{ij} \in \mathbb{R}^{n \times n}$, if its second-order partial derivatives exist. Moreover, for a mapping $\mathbf{f} : \mathbb{R}^n \to \mathbb{R}^m$ with existed second-order partial derivatives of each component $f_i$ $(i = 1, 2, \cdots, m)$. The Hessian of $\mathbf{f}$ is defined as

$$H(\mathbf{f}) = (\nabla^2 f_1, \cdots, \nabla^2 f_m).$$

## C.2 BASIC ASSUMPTIONS AND NOTATIONS

**Remark.** For conciseness of notation, we fix the input parameter $\boldsymbol{\theta}$ in problem $\mathbf{P}$ (and $\mathbf{H}$) and omit it, by which we write $\psi, \Phi, f, g_i, h, \mathcal{K}$ to replace $\psi_{\boldsymbol{\theta}}, \Phi_{\boldsymbol{\theta}}, f_{\boldsymbol{\theta}}(\cdot), g_{i,\boldsymbol{\theta}}(\cdot), h_{\boldsymbol{\theta}}(\cdot), \mathcal{K}_{\boldsymbol{\theta}}$ respectively. In the following, we make assumptions throughout the paper.

- Assumptions on $f$ and constraints $g_i$ $(i = 1, 2, \cdots, m)$ in problem $\mathbf{P}$:

  1) $f$ is $L_{f,0}$-Lipschitz continuous, i.e., $\|f(\mathbf{x}) - f(\mathbf{y})\| \le L_{f,0}\|\mathbf{x} - \mathbf{y}\|$ for any $\mathbf{x}, \mathbf{y}$.
  2) $f$ in problem $\mathbf{P}$ is differentiable and $L_f$-smooth.

3) $f^* > -\infty$ where $f^* := \min_{\mathbf{x} \in \mathcal{K}} f(\mathbf{x})$.

4) Each $g_i$ is $L_{g_i,0}$-Lipschitz continuous, differentiable, and $L_{g_i}$-smooth.

- Assumptions on the homeomorphic mapping $\boldsymbol{\psi} : \mathbb{R}^n \to \mathbb{R}^n$:

1) $\boldsymbol{\psi}$ is differentiable with non-singular Jacobian $\mathrm{J}_{\boldsymbol{\psi}}(\cdot)$,

2) $\boldsymbol{\psi}$ is $(\kappa_1, \kappa_2)$-bi-Lipschitz continuous for $\kappa_2 \geq \kappa_1 > 0$, i.e.,

$$\kappa_1 \|\mathbf{u} - \mathbf{v}\| \leq \|\boldsymbol{\psi}(\mathbf{u}) - \boldsymbol{\psi}(\mathbf{v})\| \leq \kappa_2 \|\mathbf{u} - \mathbf{v}\|.$$

Then the Jacobian matrix, $\mathrm{J}_{\boldsymbol{\psi}}(\cdot)$ and $\mathrm{J}_{\boldsymbol{\psi}^{-1}}(\cdot)$ will satisfy

$$\|\mathrm{J}_{\boldsymbol{\psi}}(\mathbf{z})\| \leq \kappa_2, \ \ \forall \mathbf{z}, \ \ \ \|\mathrm{J}_{\boldsymbol{\psi}^{-1}}(\mathbf{x})\| \leq \frac{1}{\kappa_1}, \ \forall \mathbf{x}.$$

3) $\boldsymbol{\psi}$ has $L_{\boldsymbol{\psi}}$-Lipschitz continuous Jacobian matrix, i.e.,

$$\|\mathrm{J}_{\boldsymbol{\psi}}(\mathbf{u}) - \mathrm{J}_{\boldsymbol{\psi}}(\mathbf{v})\| \leq L_{\boldsymbol{\psi}} \|\mathbf{u} - \mathbf{v}\|, \ \forall \mathbf{u}, \mathbf{v}.$$

4) $\boldsymbol{\psi}$ has continuous Hessian, i.e.,

$$\mathrm{H}_{\boldsymbol{\psi}}(\mathbf{z}) = (\nabla^2 \boldsymbol{\psi}_1, \cdots, \nabla^2 \boldsymbol{\psi}_n)$$

exists and is continuous.

**Remark.** Given a compact constrained set $\mathcal{K}$, we can relax these global assumptions to hold on a compact domain, including Lipschitz continuity and smoothness. Specifically, we only require $f$ and $\boldsymbol{\psi}$ to be Lipschitz continuous on a compact set containing the feasible constrained set $\mathcal{K}$. The following are detailed explanations. In our convergence analysis of the Hom-PGD$^+$ algorithm, we only require that the composite function $H = f \circ \Phi$ satisfies: (i) $L_H$-smoothness, and (ii) $L_{H,0}$-Lipschitz continuity on the iterates (with both constants depending on the Lipschitz constant of $f$; see Lemma D.1). Since each iterate $\mathbf{z}_k$ is feasible in the ball $\mathcal{B}$, the update $\mathbf{z}_{k+1}^+ = \mathbf{z}_k - \alpha_k \nabla H(\mathbf{z}_k)$ remains in a compact set $\mathcal{M}$ (which contains $\mathcal{B}$) for bounded $\alpha_k$ and $\|\nabla H(\mathbf{z})\|$. Thus, it suffices for $H$ to be smooth and Lipschitz continuous over $\mathcal{M}$, meaning that $f$ need only be Lipschitz continuous on the compact set $\Phi(\mathcal{M}) \supseteq \mathcal{K}$.

In addition, we summarize the commonly used notations in this paper in Table 3.

Table 3: Summary of Notations. The notations shown in the table is for problem $\mathbf{P}$ and we use the same type notations for problem $\mathbf{H}$.

| Notation | Definition |
|---|---|
| $\|\cdot\|$ | $l2$-norm $\|\cdot\|_2$ |
| $\mathcal{B}$ | unit ball centered at 0 |
| $L_{f,0}$ | Lipschitz constant of $f$ |
| $L_f$ | $L_f$-smooth property of $f$ |
| $\mu_f$ | $\mu_f$-strong convexity of $f$ |
| $\kappa_1, \kappa_2$ | bi-Lipschitz constant of $\boldsymbol{\psi}$ |
| $D$ | distortion of $\boldsymbol{\psi}$, i.e., $\kappa_2/\kappa_1$ |
| $L_{\boldsymbol{\psi}}$ | Lipschitz constant of $\mathrm{J}_{\boldsymbol{\psi}}$ |
| $\mathrm{int}(\mathcal{K}), \partial\mathcal{K}$ | the interior, boundary of $\mathcal{K}$ |

### C.3 BASIC FACTS

In this section, we list the fundamental facts we will use in this paper.

**Proposition C.1** (Properties of Orthogonal Projection, see e.g., (Beck, 2014))**.** *The projection operator $\Pi_{\mathcal{C}}$ over a closed and convex set $\mathcal{C}$ satisfies the following properties.*

*1) Optimality condition: $\forall \mathbf{y} \in \mathcal{C}, \ \langle \mathbf{x} - \Pi_{\mathcal{C}}(\mathbf{x}), \mathbf{y} - \Pi_{\mathcal{C}}(\mathbf{x}) \rangle \leq 0$.*

*2) Non-Expansiveness: $\|\Pi_{\mathcal{C}}(\mathbf{x}) - \Pi_{\mathcal{C}}(\mathbf{y})\| \leq \|\mathbf{x} - \mathbf{y}\|$.*

*3) Monotonicity:* $\langle \Pi_C(\mathbf{x}) - \Pi_C(\mathbf{y}), \mathbf{x} - \mathbf{y} \rangle \geq 0$.

We have the following lemma related to $\psi$ to help with the computation.

**Lemma C.2.** *Suppose* $\mathrm{J}_\psi$ *is* $L_\psi$ *Lipschitz, i.e.,* $\|\mathrm{J}_\psi(\mathbf{u}) - \mathrm{J}_\psi(\mathbf{z})\| \leq L_\psi \|\mathbf{u} - \mathbf{z}\|$ *for any* $\mathbf{u}$ *and* $\mathbf{z}$. *Then, we have*

$$\|\psi(\mathbf{u}) - \psi(\mathbf{z}) - \mathrm{J}_\psi(\mathbf{z})(\mathbf{u} - \mathbf{z})\| \leq \frac{L_\psi \|\mathbf{u} - \mathbf{z}\|^2}{2}, \ \forall \mathbf{u}, \mathbf{z}.$$

One can refer to Lemma 1.2.3 (Nesterov et al., 2018b) for the proof.

Next, we list the following rules for basic computation:

- Jacobian equivalence: $\mathrm{J}_{\psi^{-1}}(\mathbf{x}) = \mathrm{J}_\psi^{-1}(\mathbf{z})$ for $\mathbf{z} = \psi(\mathbf{x})$.

- Chain rule for computing gradient of $h = f \circ \psi$:

$$\nabla h(\mathbf{z}) = \mathrm{J}_\psi(\mathbf{z})^\top \nabla f(\psi(\mathbf{z})) = \mathrm{J}_\psi(\mathbf{z})^\top \nabla f(\mathbf{x}).$$

- Chain rule for computing gradient of $f$:

$$\nabla f(\mathbf{x}) = \mathrm{J}_{\psi^{-1}}(\mathbf{x})^\top \nabla h(\mathbf{z}) = \mathrm{J}_\psi^{-1}(\mathbf{z})^\top \nabla h(\mathbf{z}).$$

- Chain rule for computing Hessian of $h = f \circ \psi$:

$$\nabla^2 h(\mathbf{z}) = \mathrm{J}_\psi(\mathbf{z})^\top \nabla^2 f(\psi(\mathbf{z})) \mathrm{J}_\psi(\mathbf{z}) + \sum_{i=1}^{n} \frac{\partial f}{\partial \mathbf{x}_i}(\psi(\mathbf{z})) \nabla^2 \psi_i(\mathbf{z}).$$

# D  LANDSCAPE ANALYSIS

In this section, we provide landscape analysis to understand important relationships between problem **P** and **H**.

## D.1  ACTION OF HOMEOMORPHISM ON A CONSTRAINED SET

Recall that the constrained set is $\mathcal{K} = \{\mathbf{x} \in \mathbb{R}^n \mid \mathbf{g}(\mathbf{x}) \leq 0\}$ with $\mathbf{g} = (g_1, g_2, \cdots, g_m)$ where each $g_i$ $(i = 1, 2, \cdots, m)$ is a continuous function. For problem **H**,

$$\mathcal{B} = \psi^{-1}(\mathcal{K}) = \{\mathbf{z} \in \mathbb{R}^n \mid \psi(\mathbf{z}) \in \mathcal{K}\} = \{\mathbf{z} \in \mathbb{R}^n \mid \mathbf{G}(\mathbf{z}) := \mathbf{g}(\psi(\mathbf{z})) \leq \mathbf{0}\}$$

where $G_i$ is non-convex in general. However, $\mathcal{B}$ is assumed to be convex (actually a ball set) in this paper. One can refer to Fig. 5 for an illustration.

Moreover, we assume there are no redundant inequalities in $\mathcal{K}$, i.e., there is no $g_i$ such that $\mathcal{K} = \{\mathbf{x} \mid \mathbf{g}_{-i}(\mathbf{x}) \leq \mathbf{0}\}$ where $\mathbf{g}_{-i} = (g_1, \cdots, g_{i-1}, g_{i+1}, \cdots, g_m)$. In this case, any feasible point $\mathbf{x}$ satisfying $g_i(\mathbf{x}) = 0$ for some $i$ is on the boundary of the set $\mathcal{K}$. Thus, we have

$$\{\mathbf{x} \in \mathcal{K} \mid g_j(\mathbf{x}) = 0, g_k(\mathbf{x}) \neq 0\} \bigcap \{\mathbf{x} \in \mathcal{K} \mid g_k(\mathbf{x}) = 0, g_j(\mathbf{x}) \neq 0\} = \emptyset$$

for any $k \neq j$. Note $\mathcal{B} = \{\mathbf{z} \mid G_i(\mathbf{z}) \leq 0, i = 1, 2, \cdots, m\} = \{\mathbf{z} \mid \|\mathbf{z}\|^2 \leq 1\}$. Moreover, $\{G_i(\mathbf{z}) \leq 0, i = 1, 2, \cdots, m\}$ also has no redundant constraints by the non-singularity of the Jacobian of $\psi$ and similarly,

$$\{\mathbf{z} \in \mathcal{B} \mid G_j(\mathbf{z}) = 0, G_k(\mathbf{z}) \neq 0\} \bigcap \{\mathbf{z} \in \mathcal{B} \mid G_k(\mathbf{z}) = 0\} = \emptyset$$

for any $j \neq k$. Hence if $\mathbf{z} \in \mathcal{B}$ satisfies $G_i(\mathbf{z}) = 0$ for some $i$, it lies on the boundary of $\mathcal{B}$. Clearly, we have

$$G_i(\mathbf{z}) = \|\mathbf{z}\|^2 - 1 \quad \text{at} \quad \mathbf{z}' \in \partial \mathcal{B}, G_i(\mathbf{z}') = 0, \tag{12}$$

and

$$\nabla G_i(\mathbf{z}) = 2\mathbf{z}, \nabla^2 G_i(\mathbf{z}) = 2\mathbf{I}_n \quad \text{at} \quad \mathbf{z}' \in \partial \mathcal{B}, G_i(\mathbf{z}') = 0. \tag{13}$$

where $\mathbf{I}_n$ is the identity matrix of $n$ by $n$.

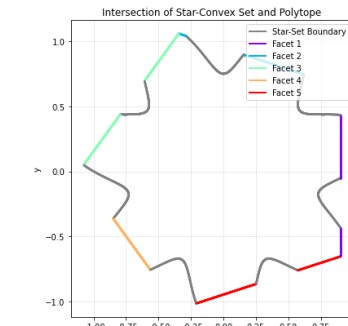 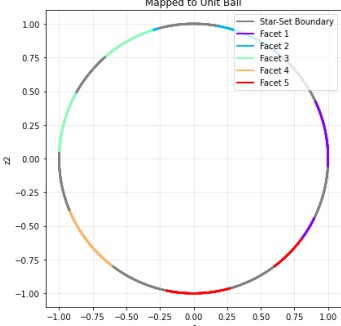

Figure 5: Illustration of the action of homeomorphism on a star-shaped set. The left figure shows the star-shaped constraints of problem $\mathbf{P}$. Each color of line represents the boundary characterized by a constraint inequality $\{\mathbf{a}_i^\top \mathbf{x} \leq b_i\}$ for some $i$. Under a homeomorphic mapping $\boldsymbol{\psi}$, the constrained set is transformed to a ball (right figure). Each constraint inequality $\{G_i(\mathbf{z}) \leq 0\}$ (colored differently) is non-convex in general.

### D.2 PROPERTIES OF FUNCTION $h = f \circ \boldsymbol{\psi}$

**Lemma D.1** (Properties of $h = f \circ \boldsymbol{\psi}$). *Under the general assumptions C.2, $h = f \circ \boldsymbol{\psi}$ has the following properties.*

*1) $h$ is $L_{h,0} := L_{f,0}\kappa_2$ Lipschitz continuous.*

*2) $h$ is $L_h$-smooth with $L_h = \kappa_2^2 L_f + L_{\boldsymbol{\psi}} L_{f,0}$.*

*3) If $f$ is convex, then $h$ is $\ell_h$-weakly convex with $\ell_h = L_{f,0} L_{\boldsymbol{\psi}}$.*

*Proof.* We prove them one by one in the following.

1) We can directly derive from basic definitions:
$$
\begin{aligned}
\|h(\mathbf{u}) - h(\mathbf{v})\| &\leq \|f(\boldsymbol{\psi}(\mathbf{u})) - f(\boldsymbol{\psi}(\mathbf{v}))\| \\
&\leq L_{f,0} \|\boldsymbol{\psi}(\mathbf{u}) - \boldsymbol{\psi}(\mathbf{v})\| \\
&\leq L_{f,0} L_{\boldsymbol{\psi}} \|\mathbf{u} - \mathbf{v}\|.
\end{aligned}
$$

2) From $L_f$-smoothness of $f$, we have
$$
\|\nabla f(\mathbf{x}) - \nabla f(\mathbf{y})\| \leq L_f \|\mathbf{x} - \mathbf{y}\|. \tag{14}
$$
Then we derive with $\mathbf{x} = \boldsymbol{\psi}(\mathbf{z}), \mathbf{v} = \boldsymbol{\psi}(\mathbf{y})$,
$$
\begin{aligned}
\|\nabla h(\mathbf{z}) - \nabla h(\mathbf{v})\| &= \left\| \mathbf{J}_{\boldsymbol{\psi}}(\mathbf{z})^\top \nabla f(\mathbf{x}) - \mathbf{J}_{\boldsymbol{\psi}}(\mathbf{v})^\top \nabla f(\mathbf{y}) \right\| \\
&= \left\| \mathbf{J}_{\boldsymbol{\psi}}(\mathbf{z})^\top (\nabla f(\mathbf{x}) - \nabla f(\mathbf{y})) + (\mathbf{J}_{\boldsymbol{\psi}}(\mathbf{z}) - \mathbf{J}_{\boldsymbol{\psi}}(\mathbf{v}))^\top \nabla f(\mathbf{y}) \right\| \\
&\leq \left\| \mathbf{J}_{\boldsymbol{\psi}}(\mathbf{z})^\top (\nabla f(\mathbf{x}) - \nabla f(\mathbf{y})) \right\| + \left\| (\mathbf{J}_{\boldsymbol{\psi}}(\mathbf{z}) - \mathbf{J}_{\boldsymbol{\psi}}(\mathbf{v}))^\top \nabla f(\mathbf{y}) \right\| \\
&\leq \kappa_2 L_f \|\boldsymbol{\psi}(\mathbf{z}) - \boldsymbol{\psi}(\mathbf{v})\| + L_{\boldsymbol{\psi}} L_{f,0} \|\mathbf{z} - \mathbf{v}\| \\
&\leq \left( \kappa_2^2 L_f + L_{\boldsymbol{\psi}} L_{f,0} \right) \|\mathbf{z} - \mathbf{v}\|.
\end{aligned}
$$
Let $L_h = \kappa_2^2 L_f + L_{\boldsymbol{\psi}} L_{f,0}$. We have the conclusion.

3) One hope to show $h(\cdot) + \frac{\ell_h}{2}\|\cdot + \mathbf{v}\|^2$ is a convex function, i.e.,
$$
h(\mathbf{v}) + \frac{\ell_h}{2}\|\mathbf{v}\|^2 \geq h(\mathbf{z}) + \frac{\ell_h}{2}\|\mathbf{z}\|^2 + \langle \nabla h(\mathbf{z}) + \ell_h \mathbf{z}, \mathbf{v} - \mathbf{z} \rangle, \ \forall \mathbf{z}, \mathbf{v}.
$$
This is equivalent to show
$$
h(\mathbf{v}) + \frac{\ell_h}{2}\|\mathbf{v} - \mathbf{z}\|^2 \geq h(\mathbf{z}) + \langle \nabla h(\mathbf{z}), \mathbf{v} - \mathbf{z} \rangle, \ \forall \mathbf{z}, \mathbf{v}.
$$

We drive with $\mathbf{x} = \boldsymbol{\psi}(\mathbf{z}), \mathbf{y} = \boldsymbol{\psi}(\mathbf{v})$ as follows,

$$
\begin{aligned}
\langle \nabla h(\mathbf{z}), \mathbf{v} - \mathbf{z} \rangle &= \langle \nabla \mathrm{J}_{\boldsymbol{\psi}}(\mathbf{z})^\top f(\mathbf{x}), \mathbf{v} - \mathbf{z} \rangle \\
&= \langle \nabla f(\mathbf{x}), \mathrm{J}_{\boldsymbol{\psi}}(\mathbf{z})(\mathbf{v} - \mathbf{z}) \rangle \\
&= \langle \nabla f(\mathbf{x}), -\boldsymbol{\psi}(\mathbf{v}) + \boldsymbol{\psi}(\mathbf{z}) + \mathrm{J}_{\boldsymbol{\psi}}(\mathbf{z})(\mathbf{v} - \mathbf{z}) \rangle + \langle \nabla f(\mathbf{x}), \boldsymbol{\psi}(\mathbf{v}) - \boldsymbol{\psi}(\mathbf{z}) \rangle \\
&\leq \|\nabla f(\mathbf{x})\| \cdot \|\boldsymbol{\psi}(\mathbf{v}) - \boldsymbol{\psi}(\mathbf{z}) - \mathrm{J}_{\boldsymbol{\psi}}(\mathbf{z})(\mathbf{v} - \mathbf{z})\| + \langle \nabla f(\mathbf{x}), \mathbf{y} - \mathbf{x} \rangle \\
&\leq L_{f,0} L_{\boldsymbol{\psi}} \|\mathbf{z} - \mathbf{v}\|^2 + f(\mathbf{y}) - f(\mathbf{x}) \\
&= L_{f,0} L_{\boldsymbol{\psi}} \|\mathbf{z} - \mathbf{v}\|^2 + h(\mathbf{v}) - h(\mathbf{z})
\end{aligned}
$$

where the first inequality is from triangular inequality and the second inequality is from Lemma C.2 and the convexity of $f$.

$\square$

### D.3 KKT CONDITIONS OF PROBLEM $\mathbf{P}$ AND $\mathbf{H}$

First, we recall some basic definitions. Consider a general optimization problem

$$
\begin{aligned}
\min_{\mathbf{x} \in \mathbb{R}^n} \ & f(\mathbf{x}), \\
\text{s.t.} \quad & g_i(\mathbf{x}) \leq 0, \forall i = 1, 2, \cdots, m; \\
& q_i(\mathbf{x}) \leq 0, \forall i = 1, 2, \cdots, p.
\end{aligned}
\tag{G}
$$

The Lagrangian function of problem (G) is defined as

$$
\mathcal{L}(\mathbf{x}, \boldsymbol{\lambda}, \boldsymbol{\nu}) = f(\mathbf{x}) + \sum_{i=1}^{m} \lambda_i g_i(\mathbf{x}) + \sum_{i=1}^{p} \nu_i q_i(\mathbf{x}).
$$

A triple $(\mathbf{x}, \boldsymbol{\lambda}, \boldsymbol{\nu})$ is said to satisfy the Karush–Kuhn–Tucker (KKT) condition of problem (G) if the following holds

$$
\begin{aligned}
\nabla f(\mathbf{x}) + \sum_{i=1}^{m} \lambda_i \nabla g_i(\mathbf{x}) + \sum_{j=1}^{p} \nu_j \nabla q_j(\mathbf{x}) &= \mathbf{0}, \\
q_j(\mathbf{x}) = 0, g_i(\mathbf{x}) \leq 0, &\quad \forall j \in [p], i \in [m]; \\
\boldsymbol{\lambda} \geq \mathbf{0}, \ \lambda_i g_i(\mathbf{x}) = 0, &\quad \forall i \in [m].
\end{aligned}
\tag{15}
$$

where $\boldsymbol{\lambda}$ (or $\boldsymbol{\nu}$) is the dual variable corresponding to inequality (resp. equality) constraints.

**Definition D.2** (KKT stationary point). A point $\mathbf{x}^*$ is said to be a KKT stationary point of (G) if there exists $\boldsymbol{\lambda}^* \in \mathbb{R}^m_{\geq 0}, \boldsymbol{\nu}^* \in \mathbb{R}^p$ such that $(\mathbf{x}^*, \boldsymbol{\lambda}^*, \boldsymbol{\nu}^*)$ satisfies KKT condition (15).

**Definition D.3** (Strict complementary slackness). It is said that the strict complementary slackness condition holds for problem (G), if

$$
\lambda_i^* > 0 \quad \text{for} \quad g_i(\mathbf{x}^*) = 0, \quad \forall i \in [m].
$$

To define the second-order KKT condition for the optimization problems, we recall that the critical cone in the following.

**Definition D.4** (Critical cone). Denote the feasible region of problem (G) as $\mathcal{G}$. Then the critical cone $\mathrm{C}_{\mathcal{G}}(\mathbf{x}^*)$ at $\mathbf{x}^*$ of problem (G) is defined as (Nocedal & Wright, 1999)

$$
\mathbf{w} \in \mathrm{C}_{\mathcal{G}}(\mathbf{x}^*) \Leftrightarrow
\begin{cases}
\nabla q_i(\mathbf{x}^*)^\top \mathbf{w} = 0, & \text{for all } i \in [p], \\
\nabla g_i(\mathbf{x}^*)^T \mathbf{w} = 0, & \text{for all } i \in \mathcal{A}(\mathbf{x}^*) \text{ with } \lambda_i^* > 0, \\
\nabla g_i(\mathbf{x}^*)^T \mathbf{w} \geq 0, & \text{for all } i \in \mathcal{A}(\mathbf{x}^*) \text{ with } \lambda_i^* = 0.
\end{cases}
$$

Here $\boldsymbol{\lambda}^*$ is the Lagrangian multiplier of inequality constraints $g_i$ and $\mathcal{A}(\mathbf{x}^*)$ is the index of active constraints.

From the definition, the critical cone of problem $\mathbf{P}$ can be written as

$$\mathbf{w} \in \mathrm{C}_{\mathcal{K}}(\mathbf{x}^*) \Leftrightarrow \begin{cases} \nabla g_i(\mathbf{x}^*)^T \mathbf{w} = 0, & \text{for all } i \in \mathcal{A}(\mathbf{x}^*) \text{ with } \lambda_i^* > 0, \\ \nabla g_i(\mathbf{x}^*)^T \mathbf{w} \geq 0, & \text{for all } i \in \mathcal{A}(\mathbf{x}^*) \text{ with } \lambda_i^* = 0. \end{cases}$$

Moreover, if *strict complementary slackness* holds, the critical cone is simplified as

$$\mathrm{C}_{\mathcal{K}}(\mathbf{x}^*) = \{\mathbf{w} \in \mathbb{R}^n \mid \nabla g_i(\mathbf{x}^*)^T \mathbf{d} = 0, \text{ for all } i \in \mathcal{A}(\mathbf{x}^*)\}.$$

Suppose *strict complementary slackness* holds for problem $\mathbf{P}$ and $\mathbf{H}$. Then, we can write KKT conditions for problem $\mathbf{P}$ and $\mathbf{H}$ in the following.

*First-order KKT conditions* on $\mathbf{x}^*$. The Lagrangian of $\mathbf{P}$ is

$$\mathcal{L}_{\mathrm{P}}(\mathbf{x}, \boldsymbol{\lambda}) = f(\mathbf{x}) + \sum_{i=1}^{m} \lambda_i g_i(\mathbf{x}).$$

The first-order KKT conditions of $\mathbf{P}$ are: there exists $\boldsymbol{\lambda}^*$ such that

$$\nabla f(\mathbf{x}^*) + \sum_{i=1}^{m} \lambda_i^* \nabla g_i(\mathbf{x}^*) = \mathbf{0}, \tag{16a}$$

$$g_i(\mathbf{x}^*) \leq 0, \quad i = 1, 2, \cdots, m \tag{16b}$$

$$\boldsymbol{\lambda}^* \geq \mathbf{0}, \ \lambda_i^* g_i(\mathbf{x}^*) = 0, \quad i = 1, 2, \cdots, m. \tag{16c}$$

*Second-order KKT conditions* on $\mathbf{x}^*$. It adds the following condition

$$\mathbf{w}^\top \nabla_{\mathbf{x}}^2 \mathcal{L}_{\mathrm{P}}(\mathbf{x}^*, \boldsymbol{\lambda}^*) \mathbf{w} \geq 0 \tag{17}$$

for any $\mathbf{w}$ satisfying $\mathbf{w}^\top \nabla g_i(\mathbf{x}^*) = 0$ with $i \in \mathcal{A}(\mathbf{x}^*)$.

*First-order KKT conditions* on $\mathbf{z}^*$. The Lagrangian of $\mathbf{H}$ is

$$\mathcal{L}_{\mathrm{H}}(\mathbf{z}, \nu) = h(\mathbf{z}) + \nu(\|\mathbf{z}\|^2 - 1).$$

The first-order KKT conditions of $\mathbf{H}$ are: there exists $\nu^*$ such that

$$\nabla h(\mathbf{z}^*) + 2\nu^* \mathbf{z}^* = \mathbf{0}, \tag{18a}$$

$$\|\mathbf{z}^*\|^2 \leq 1, \tag{18b}$$

$$\nu^* \geq 0, \ \nu^*(\|\mathbf{z}^*\|^2 - 1) = 0. \tag{18c}$$

*Second-order KKT condition* on $\mathbf{z}^*$. It will add the following condition.

$$\mathbf{d}^\top \nabla_{\mathbf{z}}^2 \mathcal{L}_{\mathrm{H}}(\mathbf{z}^*, \nu^*) \mathbf{d} \geq 0 \tag{19}$$

for any $\mathbf{d} \in \mathrm{C}_{\mathcal{B}}(\mathbf{z}^*)$. Here recall that

$$\mathrm{C}_{\mathcal{B}}(\mathbf{z}^*) = \begin{cases} \mathbb{R}^n, & \text{if } \mathbf{z}^* \in \mathrm{int}(\mathcal{B}), \\ \{\mathbf{d} : \mathbf{d}^\top \mathbf{z}^* = 0\}, & \text{if } \mathbf{z}^* \in \partial\mathcal{B}. \end{cases}$$

### D.4 Relationships of KKT Stationary Points between Problem $\mathbf{P}$ and $\mathbf{H}$

**Lemma D.5.** *Suppose strict complementary slackness holds for both problem $\mathbf{P}$ and $\mathbf{H}$. We have that $\mathbf{x}^*$ is a KKT stationary point of $\mathbf{P}$ if and only if $\mathbf{z}^*$ is also a KKT stationary point of $\mathbf{H}$ where $\mathbf{x}^* = \boldsymbol{\psi}(\mathbf{x}^*)$.*

*Proof.* 1) First, we assume that $\mathbf{x}^*$ is a KKT stationary point of $\mathbf{P}$. By assumption, there exists $\boldsymbol{\lambda}^*$ such that the KKT condition holds (16) holds. Then we have

$$\mathrm{J}_{\boldsymbol{\psi}}(\mathbf{z}^*)^\top \nabla f(\mathbf{x}^*) + \sum_{i=1}^{m} \lambda_i^* \mathrm{J}_{\boldsymbol{\psi}}(\mathbf{z}^*)^\top \nabla g_i(\mathbf{x}^*) = \mathbf{0},$$

$$g_i(\boldsymbol{\psi}(\mathbf{z}^*)) \leq 0, \quad i = 1, 2, \cdots, m$$

$$\boldsymbol{\lambda}^* \geq \mathbf{0}, \ \lambda_i^* g_i(\boldsymbol{\psi}(\mathbf{z}^*)) = 0, \quad i = 1, 2, \cdots, m.$$

This is equivalent to

$$\nabla h(\mathbf{z}^*) + \sum_{i=1}^{m} \lambda_i^* \nabla G_i(\mathbf{z}^*) = \mathbf{0}, \tag{20a}$$

$$G_i(\mathbf{z}^*) \leq 0, \quad i = 1, 2, \cdots, m \tag{20b}$$

$$\boldsymbol{\lambda}^* \geq \mathbf{0}, \ \lambda_i^* G_i(\mathbf{z}^*) = 0, \quad i = 1, 2, \cdots, m. \tag{20c}$$

Let $\nu^* = \sum_{i=1}^{m} \lambda_i^*$. According to the eq. (12,13), eq. (20a) is actually

$$\nabla h(\mathbf{z}^*) + 2\nu^* \mathbf{z}^* = \mathbf{0}.$$

By assumption, eq. (20b) is equivalent to

$$\|\mathbf{z}^*\|^2 \leq 1.$$

Note that if $G_i(\mathbf{z}^*) < 0$ for all $i$, then $\boldsymbol{\lambda}^* = 0$ and thus $\nu^* = 0$. In this case, $\nu^*(\|\mathbf{z}^*\|^2 - 1) = 0$. If $\mathbf{z}^*$ makes at least one $G_i(\mathbf{z}^*) = 0$, then we have $\|\mathbf{z}^*\|^2 = 1$. In this case, we also have $\nu^*(\|\mathbf{z}^*\|^2 - 1) = 0$. Hence, eq. (20c) implies

$$\nu^* \geq 0, \nu^*(\|\mathbf{z}^*\|^2 - 1) = 0.$$

In conclusion, there exists $\mathbf{z}^*, \nu*$ such the KKT condition holds.

2) Now, we assume $\mathbf{z}^*, \nu^*$ satisfy KKT condition for problem $\mathbf{H}$, i.e.,

$$\nabla h(\mathbf{z}^*) + 2\nu^* \mathbf{z}^* = \mathbf{0},$$

$$\|\mathbf{z}^*\|^2 \leq 1,$$

$$\nu^* \geq 0, \ \nu^*(\|\mathbf{z}^*\|^2 - 1) = 0.$$

If $\mathbf{z}^* \in \text{int}(\mathcal{B})$, then $G_i(\mathbf{z}^*) < 0$ for all $i$ and $\nu^* = 0$. In this case, there exists $\boldsymbol{\lambda}^* = 0$ such that the KKT condition with eq. (16) of problem $\mathbf{P}$ holds at $\mathbf{x}^* = \boldsymbol{\psi}(\mathbf{z}^*), \boldsymbol{\lambda}^* = \mathbf{0}$ .

If $\mathbf{z}^* \in \partial\mathcal{B}$, then there exists at least one $i \in \{1, 2, \cdots, m\}$ such that $G_i(\mathbf{z}^*) = 0$ and $\nu^* > 0$ from strict complementary slackness. Denote $\mathcal{A} = \{i : G_i(\mathbf{z}^*) = 0\}$. Note we define $\lambda_i^* = 0$ if $i \notin \mathcal{A}$ and $\lambda_i^* = \nu^*/|\mathcal{A}|$. Then we have $\mathbf{z}^*, \boldsymbol{\lambda}^*$ such that eq. 20 holds which implies $\mathbf{x}^* = \boldsymbol{\psi}(\mathbf{z}^*), \boldsymbol{\lambda}^*$ make the KKT condition of problem $\mathbf{P}$ hold.

$\square$

**Lemma D.6.** *Suppose strict complementary slackness condition holds for both problem $\mathbf{P}$ and $\mathbf{H}$. Then $\mathbf{x}^*$ is a second-order KKT stationary point of $\mathbf{P}$ if and only if $\mathbf{z}^* = \boldsymbol{\psi}^{-1}(\mathbf{x}^*)$ is also a second-order KKT stationary point of $\mathbf{H}$.*

*Proof.* From Lemma D.5, there exists $\boldsymbol{\lambda}^*$ and $\nu^*$ such that $(\mathbf{x}^*, \boldsymbol{\lambda}^*)$ holds for first-order KKT condition of $\mathbf{P}$ if and only if $(\mathbf{z}^*, \nu^*)$ holds for first-order KKT condition of $\mathbf{H}$. Hence, it suffices to show the equivalence of condition 19 and 17.

1) Let's first suppose $\mathbf{x}^*$ is a second-order KKT stationary point, i.e., eq. (17) holds.

Note

$$\nabla_{\mathbf{z}}^2 \mathcal{L}_{\mathrm{H}}(\mathbf{z}^*, \nu^*) = \nabla^2 h(\mathbf{z}^*) + 2\nu^* \mathbf{I}_n,$$

where $\mathbf{I}_n$ is identity matrix of size $n \times n$. We just need to show $\mathbf{d}^\top \nabla \mathcal{L}_{\mathrm{H}}(\mathbf{z}^*, \nu^*)\mathbf{d} \geq 0$ for any $\mathbf{d} \in \mathrm{C}_{\mathcal{B}}(\mathbf{z}^*)$. Recall that

$$\nabla^2 h(\mathbf{z}^*) = \mathrm{J}_{\boldsymbol{\psi}}(\mathbf{z}^*)^\top \nabla^2 f(\boldsymbol{\psi}(\mathbf{z}^*))\mathrm{J}_{\boldsymbol{\psi}}(\mathbf{z}^*) + \sum_{i=1}^{n} \frac{\partial f}{\partial \mathbf{x}_i}(\boldsymbol{\psi}(\mathbf{z}^*))\nabla^2 \boldsymbol{\psi}_i(\mathbf{z}^*),$$

and

$$\nabla^2 G_i(\mathbf{z}^*) = \mathrm{J}_{\boldsymbol{\psi}}(\mathbf{z}^*)^\top \nabla^2 g_i(\boldsymbol{\psi}(\mathbf{z}^*))\mathrm{J}_{\boldsymbol{\psi}}(\mathbf{z}^*) + \sum_{k=1}^{n} \frac{\partial g_i}{\partial \mathbf{x}_k}(\boldsymbol{\psi}(\mathbf{z}^*))\nabla^2 \psi_k(\mathbf{z}^*), \quad k = 1, 2, \cdots, m.$$

From eq. (12), note that

$$\nabla^2 G_k(\mathbf{z}^*) = 2\mathbf{I}_n, \forall k \in \mathcal{A}(\mathbf{x}^*) \cap \{k : G_k(\mathbf{z}^*) = 0\}.$$

From Lemma D.5, $\nu^* = \sum_i \lambda_i^*$. Then we have

$$\nabla^2 \mathcal{L}_{\mathrm{H}}(\mathbf{z}^*, \nu^*) = \nabla^2 h(\mathbf{z}^*) + \sum_{i=1}^m \lambda_i^* \nabla^2 G_i(\mathbf{z}^*) \tag{21a}$$

$$= \mathrm{J}_{\boldsymbol{\psi}}(\mathbf{z}^*)^\top \nabla^2 f(\boldsymbol{\psi}(\mathbf{z}^*)) \mathrm{J}_{\boldsymbol{\psi}}(\mathbf{z}^*) + \sum_{i=1}^m \mathrm{J}_{\boldsymbol{\psi}}(\mathbf{z}^*)^\top \lambda_i^* \nabla^2 g_i(\boldsymbol{\psi}(\mathbf{z}^*)) \mathrm{J}_{\boldsymbol{\psi}}(\mathbf{z}^*) \tag{21b}$$

$$+ \sum_{k=1}^n \frac{\partial f}{\partial \mathbf{x}_k}(\boldsymbol{\psi}(\mathbf{z}^*)) \nabla^2 \psi_k(\mathbf{z}^*) + \sum_{k=1}^n \sum_{i=1}^m \lambda_i^* \frac{\partial g_i}{\partial \mathbf{x}_k}(\boldsymbol{\psi}(\mathbf{z}^*)) \nabla^2 \psi_k(\mathbf{z}^*). \tag{21c}$$

From first-order KKT stationarity of $\mathbf{P}$, i.e.,

$$\nabla f(\mathbf{x}^*) + \sum_{i=1}^m \lambda_i^* \nabla g_i(\mathbf{x}^*) = \mathbf{0},$$

We have

$$\frac{\partial f}{\partial \mathbf{x}_k}(\mathbf{x}^*) + \sum_{i=1}^m \lambda_i^* \frac{\partial g_i}{\partial \mathbf{x}_k}(\mathbf{x}^*) = 0.$$

Hence for any $\mathbf{d} \in \mathrm{C}_{\mathcal{B}}(\mathbf{z}^*)$, we have the second term (21c) is equal to 0.

Now we note it's trivial that $\mathrm{C}_{\mathcal{K}}(\mathbf{x})^* = \mathrm{C}_{\mathcal{B}}(\mathbf{z}^*) = \mathbb{R}^n$ if $\mathbf{z}^* \in \mathrm{int}(\mathcal{K})$ where $\mathbf{x} = \boldsymbol{\psi}(\mathbf{z}^*)$. Hence in this case if $\mathbf{d} \in \mathrm{C}_{\mathcal{B}}(\mathbf{z}^*)$, we will have $\mathrm{J}_{\boldsymbol{\psi}}(\mathbf{z}^*)\mathbf{d} \in \mathrm{C}_{\mathcal{K}}(\mathbf{x}^*)$

If $\mathbf{x}^* \in \partial\mathcal{K}$. Then $\mathcal{A}(\mathbf{x}^*) \neq \emptyset$. For $\mathbf{d} \in \mathrm{C}_{\mathcal{B}}(\mathbf{z}^*)$, i.e., $\mathbf{d}^\top \mathbf{z}^* = 0$, we have

$$(\mathrm{J}_{\boldsymbol{\psi}}(\mathbf{z}^*)\mathbf{d})^\top \nabla g_i(\mathbf{x}^*) = \mathbf{d}^\top \mathrm{J}_{\boldsymbol{\psi}}(\mathbf{z}^*)^\top \nabla g_i(\mathbf{x}^*) = \mathbf{d}^\top G_i(\mathbf{z}^*) = 2\mathbf{d}^\top \mathbf{z}^* = 0, \quad \text{for } i \in \mathcal{A}(\mathbf{x}^*),$$

or $\mathrm{J}_{\boldsymbol{\psi}}(\mathbf{z}^*)\mathbf{d} \in \mathrm{C}_{\mathcal{K}}(\mathbf{x}^*)$.

So for $\mathbf{d}^\top \in \mathrm{C}_{\mathcal{B}}(\mathbf{z}^*)$, we have the following holds about the first term of $\nabla^2 \mathcal{L}_{\mathrm{H}}(\mathbf{z}^*, \nu^*)$.

$$(\mathrm{J}_{\boldsymbol{\psi}}(\mathbf{z}^*)\mathbf{d})^\top \nabla^2 f(\boldsymbol{\psi}(\mathbf{z}^*)) \mathrm{J}_{\boldsymbol{\psi}}(\mathbf{z}^*)\mathbf{d} + (\mathrm{J}_{\boldsymbol{\psi}}(\mathbf{z}^*)\mathbf{d})^\top \left( \sum_{i=1}^m \lambda_i^* \nabla^2 g_i(\boldsymbol{\psi}(\mathbf{z}^*)) \right) \mathrm{J}_{\boldsymbol{\psi}}(\mathbf{z}^*)\mathbf{d} \geq 0$$

where the last '$\geq$' is from the assumption that $\mathbf{x}^*$ is the second-order KKT stationary point of $\mathbf{P}$. Hence, we have $\mathbf{d}^\top \nabla^2 \mathcal{L}_{\mathrm{H}}(\mathbf{z}^*, \nu^*)\mathbf{d} \geq 0$ for any $\mathbf{d}^T \in \mathrm{C}_{\mathcal{B}}(\mathbf{z}^*)$, i.e., $\mathbf{z}^* = \boldsymbol{\psi}^{-1}(\mathbf{x}^*)$ is also a second-order KKT stationary point.

2) Let's suppose $\mathbf{z}^*$ is a second-order KKT stationary point and show that $\mathbf{x}^*$ is a second-order KKT stationary point.

If $\mathbf{z}^* \in \mathrm{int}(\mathcal{B})$, the proof is trivial because $\nu^* = 0$ according to the similar analysis. So we assume $\mathbf{z}^* \in \partial\mathcal{B}$. Define $\mathcal{A}(\mathbf{z}^*) = \{i : G_i(\mathbf{z}^*) = 0\}$, and $\lambda_i^* = 0$ for $i \notin \mathcal{A}(\mathbf{z}^*)$, $\lambda_i^* = \nu^*/|\mathcal{A}(\mathbf{z}^*)|$ for $i \in \mathcal{A}(\mathbf{z}^*)$.

Note for any $\mathbf{w} \in \mathrm{C}_{\mathcal{K}}(\mathbf{x}^*)$, we have

$$\mathbf{0} = \mathbf{w}^\top \nabla g_i(\mathbf{x}^*) = \mathbf{w}^\top \mathrm{J}_{\boldsymbol{\psi}}^{-1}(\mathbf{z}^*) \nabla G_i(\mathbf{z}^*) = (\mathrm{J}_{\boldsymbol{\psi}}^{-1}(\mathbf{z}^*)\mathbf{w})^\top \mathbf{z}^*, \quad \text{for } i \in \mathcal{A}(\mathbf{x}^*) = \mathcal{A}(\mathbf{z}^*).$$

Hence $J_{\psi}^{-1}(\mathbf{z}^*)\mathbf{w} \in C_{\mathcal{B}}(\mathbf{z}^*)$. Then for any $\mathbf{w} \in C_{\mathcal{K}}(\mathbf{x}^*)$,

$$\mathbf{w}^\top \nabla_{\mathbf{x}}^2 \mathcal{L}_{\mathrm{P}}(\mathbf{x}^*, \boldsymbol{\lambda}^*)\mathbf{w}$$

$$= \mathbf{w}^\top \nabla^2 f(\mathbf{x}^*)\mathbf{w} + \mathbf{w}^\top \sum_{i=1}^m \lambda_i^* \nabla^2 g_i(\mathbf{x}^*)\mathbf{w}$$

$$= (J_{\psi}^{-1}(\mathbf{z}^*)\mathbf{w})^\top J_{\psi}(\mathbf{z}^*)\nabla^2 f(\mathbf{x}^*)J_{\psi}(\mathbf{z}^*)J_{\psi}^{-1}(\mathbf{z}^*)\mathbf{w}$$

$$+ (J_{\psi}^{-1}(\mathbf{z}^*)\mathbf{w})^\top J_{\psi}(\mathbf{z}^*)(\sum_{i=1}^m \lambda_i^* \nabla^2 g_i(\mathbf{x}^*))J_{\psi}(\mathbf{z}^*)J_{\psi}^{-1}(\mathbf{z}^*)\mathbf{w}$$

$$+ (J_{\psi}^{-1}(\mathbf{z}^*)\mathbf{w})^\top [\sum_{k=1}^n \frac{\partial f}{\partial \mathbf{x}_k}(\boldsymbol{\psi}(\mathbf{z}^*))\nabla^2 \psi_k(\mathbf{z}^*) + \sum_{k=1}^n \sum_{i=1}^m \lambda_i^* \frac{\partial g_i}{\partial \mathbf{x}_k}(\boldsymbol{\psi}(\mathbf{z}^*))\nabla^2 \psi_k(\mathbf{z}^*)]J_{\psi}^{-1}(\mathbf{z}^*)\mathbf{w}$$

$$= (J_{\psi}^{-1}(\mathbf{z}^*)\mathbf{w})^\top \mathcal{L}_{\mathrm{H}}(\mathbf{z}^*, \nu^*)J_{\psi}^{-1}(\mathbf{z}^*)\mathbf{w} \geq 0$$

where the sum of last term of the second '=' is exactly $0$ and the last '$\geq$' is from the assumption that $\mathbf{z}^*$ is a second-order KKT stationary point.

$\square$

**Definition D.7** (Non-degenerate KKT stationary point). A second-order KKT point $\mathbf{x}^*$ of $\mathbf{P}$ is said to be non-degenerate if there exists $\boldsymbol{\lambda}^*$ such that

$$\mathbf{d}^\top \nabla^2 \mathcal{L}(\mathbf{x}^*, \boldsymbol{\lambda}^*)\mathbf{d} > 0$$

for all $0 \neq \mathbf{d} \in C_{\mathcal{K}}(\mathbf{x}^*)$. Here the Lagrangian function is

$$\mathcal{L}(\mathbf{x}, \boldsymbol{\lambda}) = f(\mathbf{x}) + \sum_{i=1}^m \lambda_i g_i(\mathbf{x}).$$

**Lemma D.8.** *Suppose strict complementary slackness holds for problem $\mathbf{P}$ and $\mathbf{H}$. Then $\mathbf{x}^\star$ is a non-degenerate KKT point of optimization $\mathbf{P}$ if and only if $\mathbf{z}^*$ satisfying $\mathbf{x}^\star = \boldsymbol{\psi}(\mathbf{z}^*)$ is also a non-degenerate KKT point of problem $\mathbf{H}$.*

*Proof.* 1) Suppose $\mathbf{x}^*$ is a non-degenerate KKT stationary point. Note that for $\mathbf{d} \in C_{\mathcal{B}}(\mathbf{z}^*)$, we have $J_{\psi}(\mathbf{z}^*)\mathbf{d} \in C_{\mathcal{K}}(\mathbf{x}^*)$ from the proof of Lemma D.6. Moreover, from $J_{\psi}(\mathbf{z}^*) \neq 0$ we have $J_{\psi}(\mathbf{z}^*)\mathbf{d} \neq 0$ if and only if $\mathbf{d} \neq 0$. Then the conclusion is trivial from eq. (21) in the proof of Lemma D.6.

2) Now, we suppose $\mathbf{z}^*$ is a non-degenerate KKT stationary point. It follows from the proof of Lemma D.6 that for any $\mathbf{w} \in C_{\mathcal{K}}(\mathbf{x}^*)$, we have $J_{\psi}^{-1}(\mathbf{z}^*)\mathbf{w} \in C_{\mathcal{B}}(\mathbf{z}^*)$. Hence, the conclusion is also trivial from the proof of item (2) of Lemma D.6.

$\square$

# E CONVERGENCE ANALYSIS: OPTIMIZATION OVER NON-CONVEX BH SET

In this section, we then provide the proof of Theorem 1. Before moving on, we first introduce some definitions and notations below.

**Definition E.1** (Approximate stationary point). A point $\mathbf{x}^*$ is called $\epsilon$-stationary point for problem $\min_{\mathbf{x} \in \mathcal{K}} f(\mathbf{x})$ with convex set $\mathcal{K}$, if the gradient norm mapping

$$\mathrm{Gr}_f^{\mathcal{K}}(\mathbf{x}; \alpha) := \frac{1}{\alpha}[\mathbf{x} - \Pi_{\mathcal{K}}(\mathbf{x} - \alpha \nabla f(\mathbf{x}))]$$

satisfies $\|\mathrm{Gr}_f^{\mathcal{K}}(\mathbf{x}; \alpha)\| \leq \epsilon$ for proper $\alpha > 0$.

**Definition E.2** (Normal cone). The normal cone $\mathrm{N}_S(\mathbf{x})$ of a closed and convex set $\mathcal{K}$ at $\mathbf{x} \in \mathcal{K}$ is defined as

$$\mathrm{N}_{\mathcal{K}}(\mathbf{x}) = \{\mathbf{y} : \langle \mathbf{y}, \mathbf{z} - \mathbf{x} \rangle \leq 0 \text{ for any } \mathbf{z} \in \mathcal{K}\}.$$

**Notations.**

- Recall that $\psi$ is the exact homeomorphic mapping and $\Phi$ is the learned, approximate homeomorphic mapping. Thus, we denote $\mathcal{B} := \psi^{-1}(\mathcal{K})$ as a unit ball and $\tilde{\mathcal{B}} := \Phi^{-1}(\mathcal{K})$ as an approximate unit ball. Moreover, as Assumption 2 holds, we have

$$\|\mathrm{BP}_{\tilde{\mathcal{B}}}(\mathbf{z}) - \Pi_{\mathcal{B}}(\mathbf{z})\| \leq \epsilon_{\mathrm{inn}}.$$

- We denote $h := f \circ \psi$ and $H = f \circ \Phi$.

- We denote the bi-Lipschitz continuous constant of $\Phi$ as $l_\Phi$ and $u_\Phi$, i.e.,

$$l_\Phi \|\mathbf{u} - \mathbf{v}\| \leq \|\Phi(\mathbf{u}) - \Phi(\mathbf{v})\| \leq u_\Phi \|\mathbf{u} - \mathbf{v}\|. \tag{22}$$

Recall that the bi-Lipschitz continuous property of an INN composed of affine coupling layers is satisfied by its design (Prop. B.1). Under this condition, we have

$$\|\mathrm{J}_\Phi(\mathbf{z})\| \leq u_\Phi, \|\mathrm{J}_{\Phi^{-1}}(\mathbf{z})\| \leq \frac{1}{l_\Phi}.$$

### E.1 Proof of Theorem 1

We list some help lemmas first in the following.

**Lemma E.3.** *Suppose an error $\epsilon > 0$ is sufficiently small. Consider $\min_{\mathbf{z} \in \mathcal{B}} h(\mathbf{z})$. If $\| \mathrm{Gr}_h^{\mathcal{B}}(\mathbf{z}'; \alpha)\| \leq \epsilon$ for some $\mathbf{z}' \in \mathcal{B}$, then $\mathbf{z}'$ is an $\mathcal{O}(\epsilon)$-KKT stationary point of problem $\min_{\mathbf{z} \in \mathcal{B}} h(\mathbf{z})$. Specifically, there exists $\nu^*$ such that*

$$\|\nabla h(\mathbf{z}') + 2\nu^* \mathbf{z}'\| \leq \alpha(1 + \beta)\epsilon,$$

$$\|\mathbf{z}'\| - 1 \leq 0,$$

$$\nu^* \geq 0, |\nu^*(\|\mathbf{z}'\|^2 - 1)| \leq \beta\epsilon,$$

*where $\beta$ is a constant depending on $\mathbf{z}'$.*

*Proof.* Suppose $\mathbf{z}^+ = \Pi_{\mathcal{B}}(\mathbf{z}' - \alpha \nabla h(\mathbf{z}'))$ and $\mathrm{Gr}(\mathbf{z}') = \mathrm{Gr}_h^{\mathcal{K}}(\mathbf{z}'; \alpha)$ for conciseness of notation. Then $\mathrm{Gr}(\mathbf{z}') = \frac{1}{\alpha}(\mathbf{z}' - \mathbf{z}^+)$.

From the optimality of orthogonal projection (Prop. C.1), we have

$$\langle \mathbf{z}' - \alpha \nabla h(\mathbf{z}') - \mathbf{z}^+, \mathbf{z} - \mathbf{z}^+ \rangle \leq 0$$

for any $\mathbf{z} \in \mathcal{B}$. Let $\zeta = \mathbf{z}' - \mathbf{z}^+ - \alpha \nabla h(\mathbf{z}')$. We have $\zeta \in \mathrm{N}_{\mathcal{B}}(\mathbf{z}^+)$ by [definition of the normal cone]. Moreover, the normal cone of a unit ball can be written as

$$\mathrm{N}_{\mathcal{B}}(\mathbf{z}^+) = \{\beta \mathbf{z}^+ : \beta > 0\} \text{ for } \mathbf{z}^+ \in \partial\mathcal{B}; \text{ and } \mathrm{N}_{\mathcal{B}}(\mathbf{z}^+) = \{\beta \mathbf{z}^+ : \beta = 0\} \text{ for } \mathbf{z}^+ \in \mathrm{int}(\mathcal{B}).$$

Hence we have $\zeta = \beta \mathbf{z}^+$ for some $\beta \geq 0$, i.e.,

$$\alpha \nabla h(\mathbf{z}') + \beta \mathbf{z}^+ = \mathbf{z}' - \mathbf{z}^+.$$

Equivalently,

$$\nabla h(\mathbf{z}') + \frac{1}{\alpha}[\beta \mathbf{z}' + \beta(\mathbf{z}^+ - \mathbf{z}')] = \frac{1}{\alpha}[\mathbf{z}' - \mathbf{z}^+].$$

Thus,

$$\|\nabla h(\mathbf{z}') + \frac{\beta}{\alpha} \mathbf{z}'\| \leq (1 + \beta)\|\mathrm{Gr}(\mathbf{z}')\| \leq (1 + \beta)\epsilon.$$

By defining $\nu^* = \frac{\beta}{2\alpha} \geq 0$, we have

$$\|\nabla h(\mathbf{z}') + 2\nu^* \mathbf{z}'\| \leq (1 + \beta)\|\mathrm{Gr}(\mathbf{z}')\| \leq (1 + \beta)\epsilon.$$

Next, note that $\mathbf{z}'$ is feasible, thereby $\|\mathbf{z}'\| - 1 \leq 0$.

Finally, we show

$$|\nu^*(\|\mathbf{z}'\|^2 - 1)| \leq \beta\epsilon.$$

If $\mathbf{z}^+ \in \mathrm{int}(\mathcal{B})$, we have $\beta = 0$ by the definition of $\beta$, i.e., $\nu^* = 0$. In this case, the proof is trivial. Hence, we assume $\mathbf{z}^+ \in \partial\mathcal{B}$. It follows that $\|\mathbf{z}^+\|^2 = 1$. Then we have

$$|\nu^*(\|\mathbf{z}'\|^2 - 1)| = |\nu^*(\|\mathbf{z}'\|^2 - \|\mathbf{z}^+\|^2)| \leq 2\nu^* \|\mathbf{z}' - \mathbf{z}^+\| \leq 2\nu^* \alpha \|\mathrm{Gr}(\mathbf{z}')\| \leq \beta\epsilon.$$

$\square$

**Lemma E.4.** *Consider the optimization problem* $\mathbf{H}_{\text{inn}}$: $\min_{\mathbf{z} \in \tilde{\mathcal{B}}} H(\mathbf{z})$. *Let* $\epsilon > 0$ *be a sufficiently small error and Assumption 2 hold. Suppose* $\{\mathbf{z}_k\}_{k \geq 0}$ *is a sequence generated by Hom-PGD+ with step-size* $\alpha \in (0, \frac{1}{L_H}]$. *Then* $\{\mathbf{z}_k\}_{0 \leq k \leq K}$ *contains an point* $\mathbf{z}'$ *with* $K = \mathcal{O}(L_H \epsilon^{-2})$ *such that*

$$\| \text{Gr}_H^{\mathcal{B}}(\mathbf{z}') \| \leq c\epsilon + \mathcal{O}(\sqrt{L_H \epsilon_{\text{inn}}})$$

*where* $c$ *is a constant independent of* $\epsilon$ *that can be small arbitrarily.*

*Proof.* We denote $\mathbf{z}_+ = \Pi_{\mathcal{B}}(\mathbf{z} - \alpha \nabla H(\mathbf{z}))$ and $\mathbf{z}^- = \text{BP}_{\mathcal{B}}(\mathbf{z} - \alpha \nabla H(\mathbf{z}))$. We know that $\|\mathbf{z}_+ - \mathbf{z}^-\| \leq \epsilon_{\text{inn}}$. According to the $L_H$ smoothness of $H$, we have

$$H(\mathbf{z}^-) \leq H(\mathbf{z}) + \langle \nabla H(\mathbf{z}), \mathbf{z}^- - \mathbf{z} \rangle + \frac{L_H}{2} \|\mathbf{z} - \mathbf{z}^-\|^2$$

$$= H(\mathbf{z}) + \langle \nabla H(\mathbf{z}), \mathbf{z}^- - \mathbf{z}_+ \rangle + \langle \nabla H(\mathbf{z}), \mathbf{z}_+ - \mathbf{z} \rangle + \frac{L_H}{2} \|\mathbf{z} - \mathbf{z}^-\|^2.$$

From Prop. C.1, we have

$$\langle \mathbf{z} - \alpha \nabla H(\mathbf{z}) - \mathbf{z}_+, \mathbf{z} - \mathbf{z}_+ \rangle \leq 0,$$

i.e.,

$$\langle \nabla H(\mathbf{z}), \mathbf{z} - \mathbf{z}_+ \rangle \leq -\frac{1}{\alpha} \|\mathbf{z}_+ - \mathbf{z}\|^2.$$

Hence, we have

$$H(\mathbf{z}^-) \leq H(\mathbf{z}) + \langle \nabla H(\mathbf{z}), \mathbf{z}^- - \mathbf{z}_+ \rangle + \langle \nabla H(\mathbf{z}), \mathbf{z}_+ - \mathbf{z} \rangle + \frac{L_H}{2} \|\mathbf{z} - \mathbf{z}^-\|^2$$

$$\leq H(\mathbf{z}) + (\frac{L_H}{2} - \frac{1}{\alpha}) \|\mathbf{z}_+ - \mathbf{z}\|^2 + \frac{L_H}{2} \|\mathbf{z}_+ - \mathbf{z}^-\|^2 + \|\nabla H(\mathbf{z})\| \cdot \|\mathbf{z}^- - \mathbf{z}_+\|.$$

It follows that

$$H(\mathbf{z}_k) - H(\mathbf{z}_{k+1}) + \frac{L_H}{2} \epsilon_{\text{inn}}^2 + L_{H,0} \epsilon_{\text{inn}} \geq \alpha(1 - \frac{\alpha L_H}{2}) \| \text{Gr}(\mathbf{z}_k) \|^2 \tag{23}$$

where we denote

$$\text{Gr}(\mathbf{z}) := \text{Gr}_H^{\mathcal{B}}(\mathbf{z}) = \frac{1}{\alpha}[\mathbf{z} - \Pi_{\mathcal{B}}(\mathbf{z} - \alpha \nabla H(\mathbf{z}))].$$

Let $M = \alpha(1 - \frac{\alpha L_H}{2})$. We sum up Eq. (23) from $k = 0$ to $k = K$, and then we have

$$H(\mathbf{z}_0) - H^* \geq H(\mathbf{z}_0) - H(\mathbf{z}_{K+1}) + (K+1)(\frac{L_H}{2} \epsilon_{\text{inn}}^2 + L_{H,0} \epsilon_{\text{inn}})$$

$$\geq M \sum_{k=1}^{K} \| \text{Gr}(\mathbf{z}_k) \|^2 \geq (K+1) \| \text{Gr}(\mathbf{z}') \|^2$$

where $\mathbf{z}' = \arg \min_{k=0,1,\cdots,K} \| \text{Gr}(\mathbf{z}_k) \|$. It follows that

$$\| \text{Gr}(\mathbf{z}') \| \leq \sqrt{\frac{H(\mathbf{z}_0) - H^*}{M(K+1)} + \frac{L_H}{2} \epsilon_{\text{inn}}^2 + L_{H,0} \epsilon_{\text{inn}}} = \mathcal{O}(\frac{1}{\sqrt{K}}) + \mathcal{O}(\sqrt{L_H \epsilon_{\text{inn}}}).$$

With $K = \mathcal{O}(L_H \epsilon^{-2})$, we get the conclusion. $\qquad\square$

**Lemma E.5.** *If* $\mathbf{z}'$ *is a **feasible** $\epsilon$-approximate KKT point of problem* $\min_{\mathbf{z} \in \mathcal{B}} H(\mathbf{z}) = f \circ \Phi(\mathbf{z})$ *over a unit ball,i.e.,* $\mathbf{z} \in \mathcal{B}$, *then* $\mathbf{x}' = \Phi(\mathbf{z}')$ *is an* $(\epsilon/ \min\{l_\Phi, 1\} + \mathcal{O}(\epsilon_{\text{inn}}))$-*approximate KKT point of problem* $\mathbf{P}$.

*Proof.* Note that

$$\mathcal{B} := \{\|\mathbf{z}\|^2 - 1 \leq 0\} = \{G_i(\mathbf{z}) := g_i(\psi(\mathbf{z})) \leq 0, i = 1, 2, \cdots, m\}$$

and

$$\tilde{\mathcal{B}} = \Phi^{-1}(\mathcal{K}) = \{Q_i(\mathbf{z}) := g_i(\Phi(\mathbf{z})) \leq 0, i = 1, 2, \cdots, m\}.$$

We derive

$$
\begin{aligned}
\|\nabla G_i(\mathbf{z}) - \nabla Q_i(\mathbf{z})\| &= \|\mathrm{J}_{\boldsymbol{\psi}}(\mathbf{z})\nabla g_i(\boldsymbol{\psi}(\mathbf{z})) - \mathrm{J}_{\Phi}(\mathbf{z})\nabla g_i(\Phi(\mathbf{z}))\| \\
&\leq \|\mathrm{J}_{\boldsymbol{\psi}}(\mathbf{z})\nabla g_i(\boldsymbol{\psi}(\mathbf{z})) - \mathrm{J}_{\boldsymbol{\psi}}(\mathbf{z})\nabla g_i(\Phi(\mathbf{z}))\| + \\
&\quad \|\mathrm{J}_{\boldsymbol{\psi}}(\mathbf{z})\nabla g_i(\Phi(\mathbf{z})) - \mathrm{J}_{\Phi}(\mathbf{z})\nabla g_i(\Phi(\mathbf{z}))\| \\
&\leq L_{\boldsymbol{\psi},0} L_{g_i} \epsilon_{\mathrm{inn}} + L_{g_i,0}\epsilon_{\mathrm{inn}}.
\end{aligned}
$$

By assumption, there exists $\nu' \geq 0$ such that

$$
\begin{aligned}
\|\nabla h(\mathbf{z}') + 2\nu'\mathbf{z}'\| &\leq \epsilon, \\
\|\mathbf{z}'\|^2 - 1 &\leq 0, \\
|\nu'(\|\mathbf{z}'\|^2 - 1)| &\leq \epsilon.
\end{aligned}
$$

First, we show it is a fact that there exists $\boldsymbol{\lambda}'$ such that

$$
\begin{aligned}
\|\nabla h(\mathbf{z}') + \sum_{i=1}^m \lambda_i' \nabla G_i(\mathbf{z}')\| &\leq \epsilon, \\
G_i(\mathbf{z}') &\leq 0, \quad i = 1, 2, \cdots, m \\
\boldsymbol{\lambda}' \geq \mathbf{0}, \ |\lambda_i' G_i(\mathbf{z}')| &\leq \epsilon/|\mathcal{A}|, \quad i = 1, 2, \cdots, m.
\end{aligned}
$$

where we define $\mathcal{A} := \{i \mid G_i(\mathbf{z}) = \|\mathbf{z}\|^2 - 1 \text{ at } \mathbf{z}'\}$ and denote $\lambda_i' := 0$ for $i \notin \mathcal{A}$ and $\lambda_i' := \nu'/|\mathcal{A}|$ for $i \notin \mathcal{A}$. Moreover, the second inequality is from the feasibility of $\mathbf{z}'$. Now, it is easy to check that the above approximate KKT condition holds.

Note that $G_i(\mathbf{z}') \leq 0$ implies $[G_i(\mathbf{z}^+)]_+ = 0$. Next, we derive the following.

$$
\begin{aligned}
\left\|\nabla h(\mathbf{z}') + \sum_{i=1}^m \lambda_i' \nabla Q_i(\mathbf{z}')\right\| &\leq \left\|\nabla h(\mathbf{z}') + \sum_{i=1}^m \lambda_i' \nabla G_i(\mathbf{z}')\right\| + \left\|\sum_{i=1}^m \lambda_i' \nabla Q_i(\mathbf{z}') - \sum_{i=1}^m \lambda_i' \nabla G_i(\mathbf{z}')\right\| \\
&\leq \epsilon + \mathcal{O}(\epsilon_{\mathrm{inn}}), \\
[Q_i(\mathbf{z}')]_+ &\leq [G_i(\mathbf{z}')]_+ + [Q_i(\mathbf{z}') - G_i(\mathbf{z}')]_+ \leq L_{g_i,0}\epsilon_{\mathrm{inn}}, \\
|\lambda_i' Q_i(\mathbf{z}')| &\leq |\lambda_i' G_i(\mathbf{z}')| + |\lambda_i'(Q_i(\mathbf{z}') - G_i(\mathbf{z}'))| \leq \epsilon + \lambda_i' L_{g_i,0}\epsilon_{\mathrm{inn}}.
\end{aligned}
$$

Moreover, we have

$$
\left\|\nabla h(\mathbf{z}') + \sum_{i=1}^m \lambda_i' \nabla Q_i(\mathbf{z}')\right\| \leq \epsilon + \mathcal{O}(\epsilon_{\mathrm{inn}}),
$$

$$
\left\|[\mathbf{Q}(\mathbf{z}')]_+\right\| \leq \sum_{i=1}^m [Q_i(\mathbf{z}')]_+ \leq \sum_{i=1}^m \left([G_i(\mathbf{z}')]_+ + [Q_i(\mathbf{z}') - G_i(\mathbf{z}')]_+\right) \leq m L_{\mathbf{g},0}\epsilon_{\mathrm{inn}},
$$

$$
\begin{aligned}
\sum_{i=1}^m |\lambda_i' Q_i(\mathbf{z}')| &\leq \sum_{i=1}^m \left(|\lambda_i' G_i(\mathbf{z}')| + |\lambda_i'(Q_i(\mathbf{z}') - G_i(\mathbf{z}'))|\right) \\
&= \sum_{i \in \mathcal{A}} |\lambda_i' G_i(\mathbf{z}')| + \sum_{i \notin \mathcal{A}} |\lambda_i' G_i(\mathbf{z}')| + \sum_{i=1}^m |\lambda_i'(Q_i(\mathbf{z}') - G_i(\mathbf{z}'))| \\
&\leq \epsilon + \lambda_{\max}'|\mathcal{A}| L_{\mathbf{g},0}\epsilon_{\mathrm{inn}}
\end{aligned}
$$

where $L_{\mathbf{g},0} = \max_{i=1,2,\ldots,m}\{L_{g_i,0}\}$ and $\lambda_{\max}' = \max_{i=1,2,\cdots,m}\{\lambda_i'\}$. Here in the second line we use the inequality $[a + b]_+ \leq [a]_+ + [b]_+$ for $a, b \in \mathbb{R}$.

That is $\mathbf{z}'$ is an $\epsilon + \mathcal{O}(\epsilon_{\mathrm{inn}})$-KKT points of problem $\mathbf{H}$ with homeomorphic mapping $\Phi$.

Next, we derive

$$\left\| \nabla f\left(\mathbf{x}'\right) + \sum_{i=1}^{m} \lambda_i' \nabla g_i\left(\mathbf{x}'\right) \right\| \leq \left\| J_\Phi\left(\mathbf{z}'\right)^{-\top} \right\| \cdot \left\| J_\Phi\left(\mathbf{z}'\right)^\top \nabla f\left(\mathbf{x}'\right) + \sum_{i=1}^{m} \lambda_i' J_\Phi\left(\mathbf{z}'\right)^\top \nabla g_i\left(\mathbf{x}'\right) \right\|,$$

$$\leq \frac{\epsilon}{l_\Phi} + \mathcal{O}\left(\epsilon_{\text{inn}}\right),$$

$$\left[\mathbf{g}\left(\mathbf{x}'\right)\right]_+ = \left[\mathbf{Q}\left(\mathbf{z}'\right)\right]_+ \leq \mathcal{O}\left(\epsilon_{\text{inn}}\right),$$

$$\sum_{i=1}^{m} |\lambda_i' g_i\left(\mathbf{x}'\right)| = \sum_{i=1}^{m} |\lambda_i' Q_i\left(\mathbf{z}_i\right)| \leq \epsilon + \mathcal{O}\left(\epsilon_{\text{inn}}\right).$$

It follows that $\mathbf{x}' = \Phi(\mathbf{z}')$ is an $(\epsilon/\min\{1, l_\Phi\} + \mathcal{O}(\epsilon_{\text{inn}}))$-approximate KKT point. $\qquad \square$

*Proof of Theorem 1.* This is the direct corollary of the above lemmas. From Lemma E.4, we have that Hom-PGD+ can find an approximate stationary point $\mathbf{z}'$ such that

$$\|\text{Gr}_H^{\mathcal{B}}(\mathbf{z}')\| \leq c\epsilon + \mathcal{O}(\sqrt{L_H \epsilon_{\text{inn}}})$$

in $\mathcal{O}(L_H \epsilon^2)$ iterations.

Then, it follows from Lemma E.3 that $\mathbf{z}'$ is also an approximate KKT point of optimization $\min_{\mathbf{z} \in \mathcal{B}} H(\mathbf{z})$. Specifically, we have that there exists $\nu^* \in \mathbb{R}_{\geq 0}$

$$\|\nabla H(\mathbf{z}') + 2\nu^* \mathbf{z}'\| \leq \alpha(1 + \beta)c\epsilon + \mathcal{O}(\sqrt{L_H \epsilon_{\text{inn}}}),$$

$$\|\mathbf{z}'\| - 1 \leq 0,$$

$$\nu^* \geq 0, |\nu^*(\|\mathbf{z}'\|^2 - 1)| \leq c\beta\epsilon + \mathcal{O}(\sqrt{L_H \epsilon_{\text{inn}}}),$$

Finally, by Lemma E.5, $\mathbf{x}' = \Phi(\mathbf{z}')$ is an $\left[c\alpha(1 + \beta)\epsilon/\min\{1, l_\Phi\} + \mathcal{O}(\sqrt{L_H \epsilon_{\text{inn}}})\right]$-approximate KKT point of problem $\mathbf{P}$. By choosing appropriate $c$, e.g.,

$$c = \min\left\{\frac{l_\Phi}{\alpha(1 + \beta)}, 1\right\},$$

$\mathbf{x}' = \Phi(\mathbf{z}')$ becomes an $\left[\epsilon + \mathcal{O}(\sqrt{L_H \epsilon_{\text{inn}}})\right]$-approximate KKT point of problem $\mathbf{P}$. $\qquad \square$

## F  EXPERIMENTS SETTING

### F.1  PROBLEM FORMULATIONS AND INSTANCE GENERATION

#### F.1.1  NON-CONVEX QUADRATICALLY CONSTRAINED QUADRATIC PROGRAMMING

We consider the following non-convex QCQP problem:

$$\min_{L \leq \mathbf{x} \leq U} \quad \frac{1}{2}\mathbf{x}^\top \mathbf{Q}_0 \mathbf{x} + \mathbf{q}_0^\top \mathbf{x} + r_0, \tag{24}$$

$$\text{s.t.} \quad \frac{1}{2}\mathbf{x}^\top \mathbf{Q}_i \mathbf{x} + \mathbf{q}_i^\top \mathbf{x} + r_i \leq 0, \quad i = 1, \ldots, m, \tag{25}$$

where $\mathbf{x} \in [L, U]^n$ is the decision variable, $\mathbf{Q}_i \in \mathbb{R}^{n \times n}$ are symmetric matrices (not necessarily positive semidefinite), $\mathbf{q}_i \in \mathbb{R}^n$, and $r_i \in \mathbb{R}$.

**Instance Generation:** For the objective matrix $\mathbf{Q}_0$, we generate eigenvalues uniformly from $[-1, 1]$ to create a mix of positive and negative eigenvalues, ensuring non-convexity. We construct $\mathbf{Q}_0 = \mathbf{U}\text{diag}(\boldsymbol{\lambda})\mathbf{U}^\top/n$, where $\mathbf{U}$ is a random orthogonal matrix obtained via QR decomposition of a standard Gaussian matrix, and $\boldsymbol{\lambda}$ contains the mixed eigenvalues. The linear term $\mathbf{p}$ is sampled from $\mathcal{N}(0, 1/n)$. For the constraint matrices $\{\mathbf{Q}_i\}_{i=1}^m$, eigenvalues are uniformly sampled from $[-1, 1]$ to maintain the non-convex structure across constraints. Each $\mathbf{Q}_i$ is constructed using the same eigendecomposition approach with independent random orthogonal matrices and normalized by $1/n$. The corresponding linear terms $\mathbf{p}_i$ are sampled from $\mathcal{N}(0, 1/n)$. To ensure feasibility, we first generate a random initial point $\mathbf{x}_0 \sim \mathcal{N}(0, 0.1)$ and clip it to satisfy the box constraints with a margin of 0.1. The constraint bounds are then set as $b_i = \frac{1}{2}\mathbf{x}_0^\top \mathbf{Q}_i \mathbf{x}_0 + \mathbf{p}_i^\top \mathbf{x}_0 + \epsilon_i$, where $\epsilon_i \sim |\mathcal{N}(0, 1)| \cdot 0.1$ provides a feasibility margin. This construction guarantees that $\mathbf{x}_0$ is feasible and ensures the problem has a non-empty feasible region. For the illustrative example, we sample a 2-dimensional instance with 2 quadratic constraints.

#### F.1.2  JOINT CHANCE CONSTRAINED DC OPTIMAL POWER FLOW

In electrical power systems, operators must satisfy stochastic demand while maintaining system reliability across multiple nodes simultaneously. This presents a challenging multi-constraint optimization problem under uncertainty, where violations at any node can compromise system-wide stability.

We first introduce the standard DC optimal power flow (DC-OPF) problem:

$$\min_{\mathbf{p}, \boldsymbol{\theta}} \quad \sum_{i=1}^{G} \left( c_i^q p_i^2 + c_i^l p_i \right), \tag{26}$$

$$\text{s.t.} \quad \mathbf{p}^{\min} \leq \mathbf{p} \leq \mathbf{p}^{\max}, \quad \boldsymbol{\theta}^{\min} \leq \boldsymbol{\theta} \leq \boldsymbol{\theta}^{\max}, \tag{27}$$

$$\mathbf{B}_{\text{bus}}\boldsymbol{\theta} = \mathbf{p} - \mathbf{d}, \tag{28}$$

$$\mathbf{B}_{\text{line}}\boldsymbol{\theta} \leq \mathbf{S}^{\max}, \tag{29}$$

where $\mathbf{p} \in \mathbb{R}^G$ is the power generation vector, $\boldsymbol{\theta} \in \mathbb{R}^B$ are voltage phase angles, and $\mathbf{d} \in \mathbb{R}^B$ is the demand vector. The matrices $\mathbf{B}_{\text{bus}} \in \mathbb{R}^{B \times B}$ and $\mathbf{B}_{\text{line}} \in \mathbb{R}^{L \times B}$ are the bus and line susceptance matrices, with $B$ buses, $L$ transmission lines, and $G$ generators. The vector $\mathbf{S}^{\max} \in \mathbb{R}^L$ denotes maximum line capacities.

To handle dependency between decision variables and uncertain parameters, we eliminate the slack bus from the system equations. Let $\tilde{\mathbf{B}}_{\text{bus}} \in \mathbb{R}^{(B-1) \times (B-1)}$ be the reduced bus susceptance matrix, and $\tilde{\mathbf{p}} \in \mathbb{R}^{G-1}, \tilde{\boldsymbol{\theta}} \in \mathbb{R}^{B-1}, \tilde{\mathbf{d}} \in \mathbb{R}^{B-1}, \tilde{\boldsymbol{\xi}} \in \mathbb{R}^{B-1}$ be the corresponding reduced vectors. The phase angles for non-slack buses are:

$$\tilde{\boldsymbol{\theta}}(\boldsymbol{\xi}) = \tilde{\mathbf{B}}_{\text{bus}}^{-1} \left( \tilde{\mathbf{p}} - \tilde{\mathbf{d}} - \tilde{\boldsymbol{\xi}} \right), \tag{30}$$

and the slack bus generation adjusts to maintain power balance:

$$p_s(\boldsymbol{\xi}) = \sum_{i \in \mathcal{N}} (d_i + \xi_i) - \sum_{j \in \mathcal{G} \backslash s} p_j, \tag{31}$$

Table 4: Network characteristics and DC-OPF formulation complexity for PGLib test cases

| Power Grids | 200-Bus | 500-Bus |
|---|---|---|
| **Network Topology** | | |
| Buses | 200 | 500 |
| Generators | 69 | 145 |
| Branches | 245 | 597 |
| **DC-OPF Formulation** | | |
| *Decision Variables* | | |
| Real Power Generation ($P_g$) | 69 | 145 |
| Voltage Angles ($\theta$) | 199 | 499 |
| **Total Variables** | **268** | **644** |
| *Equality Constraints* | | |
| Power Balance | 200 | 500 |
| *Inequality Constraints* | | |
| Generator Limits | 138 | 290 |
| Voltage Angle Limits | 398 | 998 |
| Line Flow Limits | 490 | 1194 |
| **Total Inequalities** | **1026** | **2482** |

where $\mathcal{N}$, $\mathcal{G}$, and $s$ denote the sets of all buses, generator buses, and the slack bus, respectively.

The joint chance-constrained optimal power flow (JCC-OPF) extends the deterministic DC-OPF to handle demand uncertainty $\boldsymbol{\xi}$ while ensuring system reliability:

$$\min_{\tilde{\mathbf{P}}} \quad \mathbb{E}_{\boldsymbol{\xi}} \left[ \sum_{i=1}^{G} \left( c_i^q p_i(\boldsymbol{\xi})^2 + c_i^l p_i(\boldsymbol{\xi}) \right) \right], \tag{32}$$

$$\text{s.t.} \quad \mathbb{P} \left( \begin{array}{c} \mathbf{p}^{\min} \leq \mathbf{p}(\boldsymbol{\xi}) \leq \mathbf{p}^{\max} \\ \boldsymbol{\theta}^{\min} \leq \boldsymbol{\theta}(\boldsymbol{\xi}) \leq \boldsymbol{\theta}^{\max} \\ \mathbf{B}_{\text{line}}\boldsymbol{\theta}(\boldsymbol{\xi}) \leq \mathbf{S}^{\max} \end{array} \right) \geq 1 - \epsilon, \tag{33}$$

where $\epsilon \in (0, 1)$ is the prescribed violation probability. All operational constraints must be satisfied jointly with probability at least $1 - \epsilon$, ensuring comprehensive system reliability under uncertainty.

Given sampled scenarios $\boldsymbol{\xi}^{(k)}k = 1^N$, we have the Sample Average Approximation (SAA) for the chance constraints:

$$\frac{1}{N} \sum_{k=1}^{N} \mathbb{I} \left( \begin{array}{c} \mathbf{p}^{\min} \leq \mathbf{p}(\boldsymbol{\xi}^{(k)}) \leq \mathbf{p}^{\max} \\ \boldsymbol{\theta}^{\min} \leq \boldsymbol{\theta}(\boldsymbol{\xi}^{(k)}) \leq \boldsymbol{\theta}^{\max} \\ \mathbf{B}_{\text{line}}\boldsymbol{\theta}(\boldsymbol{\xi}^{(k)}) \leq \mathbf{S}^{\max} \end{array} \right) \geq 1 - \epsilon, \tag{34}$$

where $\mathbb{I}(\cdot)$ is the indicator function that equals 1 if all constraints are satisfied and 0 otherwise.

To solve it exactly via an existing solver such as GUROBI, we can reformulate it using the mixed-integer formulations by introducing binary variables $z^{(k)} \in \{0, 1\}$ for each scenario::

$$\frac{1}{N} \sum_{k=1}^{N} z^{(k)} \geq 1 - \epsilon, \tag{35}$$

$$\mathbf{p}^{\min} - M(1 - z^{(k)}) \leq \mathbf{p}(\boldsymbol{\xi}^{(k)}) \leq \mathbf{p}^{\max} + M(1 - z^{(k)}), \quad k = 1, \ldots, N, \tag{36}$$

$$\boldsymbol{\theta}^{\min} - M(1 - z^{(k)}) \leq \boldsymbol{\theta}(\boldsymbol{\xi}^{(k)}) \leq \boldsymbol{\theta}^{\max} + M(1 - z^{(k)}), \quad k = 1, \ldots, N, \tag{37}$$

$$\mathbf{B}_{\text{line}}\boldsymbol{\theta}(\boldsymbol{\xi}^{(k)}) \leq \mathbf{S}^{\max} + M(1 - z^{(k)}), \quad k = 1, \ldots, N, \tag{38}$$

$$z^{(k)} \in \{0, 1\}, \quad k = 1, \ldots, N, \tag{39}$$

where $z^{(k)}$ is a binary indicator that equals 1 if all constraints are satisfied for scenario $k$, and $M$ is a sufficiently large constant. This mixed-integer linear programming formulation provides a tractable approximation with convergence guarantees as $N$ increases.

**Instance Generation:** We use IEEE test systems from PGLIB (Babaeinejadsarookolaee et al., 2019), which provide standardized network topologies, transmission line parameters, generator characteristics, and baseline demand profiles for power system benchmarking. Uncertainty scenarios $\{\boldsymbol{\xi}^{(k)}\}_{k=1}^{N}$ are generated from multivariate normal distributions $\mathcal{N}(\mathbf{0}, \boldsymbol{\Sigma})$, where $\boldsymbol{\Sigma}$ captures spatial correlation in demand uncertainty. We construct $\boldsymbol{\Sigma}$ using an exponential decay model based on geographical distance: $\Sigma_{ij} = \sigma_i \sigma_j \exp\left(-\frac{d_{ij}}{\ell}\right)$, where $\sigma_i$ is the standard deviation of demand uncertainty at bus $i$ (set to 5% of nominal demand $d_i$), $d_{ij}$ is the electrical distance between buses $i$ and $j$ measured by the shortest path length in the network graph, and $\ell$ is the correlation length parameter that controls the spatial decay rate. We sample $\ell$ from $[1, 5]$ to generate instances with different correlation structures: small $\ell$ values produce localized correlations, while large $\ell$ values create system-wide correlated demand fluctuations.

## F.2 BASELINE ALGORITHMS AND HYPER-PARAMETERS

We implement the baselines as follows:

- **EPM** : Exact Penalty Method (Cartis et al., 2011). It solves an unconstrained reformulated problem of (**P**) as follows

$$\min_{\mathbf{x}} f(x) + \rho \|\mathbf{g}(x)\| \tag{40}$$

  where $\rho$ is the penalty parameter. Moreover, for a large enough parameter $\rho$, the critical points of the unconstrained reformulation (40) correspond to the KKT stationary points of the original problem (**P**), provided by usual constraint qualifications Nocedal & Wright (1999). Based on this reformulation, one can use any appropriate algorithm to solve (40), such as gradient descent methods, trust region methods Cartis et al. (2011).

- **ALM**: Augmented Lagrangian Methods (Sahin et al., 2019; Xie & Wright, 2019; Birgin et al., 2003).

$$\mathbf{x}_{k+1} = \arg\min_{\mathbf{x}} \{ f(\mathbf{x}) + \boldsymbol{\lambda}_k^T \mathbf{g}(\mathbf{x}) + \rho_k \|[\mathbf{g}(\mathbf{x})]_+\|^2 \}, \tag{41}$$

$$\boldsymbol{\lambda}_{k+1} = [\boldsymbol{\lambda}_k + \rho_k \cdot \mathbf{g}(\mathbf{x}_{k+1})]_+, \tag{42}$$

  where $\boldsymbol{\lambda}_k$ is the Lagrange multipliers, $\mathbf{g}(\mathbf{x})$ represents the constraint functions, and $\rho_k > 0$ is the dual step size. The inner unconstrained optimization problem is non-convex due to the non-convexity of the constraint functions $\mathbf{g}$ and is solved using gradient descent to a stationary point, making it an inexact method.

- **PPP** : Proximal-Point Penalty Method (Lin et al., 2022). For the optimization (**P**), let

$$\phi_k(\mathbf{x}) := f(\mathbf{x}) + \frac{\gamma_k}{2} \|\mathbf{x} - \mathbf{x}_k\|^2 + \frac{\beta_k}{2} \left( \|[\mathbf{g}(\mathbf{x})]_+\|^2 \right), \tag{43}$$

  where $\beta_k > 0$ is the penalty parameter and $\gamma_k > 0$ is the proximal parameter. A sufficiently large parameter $\gamma_k$ will make the problem (43) a strongly convex optimization provided by the weakly convex constraints $\mathbf{g}$.

  In each iteration, one will solve the problem (43) to a stationary point using (sub)gradient descent by fining $\mathbf{x}_{k+1}$ such that

$$\|\nabla \phi_k(\mathbf{x}_{k+1})\| \le \hat{\varepsilon}_k \tag{44}$$

  given a desired error $\hat{\epsilon}_k > 0$.

- **Hom-PGD$^+$**. Given the reformulated problem ($\mathbf{H}_{\text{inn}}$) and a step-size $\alpha_k$ in each iteration, we update by the rules

$$\mathbf{z}_{k+1} = \text{BP}_{\tilde{\mathcal{B}}}(\mathbf{z}_k - \alpha_k \nabla f(\Phi(\mathbf{z}_k))) \tag{45}$$

  where BP denotes the bisected projection onto the approximate unit ball $\tilde{\mathcal{B}}$, and $\Phi$ is the INN-learned homeomorphism. The solution is mapped to the original space after convergence as $\mathbf{x}^* = \boldsymbol{\Phi}(\mathbf{z}^*)$.

- **IPOPT**: Interior Point Optimizer, a state-of-the-art nonlinear programming solver that implements a primal-dual interior point method with line search. It uses exact second-order information and adaptive barrier parameter updates to handle inequality constraints through logarithmic barrier functions. IPOPT is particularly effective for large-scale continuous optimization problems with smooth nonlinear constraints.

- **GUROBI**: Commercial mixed-integer programming solver that employs branch-and-bound algorithms with advanced cutting plane generation, presolving techniques, and heuristics. For the SAA formulation of the JCC-OPF problem, GUROBI provides the exact optimal solution to the mixed-integer linear program, serving as the ground truth baseline for comparison with other approximate methods.

**Gradient calculation**: For simple quadratic objective functions, gradients are calculated via closed-form formulations. Other non-trivial gradient calculations across the various algorithms are implemented using auto-differentiation in PyTorch. We note that replacing auto-differentiation with closed-form gradient implementations could further improve the computational efficiency of the algorithms.

**Handing Non-differentiable Chance Constraint**: Since the indicator-based chance constraint is non-differentiable, making direct application of all first-order algorithms challenging. To tackle this challenge, we compute the robust scenario penalty following (Nemirovski & Shapiro, 2006), which computes the constraint violation for the worst-case scenario and treats it as a penalty in INN training or as the constraint violation/residual/penalty for other first-order algorithms. Specifically, we replace the non-differentiable indicator function with a smooth approximation:

$$\frac{1}{N} \sum_{k=1}^{N} \mathbb{I}(\mathbf{g}(\mathbf{x}, \boldsymbol{\xi}_k) \leq 0) \geq 1 - \epsilon \quad \Rightarrow \quad \max_{k \in \{1, \ldots, N\}} [\mathbf{g}(\mathbf{x}, \boldsymbol{\xi}^{(k)})]_+ \leq 0 \tag{46}$$

Notably, when evaluating the chance constraint feasibility, we still follow the exact indicator-based formulation, which is used in the membership oracle for our Hom-PGD$^+$ method to ensure accurate feasibility assessment during optimization or the final evaluation for solutions obtained from different algorithms.

**Step-size**: Theoretically, different algorithms employ their own step size selection strategies, such as explicit dependence on smoothness and convexity parameters, or implicit step sizes that depend on the optimal objective value Grimmer (2024b). For practical implementation, we initialize a fixed step size (e.g., $10^{-3}$) and decay it by a factor of 0.999 if the objective value does not decrease, which helps identify a sufficient step size for convergence.

**Computation environment**: All algorithms are implemented in Pytorch and executed on an Ubuntu server with an NVIDIA A800 GPU and an AMD EPYC 7763 64-Core Processor.

### F.3 INVERTIBLE NEURAL NETWORK IMPLEMENTATION

We adopt the coupling layer-based INN as our homeomorphism approximator. Specifically, it consists of 3 layers, each layer containing two sub-layers:

- **Invertible Linear Layers**: Following the GLOW architecture (Kingma & Dhariwal, 2018), we employ invertible linear layers with learnable bias terms. These layers implement affine transformations of the form $\mathbf{y} = \mathbf{W}\mathbf{x} + \mathbf{b}$, where the weight matrix $\mathbf{W}$ is constrained to be invertible through LU decomposition parameterization. This parameterization ensures invertibility by construction while allowing efficient computation of the log-determinant of the Jacobian as the sum of logarithms of the diagonal elements from the decomposition.

- **Coupling layer**: We implement coupling layers using MADE (Masked Autoencoder for Distribution Estimation) (Germain et al., 2015), which enables highly efficient computation through masked forward propagation. MADE applies element-wise affine transformations in an autoregressive manner, where each output dimension is conditioned on all preceding input dimensions according to a predefined ordering. This structure maintains the coupling layer property while providing computational efficiency through parallelizable masked operations.

**Conditional Embedding**: To incorporate conditional input $\boldsymbol{\theta}$, we employ a dedicated fully connected neural network that embeds the conditional information into a latent representation. This embedding

is then added to the intermediate variables at each coupling layer, allowing the transformation to adapt based on the conditioning information. For the scenario-based input in JCC-DC-OPF, where the number of scenarios can vary across problem instances, we adopt a DeepSet-based architecture (Zaheer et al., 2017) to handle the permutation invariance property inherent in scenario sets. The DeepSet encoder maps variable-size scenario collections into a fixed-dimensional embedding space (64 dimensions in our implementation), ensuring consistent representation regardless of the number of scenarios while preserving the exchangeability of individual scenarios.

**INN Training**: We apply the Adam (Kingma & Ba, 2014) optimizer to train the INN with a batch size of 64, where each batch is sampled from the unit ball and input parameter space. We set the initial learning rate to $5 \times 10^{-4}$ with a decay factor of 0.9 every 1,000 iterations. The maximum number of training iterations is set to 10,000. The coefficient for the penalty term is 10, and the Lipschitz regularizer is 0.1.

# G SUPPLEMENTARY EXPERIMENTS RESULTS

## G.1 INN TRAINING DETAILS

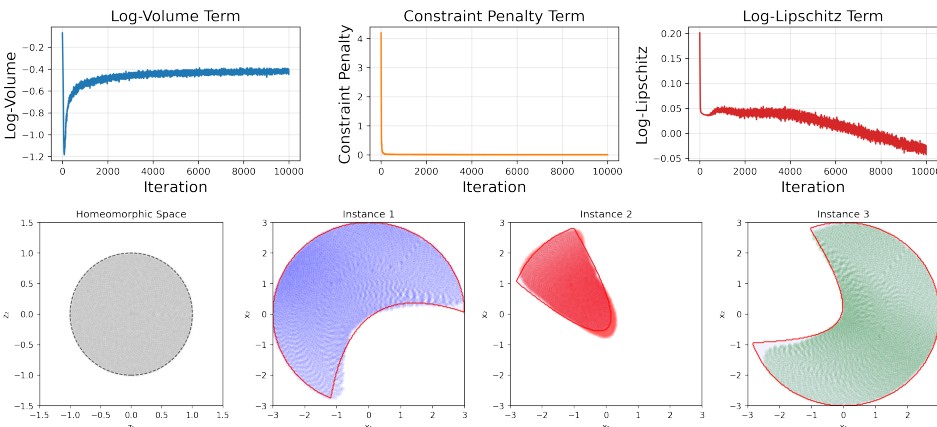

Figure 6: Training and evaluation of the 3-layer INN. Top: convergence of the volume term, penalty term, and Lipschitz term across different sampled input parameters $\theta$ during training. Bottom: visualization of the trained INN mapping the unit ball to different target constraint sets under various test input parameters. The training algorithm stably learns homeomorphisms by maximizing volume within constraints while regularizing the Lipschitz constant, demonstrating effective approximation quality and capturing the complex constraint geometry under unseen input parameters.

We provide training details for the invertible neural network used in homeomorphism learning. Specifically, we examine the convergence behavior of the training loss components and demonstrate the network's ability to learn bidirectional mappings between unit balls and constraint sets.

**Training Convergence**: The INN is trained by optimizing three loss components: the volume term (ensuring volume preservation), the penalty term (enforcing constraints), and the Lipschitz term (controlling smoothness). Figure 6 (top) shows the convergence of these components across different sampled input parameters $\theta$, demonstrating stable optimization. The training dynamics include three stages:

- Initialization phase: The INN parameters are randomly initialized (e.g., Gaussian), causing the initial mapping output $\Phi(\mathcal{B})$ to violate the constraint $\Phi(\mathcal{B}) \subseteq \mathcal{K}$. This results in a large constraint penalty term that dominates the total loss (as evident in the second subfigure showing high penalty loss).
- Shrinking phase: To reduce constraint violations, the network learns to shrink the mapped region and adjust its position. This shrinking decreases the volume (and thus log-volume drop), while it also reduces the constraint penalty by pushing $\Phi(\mathcal{B})$ fits within $\mathcal{K}$. During this phase, minimizing the penalty term takes priority over maximizing volume.

- Expansion phase: Once the constraint is approximately satisfied (indicated by low penalty loss in the second subfigure), the volume maximization term becomes dominant. The network then learns to expand $\Phi(\mathcal{B})$ to occupy as much of $\mathcal{K}$ as possible, ultimately approaching a homeomorphism approximately.

**Learned Mapping Properties**: The trained INN learns parameter-dependent bidirectional mappings. In the forward direction, it maps the unit ball to constraint sets that vary with the input parameter $\theta$. In the inverse direction, it maps points from these constraint sets back to the unit ball, providing a normalized representation of the feasible region.

- Assumption 2 requires bounded homeomorphism error, meaning the trained INN must approximate the true homeomorphism between the unit ball and the constraint set with bounded error $\epsilon_{\mathrm{inn}}$. Due to the bijective property of homeomorphisms, this is equivalent to requiring that $\Phi(\mathcal{B})$ closely approximates the true constraint set $\mathcal{K}$ (or equivalently, that $\Phi^{-1}(\mathcal{K})$ approximates $\mathcal{B}$). For straightforward visualization and comparison, we validate the forward direction by examining how well $\Phi(\mathcal{B})$ covers and matches the true constraint set $\mathcal{K}$.

- As shown in Figure 6, the mapped set $\Phi(\mathcal{B})$ accurately approximates the non-convex geometry of the target constraint set under different input parameters, demonstrating the effectiveness of our INN training method. To quantify this approximation quality, we can compute the Hausdorff distance between $\Phi(\mathcal{B})$ and $\mathcal{K}$, defined as

$$d_H(\Phi(\mathcal{B}), \mathcal{K}) = \max \left\{ \sup_{x \in \Phi(\mathcal{B})} \inf_{y \in \mathcal{K}} \|x - y\|, \sup_{y \in \mathcal{K}} \inf_{x \in \Phi(\mathcal{B})} \|x - y\| \right\},$$

which measures the maximum distance between the two sets. if $d_H(\Phi(\mathcal{B}), \mathcal{K}) = 0$, then $\Phi(\mathcal{B}) = \mathcal{K}$ given $\mathcal{B} \cong \mathcal{K}$, meaning INN $\Phi$ is a perfect homeomorphic mapping between $\mathcal{B}$ and $\mathcal{K}$ and $\epsilon_{\mathrm{inn}} = 0$.

## G.2 ABALATION STUDY

We conduct ablation studies on QCQP optimization problems to analyze two key aspects of our method: **(i)** *INN Complexity and Performance*: We examine how INN depth (1/3/5 layers) affects approximation error (Assumption 2), Lipschitz constants, and downstream optimization performance, demonstrating that a 3-layer INN achieves the best balance between approximation capability and parameter complexity. **(ii)** *Bisection Complexity and Performance*: We show that reducing bisection iterations decreases per-iteration cost but may increase the optimality gap.

## G.3 MORE QCQP RESULTS

We visualize the comparison of Hom-PGD$^+$ and other baseline methods on QCQP optimization under different input parameters. We show the convergence with respect to iteration and total time, the constraint violation with respect to running time and per-iteration cost, and visualize the iteration trajectory of different methods.

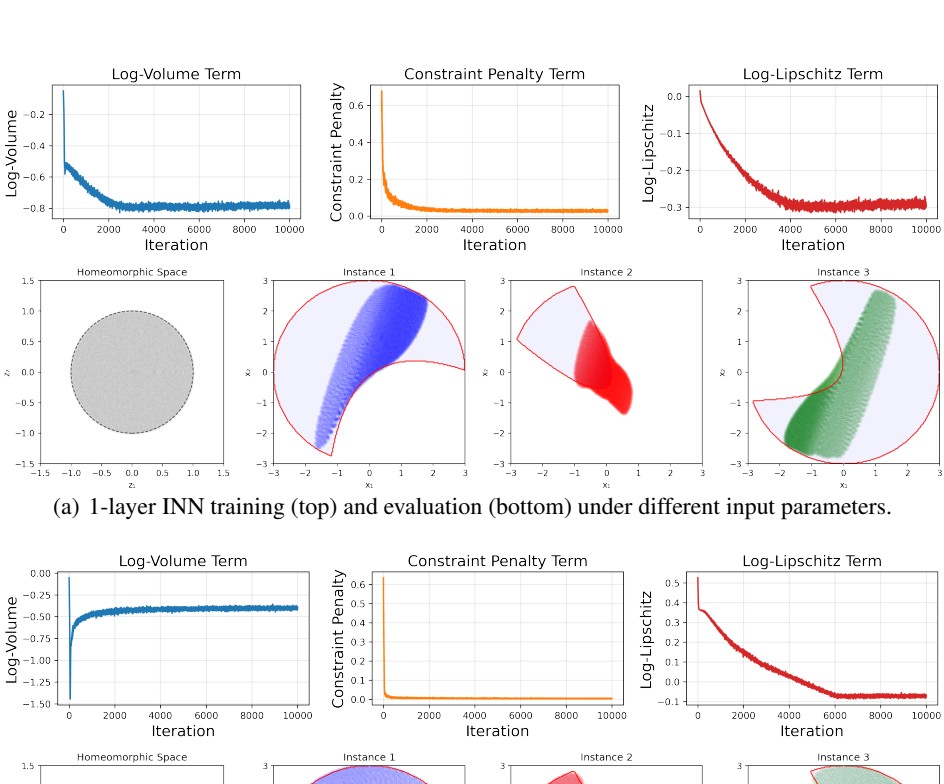

(a) 1-layer INN training (top) and evaluation (bottom) under different input parameters.

(b) 5-layer INN training (top) and evaluation (bottom) under different input parameters.

Figure 7: INN training and evaluation across different network depths. Top panels show training loss (Eq. (1)) convergence, including volume, penalty, and Lipschitz terms. Bottom panels visualize learned mappings under different input parameters. Key observations: (i) The 1-layer INN fails to capture constraint geometry accurately (average Hausdorff distance $> 1.5$), while 3- and 5-layer INNs achieve better approximation quality (average Hausdorff distance $< 0.3$). (ii) The 1-layer INN exhibits the smallest Lipschitz constant due to limited model expressiveness, whereas deeper networks show larger Lipschitz constants during training. The trade-off between approximation accuracy and smoothness can be controlled via the Lipschitz regularization term in the INN loss function.

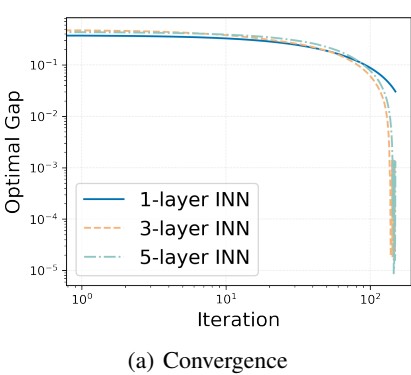

(a) Convergence

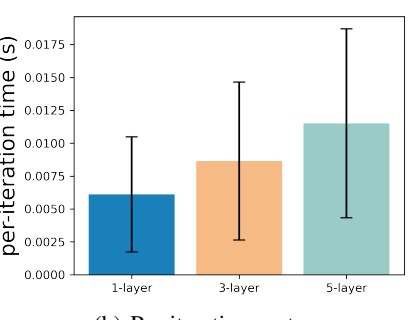

(b) Per-iteration costs.

Figure 8: Performance comparison of Hom-PGD$^+$ across different INN architectures (1-layer, 3-layer, 5-layer). Single-layer INNs exhibit poor approximation capability, leading to large learning errors when approximating the constraint set. In contrast, 3-layer and 5-layer INNs provide sufficient representational capacity to capture the constraint set and demonstrate superior convergence behavior.

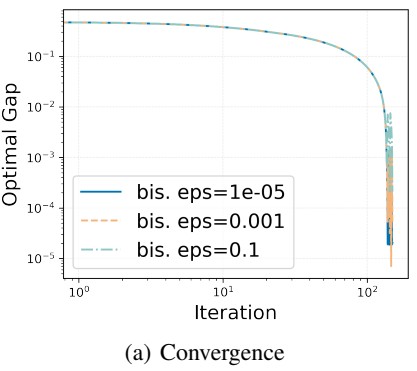

(a) Convergence

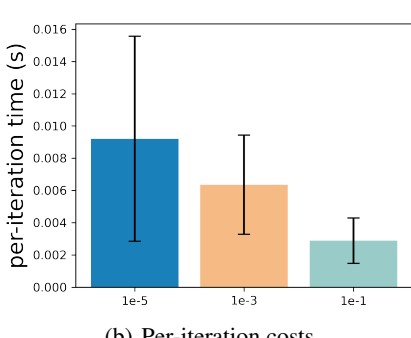

(b) Per-iteration costs

Figure 9: Performance comparison of Hom-PGD$^+$ across different bisection tolerance levels ($10^{-5}$, $10^{-3}$, $10^{-1}$). Higher tolerance values accelerate the algorithm by reducing bisection iterations within the projection operator, but result in larger optimality gaps due to less precise convergence to the constraint boundary.

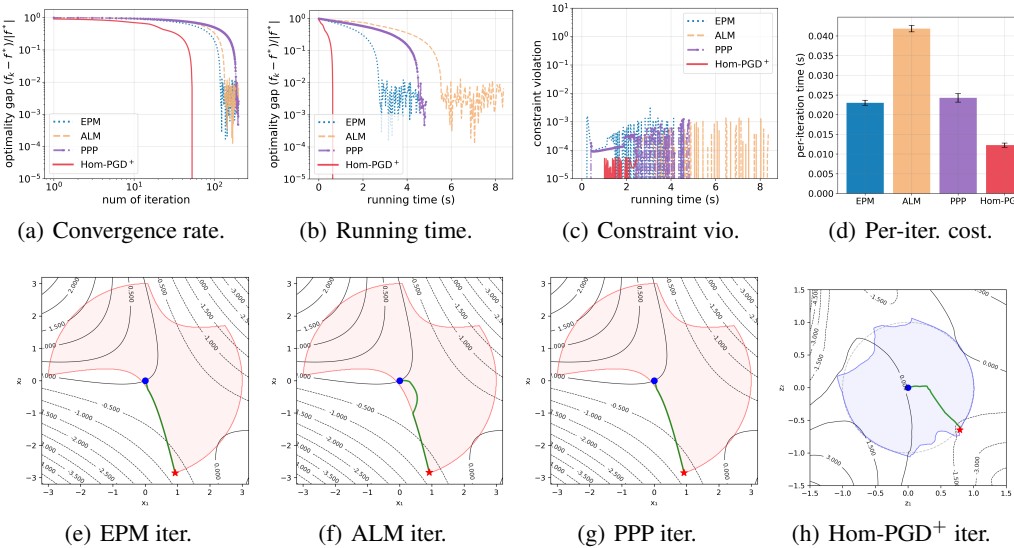

(a) Convergence rate.  (b) Running time.  (c) Constraint vio.  (d) Per-iter. cost.

(e) EPM iter.  (f) ALM iter.  (g) PPP iter.  (h) Hom-PGD$^+$ iter.

Figure 10: Illustrative examples of Hom-PGD$^+$ for solving QCQP with non-convex BH constraints.

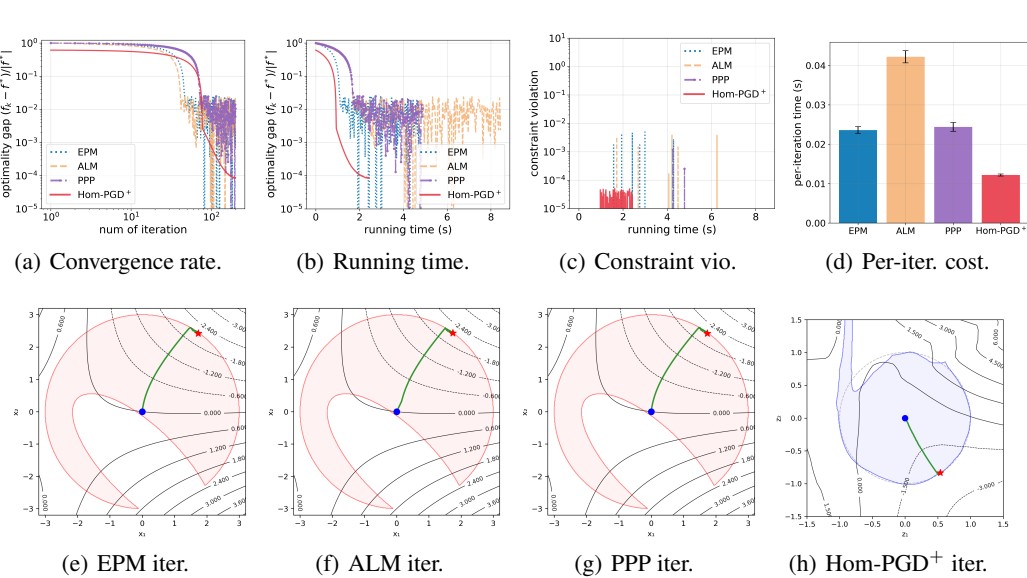

Figure 11: Illustrative examples of Hom-PGD$^+$ for solving QCQP with non-convex BH constraints.

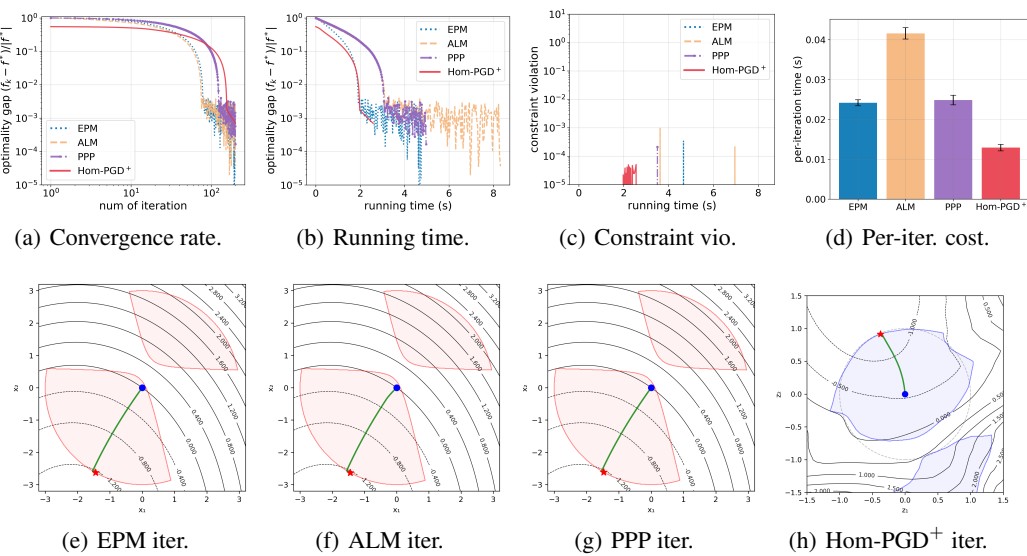

Figure 12: Illustrative examples of Hom-PGD$^+$ for solving QCQP with non-BH constraints.

