# OpenReview forum: "Hom-PGD+: Fast Reparameterized Optimization over Non-convex Ball-Homeomorphic Set"
_ICLR.cc/2026/Conference — Submitted to ICLR 2026_

### Official Review · Reviewer_rwv2 · 2025-10-30

**Soundness:** 3
**Presentation:** 3
**Contribution:** 3
**Rating:** 6
**Confidence:** 3

**Summary:**

This paper proposes Hom-PGD+ for constrained optimization over ball-homeomorphic sets. An invertible neural network (INN) is trained to learn a homeomorphism from the unit ball to the constraint set, after which the problem is reformulated as an approximately ball-constrained one and solved via projection gradient descent with bisected projection. The final solution is mapped back using the inverse homeomorphism.

The method requires INN training and incurs $O(W)$ run-time complexity for the bisected projection, where $W$ is the number of INN parameters. Assuming the learned mapping has approximation error $\epsilon_{\mathrm{inn}}$, Hom-PGD+ with a constant step size finds an $\epsilon + \sqrt{L_H \epsilon_{\mathrm{inn}}}$-approximate KKT point in $O(L_H \epsilon^{-2})$ iterations.

Experiments on QCQPs and power-grid optimization problems, along with comparisons to baseline methods, support the effectiveness of the approach.

**Strengths:**

The proposed algorithm handles constrained sets with ball-homeomorphic structure by learning the homeomorphism using an invertible neural network. This reformulation reduces the problem to a ball-constrained one, enabling the use of projection gradient descent to solve the transformed optimization.

**Weaknesses:**

Assumption 2 for the convergence guarantee appears difficult to verify in practice. In particular, it is unclear how to estimate $\epsilon_{\mathrm{inn}}$.

**Questions:**

- How should the $k$’s (splitting components) be chosen in each layer of the INN?

- What does $L_H$ stand for in Theorem 1?

- In Appendix B.5, Eq. (5), should the supremum be taken instead of the infimum?

- In Appendix B.5, Eq. (8), the index $i$ is unused in the summation and should be clarified.

---

> ### Author Response · Authors · 2025-11-24
> **Official Response by Authors (Part I)**
>
> Dear Reviewer rwv2,
>
> We thank the reviewer for the effort in reviewing our paper. We are happy to receive your overall positive feedback. For your comments and questions, we address them one by one as follows.
>
> -----
> > *Comment 1. Assumption 2 for the convergence guarantee appears difficult to verify in practice. In particular, it is unclear how to estimate $\epsilon_{inn}$.*
> -----
>
> **Response:**
>
> Thank you for this important comment. We acknowledge that Assumption 2 is difficult to verify for high-dimensional problems. We appreciate the opportunity to clarify (1) why it is necessary for our convergence analysis, (2) how our algorithmic design helps satisfy it in practice, and (3) empirical performance on QCQP and other problems.
>
> Assumption 2 consists of two key components:
>
> (i) **Bounded homeomorphism error:**
> $$
> \mathcal{B}(0,1-\epsilon_{\operatorname{inn}}) \subseteq \Phi^{-1}(\mathcal{K}) \subseteq \mathcal{B}(0,1+\epsilon_{\operatorname{inn}}), \quad \|\boldsymbol{\psi}-\Phi\| \leq \epsilon_{\rm inn}.
> $$
> - This ensures that $\Phi$ closely approximates the true homeomorphism $\psi$ between the unit ball and the constraint set $\mathcal{K}$. Our INN training framework (Equation 1) is specifically designed to achieve this through: (a) the constraint penalty term ensuring $\Phi(\mathcal{B}) \subseteq \mathcal{K}$, and (b) the volume maximization term ensuring maximal coverage of $\Phi(\mathcal{B})$ within $\mathcal{K}$.
>
> - **$\epsilon_{\text{inn}}$ (INN approximation error):** It can be represented as the Hausdorff distance between the true constraint set and the INN-approximated set (zero Hausdorff distance indicates the INN perfectly maps the ball to the constraint set, and reaches the homeomorphism with $\epsilon_{inn}=0$). Using random sampling on boundary points, we estimate the average distance to be less than 0.3 across different problem instances, as visualized in **Figure 6 (Appendix G)**. This demonstrates that the approximated constraint set closely matches the target set.
>
> (ii) **Bounded Jacobian approximation error:**  $\|J_{\boldsymbol{\psi}} -J_{\Phi}\| \leq \epsilon_{\rm inn}.$
> - This stronger condition ensures that the derivatives (local sensitivities) of $\Phi$ and $\psi$ are close, which is crucial for bounding the approximation error of KKT solutions in our convergence analysis. While this is a strong requirement, it enables worst-case theoretical guarantees and advances state-of-the-art convergence results for non-convex constrained optimization.
>
> - Although we cannot directly verify this without knowing the ground truth $\psi$, our Lipschitz regularization (spectral norm penalty on $\mathrm{J}_{\Phi}$ in Equation 1) prevents $\Phi$ from having large local variations. **Figures 6-7 (Appendix G)** show that the Lipschitz regularization term decreases consistently during training, indicating stable, smooth mappings.
>
> **Practical perspective:** For high-dimensional problems such as chance-constrained optimization, directly verifying these conditions is indeed non-trivial. However, Assumption 2 serves primarily to enable worst-case theoretical guarantees. In practice, our focus is on solution quality metrics—specifically, the optimality gap—which demonstrates strong performance compared to industrial solvers in our high-dimensional experiments (Section 5.3).
>
> We have expanded this discussion below Assumption 2 (**lines 313-319**) in the revised manuscript.
>
> -----
> > *Question 1. How should the $\boldsymbol{k}$'s (splitting components) be chosen in each layer of the INN?*
> -----
>
> **Response:**
> Thank you for your question on the INN design. We follow the standard coupling layer structure from [1]:
>
> - **Splitting strategy:** The splitting dimension $k$ is fixed as $\lfloor n/2 \rfloor$ in each coupling layer, meaning the first half of the variables remains fixed while transforming the second half.
>
> - **Variable permutation:** After each coupling layer, we apply an invertible linear transformation [1] that "permutes" the variables. This ensures that different variable subsets are transformed across layers, allowing the network to capture complex dependencies throughout the entire input space.
>
> This architecture design has been shown to be effective for learning expressive invertible mappings in [1] and works well in our constraint learning setting. We have added these clarifications in **lines 1156-1168** of our manuscript.
>
> [1] Kingma DP, Dhariwal P. Glow: Generative flow with invertible 1x1 convolutions. Advances in Neural Information Processing Systems. 2018;31.

---

> ### Author Response · Authors · 2025-11-24
> **Official Response by Authors (Part II)**
>
> -----
> > *Question 2. What does $L_H$ stand for in Theorem 1?*
> -----
>
> **Response:**
> $L_H$ denotes the Lipschitz constant of the gradient of the function $H = f \circ \Phi$, where $\Phi$ is the learned INN. More precisely, as detailed in **line 365**, we have
> $$
> L_H = u_{\Phi}^2 L_f + L_{\Phi} L_{f, 0},
> $$
> where $u_{\Phi}$ is the forward Lipschitz constant of $\Phi$, $L_f$ is the Lipschitz constant of the gradient of $f$, $L_{\Phi}$ is the Lipschitz constant of $J_\Phi$, and $L_{f, 0}$ is the Lipschitz constant of $f$.
>
> Moreover, we acknowledge that we should first define $L_H$ before presenting Theorem 1. To this end, we reorganized our assumption section (**lines 290-319**) in the revised manuscript.
>
> -----
> > *Question 3. In Appendix B.5, Eq. (5), should the supremum be taken instead of the infimum?*
> -----
>
> **Response:**
> Yes, it's a typo. Thanks for pointing it out. We have modified it (**line 1241**) to
> $$
> \mathrm{L}(\boldsymbol{\psi}) = \sup_{\mathbf{z}\neq\mathbf{u}\in\mathcal{K}} \frac{\|\boldsymbol{\psi}(\mathbf{z}) - \boldsymbol{\psi}(\mathbf{u})\|}{\|\mathbf{z} - \mathbf{u}\|}.
> $$
>
> -----
> > *Question 4. In Appendix B.5, Eq. (8), the index $\boldsymbol{i}$ is unused in the summation and should be clarified.*
> -----
>
> **Response:**
> Thanks for pointing it out. We have modified this typo (**line 1257**) to
> $$
> \widehat{V}\left(\Phi_{\boldsymbol{\theta}}(\mathcal{B})\right) = \frac{1}{V(\mathcal{B})} \int_{\mathcal{B}} \sum_{i=1}^n \sum_{j=1}^l \log \sigma_i\left(J_{\Phi_{\boldsymbol{\theta}}^j}\left(\mathbf{z}^j\right)\right)  d\mathbf{z} + \log V(\mathcal{B}).
> $$
>
> We will carefully check the typos in our paper to improve the flow. Thanks again for finding these issues.

---

### Official Review · Reviewer_F6JT · 2025-10-31

**Soundness:** 3
**Presentation:** 2
**Contribution:** 2
**Rating:** 4
**Confidence:** 4

**Summary:**

This paper studies parametric constrained optimization problems where the constraint set is compact and homeomorphic to the unit ball. The authors propose a transformation of the problem, where the constraint becomes simply the unit ball and study the equivalence of the stationary points between the two problems. Moreover, they propose a method that estimates the homeomorphism using an invertible neural network (INN), leading to a constraint set that only approximates the unit ball and thus requires a different projection that is then handled by the BP operator (Algorithm 2). The convergence rate of the method to approximate stationary points is proven and finally numerical simulations that showcase the superiority of the method compared to other constrained optimization solvers are presented.

**Strengths:**

1. The main idea of the paper is interesting and easy to follow. The problem that is considered is interesting and the proposed way of tackling constrained problems seems different from standard optimization approaches.
2. The experimental results show that the proposed algorithm performs similarly or better than state-of-the-art methods.

**Weaknesses:**

1. The contribution seems incremental compared to prior works, mainly combining approaches from the related literature, namely [1, 2]. The proofs are mostly straightforward applications of existing techniques.
2. The obtained complexity results are not easy to compare to existing results, since the authors utilize a neural network to approximate the homeomorphism. Although one can approximate such functions using NNs, the complexity of this procedure may not be reflected in the obtained guarantees. In that sense the phrase “it achieves a per-iteration complexity of $O(W)$, where $W$ is the number of INN parameters and setting $W = O(n^2)$ is sufficient to achieve strong performance in practice” is not suitable to describe the supposedly better performance of the method in theory, but rather the better performance in some practical applications. Moreover, further discussion on Assumption 2 is required.
3. The authors claim that the proposed method can find application in many practical scenarios and yet provide experimental results for standard quadratically constrained quadratic problems and electrical power systems. Are there any relevant examples of optimization problems from Machine Learning where the method is applicable?

---

References:

[1] Enming Liang, Minghua Chen, and Steven H. Low. Low complexity homeomorphic projection to ensure neural-network solution feasibility for optimization over (non-)convex set. In International Conference on Machine Learning. PMLR, 2023.

[2] Enming Liang and Minghua. Chen. Efficient bisection projection to ensure neural-network solution feasibility for optimization over general set. In International Conference on Machine Learning. PMLR, 2025.

**Questions:**

Minor questions and typos:
- In (P), the unit ball $\mathcal{B}$ is the Euclidean norm ball, $\\{x \in \mathbb R^n: \|x\|_2 \leq 1\\}$?
- In general, the phrase projected gradient descent is more common than “projection gradient descent” that is found throughout the text.
- Line 222: “the Lipschitz of the homeomorphism”. Should it be the Lipschitz continuity constant of the homeomorphism?
- Line 225: “to train the homeomorphism” should be phrased better.
- Definition 3.2: symbol $\mathcal{H}^n$ not defined. It is better to provide a link to the notation. The same holds for the assumptions of the paper which are found in the appendix.
- Line 257: What is a non-perfect and non-convex ball? Do the authors mean it in a topological sense? I suggest adding further details on such concepts which are currently missing.
- Line 323: "this results".
- Line 366: "The final convergent solution".
- Line 1390: $L_f$-smooth instead of $L_f$ smooth.
- Line 871: "Nerual" -> "Neural"
- Line 1558: The KKT conditions are presented and then the Lagrangian of the problem is presented in line 1588 and the KKT conditions for P follow. I would suggest switching the order.
- Lemma D.4: $ z^* = \\psi^{-1}(x^*) $ is missing.
- Prood of Lemma D.4: the statement of the lemma is that $ z^* $ is a KKT point of $ H $, while the proof starts with $ z^* $ being a stationary point of P. Lines 1628-1631 follow by multiplying 1594 with $ J_{\\psi}(z^*)^T $.
- Line 1649: should be “implies”.
- Line 1659: What does "$\lambda^*$ as eq.18 mean"?
- Line 1664: "holds" -> "hold".
- Line 1820: by definition of the normal cone.

---

> ### Author Response · Authors · 2025-11-24
> **Official Response by Authors (Part I)**
>
> Dear Reviewer F6JT,
>
> We thank the reviewer for the effort in reviewing our paper carefully. We appreciate the positive feedback on the interesting methods of our methods. For your comments and questions, we respond to them one by one in the following.
>
> -----
> > *Comment 1. The contribution seems incremental compared to prior works, mainly combining approaches from the related literature, namely [1, 2]. The proofs are mostly straightforward applications of existing techniques.*
> ----
>
> [1] Liang E, Chen M, Low S. Low complexity homeomorphic projection to ensure neural-network solution feasibility for optimization over (non-) convex set. ICML 2023
>
> [2] Liang E, Chen M. Efficient Bisection Projection to Ensure Neural-Network Solution Feasibility for Optimization over General Set. ICML 2025
>
> **Response:** We thank the reviewer for this comment. While our work is motivated by similar INN-based mapping techniques as in [1,2], our problem setting, algorithmic framework, and theoretical contributions are fundamentally different.
>
> - **Problem Setting and Algorithmic Framework:**
>
>     - The works [1,2] focus primarily on guaranteeing the feasibility of neural network outputs for constrained optimization—essentially addressing the projection subproblem. In fact, [1] can be viewed as a single component (the projection step in Algorithm 2) within our overall projected gradient descent (PGD) framework.
>
>     - In contrast, we develop a complete PGD framework that achieves efficient optimization over non-convex constraint sets with provable convergence guarantees. Our contribution extends beyond feasibility to providing a full iterative optimization algorithm with comprehensive convergence analysis, which [1,2] do not address.
>
> - **Theoretical Analysis.**
>
>     We provide rigorous landscape analysis (Proposition 4.1) and convergence guarantees (Theorem 1) establishing $O(\epsilon^{-2})$ convergence rates for non-convex constrained optimization—a substantial improvement over existing $O(\epsilon^{-3})$ or $O(\epsilon^{-4})$ rates (discussed below Theorem 1). Additionally, our analysis also provides explicit bounds that quantify the INN approximation error in the final solution.
>
> These contributions represent a significant advancement beyond the feasibility-focused techniques in [1,2], establishing both the algorithmic framework and theoretical foundations for efficient non-convex constrained optimization. We have clarified this positioning in the Introduction **(lines 85–90)** of the revised manuscript.
>
> -----
> > *Comment 2.1. The obtained complexity results are not easy to compare to existing results, since the authors utilize a neural network to approximate the homeomorphism. Although one can approximate such functions using NNs, the complexity of this procedure may not be reflected in the obtained guarantees. In that sense the phrase "it achieves a per-iteration complexity of $O(W)$, where $W$ is the number of INN parameters and setting $W=O\left(n^2\right)$ is sufficient to achieve strong performance in practice" is not suitable to describe the supposedly better performance of the method in theory, but rather the better performance in some practical applications.*
> ----
>
> **Response:**
>
> We thank the reviewer for this important observation and the opportunity to clarify our complexity claims.
>
> - Our theoretical analysis establishes that the per-iteration cost is $\mathcal{O}(W)$, where $W$ denotes the number of INN parameters. We acknowledge that it does not automatically guarantee lower per-iteration complexity than previous methods. The primary theoretical advantage of our framework lies in achieving a faster *convergence rate* ($\mathcal{O}(\epsilon^{-2})$ vs. $\mathcal{O}(\epsilon^{-3})$ or $\mathcal{O}(\epsilon^{-4})$) for optimization over non-convex constraint sets, as detailed in Table 1.
>
> - The statement about $W=\mathcal{O}(n^2)$ refers to our *empirical findings* rather than theoretical guarantees. In practice, we use a 3-layer INN with width $n$, yielding $W=\mathcal{O}(n^2)$ parameters and a per-iteration complexity of $\mathcal{O}(n^2)$. This configuration delivers strong performance across our benchmarks, demonstrating the practical efficiency of our approach. However, we agree that it should be clearly distinguished from theoretical complexity guarantees.
>
> In the manuscript (**line 394-398**), we have clarified that the $W=\mathcal{O}(n^2)$ setting and associated per-iteration complexity reflect empirical observations rather than theoretical claims.

---

> ### Author Response · Authors · 2025-11-24
> **Official Response by Authors (Part II)**
>
> -----
> > *Comment 2.2: Moreover, further discussion on Assumption 2 is required.*
> ----
>
> **Response:** Thanks for the comment on Assumption 2. We appreciate the opportunity to clarify (1) why it is necessary for our convergence analysis, (2) how our algorithmic design helps satisfy it in practice, and (3) empirical performance on QCQP and other problems.
>
> Assumption 2 consists of two key components:
>
> (i) **Bounded homeomorphism error:**
> $$
> \mathcal{B}(0,1-\epsilon_{\operatorname{inn}}) \subseteq \Phi^{-1}(\mathcal{K}) \subseteq \mathcal{B}(0,1+\epsilon_{\operatorname{inn}}), \quad \|\boldsymbol{\psi}-\Phi\|\leq \epsilon_{\rm inn}.
> $$
> - This condition ensures that $\Phi$ closely approximates the true homeomorphism $\psi$ between the unit ball and the constraint set $\mathcal{K}$. Our INN training framework is specifically designed to achieve this through the loss function in Equation (1), which simultaneously encourages: (a) $\Phi(\mathcal{B}) \subseteq \mathcal{K}$ via the constraint penalty term, and (b) maximal coverage of $\Phi(\mathcal{B})$ within $\mathcal{K}$ via the volume maximization term.
>
> - For QCQP problems specifically, we also provide additional validation in Appendix G (**Page 45**) to show how the INN approximates the target constraints under different input parameters with small errors (**Figs. 6-7**).
>
> (ii) **Bounded Jacobian approximation error:**  $\|J_{\boldsymbol{\psi}}-J_{\Phi}\|\leq \epsilon_{\rm inn}.$
> - This condition is stronger than merely controlling the function value error; it ensures that the derivatives (or local sensitivities) of $\Phi$ and $\psi$ are close. This assumption is indeed needed in our theoretical analysis and is crucial for bounding the approximation error of the KKT solutions in constrained optimization problems. Although this is a strong requirement in theory, it provides an essential upper bound for the worst-case analysis and advances the state-of-the-art results for non-convex constrained optimization.
>
> - While we cannot directly verify this condition without knowing the ground truth $\psi$, our algorithm design promotes this property through Lipschitz regularization (the spectral norm penalty on $\mathrm{J}_{\Phi}$ in Equation (1)), which prevents $\Phi$ from having large local variations. **Figs. 6-7** in Appendix G (**Page 44-46**) show that the Lipschitz regularization term decreases consistently during training, indicating that $\Phi$ achieves stable, smooth mappings.
>
> (iii) For high-dimensional problems such as chance-constrained optimization, directly verifying these conditions is indeed non-trivial. However, we emphasize that Assumption 2 serves primarily to enable **worst-case theoretical guarantees**. In practice, we focus on solution quality metrics—specifically, the optimality gap—which demonstrates strong performance compared to industrial solvers in our high-dimensional experiments (Section 5.3).
>
> These explanations have been made below Assumption 2 (**line 313-319**) in the revised manuscript.

---

> ### Author Response · Authors · 2025-11-24
> **Official Response by Authors (Part III)**
>
> -----
> > *Comment 3. The authors claim that the proposed method can find application in many practical scenarios and yet provide experimental results for standard quadratically constrained quadratic problems and electrical power systems. Are there any relevant examples of optimization problems from Machine Learning where the method is applicable?*
> ----
>
> **Response:** Yes, our framework is applicable to several machine learning problems beyond QCQP and power systems.
>
> **Adversarial attacks with non-convex constraints:** As mentioned in the introduction (first paragraph), our method applies to $\ell_p$-constrained adversarial attacks in neural networks [1] with $0<p<1$. In these scenarios, the $\ell_p$-norm ball is star-shaped and thus ball-homeomorphic, making our approach well-suited for optimizing adversarial perturbations under such non-convex constraints. By mapping the $\ell_p$-norm ball into the $\ell_2$-norm ball, we can efficiently conduct PGD to find adversarial samples.
>
> However, for standard $\ell_p$-norm constraints with $p \geq 1$, closed-form homeomorphic mappings already exist (e.g., the gauge mapping [2,3]) that directly map these sets to the $\ell_2$-norm ball. Since these analytical mappings are readily available, INN-based constraint learning is unnecessary for such cases, and we did not include them in our experiments.
>
> Our experimental focus on QCQP and power systems reflects problem domains where (i) constraints are genuinely non-convex without closed-form mappings, and (ii) the computational benefits of our approach are most pronounced. However, the framework's applicability extends to any optimization problem over ball-homeomorphic constraint sets, including the ML applications mentioned above.
>
> [1] Erdemir, E., Bickford, J., Melis, L., Aydore, S. (2021). Adversarial robustness with non-uniform perturbations. Advances in Neural Information Processing Systems, 34, 19147-19159.
>
> [2] Tabas D, Zhang B. Computationally efficient safe reinforcement learning for power systems. In 2022 American Control Conference (ACC) 2022 Jun 8 (pp. 3303-3310). IEEE.
>
> [3] Liu C, Liang E, Chen M. Fast Projection-Free Approach (without Optimization Oracle) for Optimization over Compact Convex Set. In the Thirty-ninth Annual Conference on Neural Information Processing Systems.
>
> -----
> > *Minor questions and typos:*
> ----
>
> **Response:** **We sincerely thank the reviewer for carefully reading our paper and pointing out these typos.** We will thoroughly review the manuscript to correct these issues and improve its overall flow. Below, we provide brief responses to each of the questions.
>
> - *In (P), the unit ball $\mathcal{B}$ is the Euclidean norm ball, $\{x \in \mathbb{R}^n: \|x\|_2 \leq 1\}$?*
>   **Response:** Yes. We have added a footnote in Assumption 1 (**Page 3**) of the revised manuscript to clarify this definition.
>
> - *In general, the phrase "projected gradient descent" is more common than "projection gradient descent" that is found throughout the text.*
>   **Response:** We agree with the reviewer’s suggestion and have updated the relevant terminology, primarily in Section 3.
>
> - *Line 222: "the Lipschitz of the homeomorphism". Should it be the Lipschitz continuity constant of the homeomorphism?*
>   **Response:** Yes, it's a typo and we have modified it. (**line 230**)
>
> - *Line 225: "to train the homeomorphism" should be phrased better.*
>   **Response:** We agree with the suggestion. We modified it as "to train the INN for learning the homeomorphism under different $\theta$". (**line 233**)
>
> - *Definition 3.2: symbol $\mathcal{H}^n$ not defined. It is better to provide a link to the notation. The same holds for the assumptions of the paper which are found in the appendix.*
>   **Response:** Here, $\mathcal{H}^n:=\{\boldsymbol{\phi}:\mathbb{R}^n\to\mathbb{R}^n \mid \boldsymbol{\phi}\ \text{is a homeomorphism}\}$. Moreover, we think it is redundant in Def. 3.2 and hence we delete this notation in Def. 3.2. For details, we define $\mathcal{H}^n$ in Appendix B.4 (**line 1230**).
>
> - *Line 257: What is a non-perfect and non-convex ball? Do the authors mean it in a topological sense? I suggest adding further details on such concepts which are currently missing.*
>   **Response:** Here, the learned ball $\tilde{\mathcal{B}}$ is an "approximate ball", i.e., $(1-\epsilon)\mathcal{B}\subset\tilde{\mathcal{B}}\subset(1+\epsilon)\mathcal{B}$ for a small error $\epsilon$, because the homeomorphism generated by the INN may not produce a perfectly convex ball. In other words, the learned set is close to being a ball but might exhibit minor non-convexities, which is why we describe it as a non-perfect and non-convex ball. We add a footnote to explain this in the revised manuscript (**line 268, Page 5**).
>
> - *Line 323: "this results".*
>   **Response:** Thanks for pointing out this typo. We have modified it (**line 352**).
>
> (to be continued)

---

> ### Author Response · Authors · 2025-11-24
> **Official Response by Authors (Part IV)**
>
> - *Line 366: "The final convergent solution".*
>   **Response:** Thanks for pointing it out. It should be "the final converged solution" (**line 393**).
>
> - *Line 1390: $L_f$-smooth instead of $L_f$ smooth.*
>   **Response:** Thanks for pointing it out. We have modified it (**line 1457**).
>
> - *Line 871: "Nerual" -> "Neural".*
>   **Response:** It's a typo and we modified it (**line 925**), thanks!
>
> - *Line 1558: The KKT conditions are presented and then the Lagrangian of the problem is presented in line 1588 and the KKT conditions for P follow. I would suggest switching the order.*
>   **Response:** We agree with this suggestion. It should be clarified that in Appendix D.3, we want to introduce definitions (including Lagrangian function, KKT conditions) for a general optimization problem and then write the KKT conditions for problem (P) and problem (H). We have modified the structure in Appendix D.3 (**Page 31**) to improve the flow. Thanks again for this suggestion.
>
> - *Lemma D.4: $z^\star=\psi^{-1}(x^\star)$ is missing.*
>   **Response:** Thanks for finding this, we have modified it.
>
> - *Proof of Lemma D.4: the statement of the lemma is that $z^\star$ is a KKT point of $H$, while the proof starts with $z^\star$ being a stationary point of P. Lines 1628-1631 follow by multiplying 1594 with $J_\psi(z^\star)^T$.*
>
>   **Response:** You are correct. At first, it should be we assume $x^*$ is a KKT stationary point of (P). We have modified this (**line 1722**) in the revised manuscript.
>
> - *Line 1649: should be "implies".*
>   **Response:** Thanks for pointing it out. We have modified this (**line 1744**).
>
> - *Line 1659: What does "$\lambda^\star$ as eq. 18" mean?*
>   **Response:** The sentence should be: "In this case, there exists $\boldsymbol{\lambda}^*=0$ such that the KKT condition with eq. (16) of problem (P) holds at $\mathbf{x}^\star=\boldsymbol{\psi}(\mathbf{z}^\star), \boldsymbol{\lambda}^\star=\mathbf{0}$". We have modified this to make this sentence clearer (**line 1755**).
>
> - *Line 1664: "holds" -> "hold".*
>   **Response:** Thanks, we have modified it (**line 1759**).
>
> - *Line 1820: by definition of the normal cone.*
>   **Response:** Thanks for pointing out this typo, and we have modified it (**line 1922**).

---

> > ### Comment · Reviewer_F6JT · 2025-11-27
> >
> > Dear authors,
> >
> > I am satisfied with the response. I will increase my score.

---

### Official Review · Reviewer_SbQJ · 2025-11-01

**Soundness:** 3
**Presentation:** 3
**Contribution:** 3
**Rating:** 6
**Confidence:** 4

**Summary:**

The paper studies optimization over non‑convex constraint sets that are homeomorphic to a ball. It proposes Hom‑PGD+, which learns a homeomorphism from a unit ball to the feasible set with an invertible neural network (INN), solves the transformed problem with projected gradient descent in the ball space, and maps the result back. The algorithm uses a bisected projection along the origin–point ray to handle the approximate ball produced by the learned map.

**Strengths:**

1. The idea is very clear: onece the INN is trained, the online method is standard PGD in a ball with a cheap ray‑bisection projection.
2. The convergence rate mathcing convex‑like first order methods
3. The overall writing is well-organized.

**Weaknesses:**

1. In general, it's very hard to verify whether the set is homeomorphic to ball (requires exponential sample complexity).
2. The approximation assumption used in the paper is very strong: In Assumption 2, it requires a uniform bounded on both the function and its Jocabian.

**Questions:**

1. For JCC/QCQP, could you estimate $\epsilon_{inn}$ from held-out samples and show how good $\epsilon+O(\sqrt{L_H}\epsilon_{inn}) $ bound is?

---

> ### Author Response · Authors · 2025-11-24
> **Official Response by Authors (Part I)**
>
> Dear Reviewer SbQJ,
>
> Thank you for the effort in reviewing our paper and we are happy to receive your positive feedback of our paper. For your comments and questions, we address them one by one in the following.
>
> > *Comment 1. In general, it's very hard to verify whether the set is homeomorphic to ball (requires exponential sample complexity).*
>
> **Response:**  Thank you for raising this important point about the computational complexity of verifying ball-homeomorphism.
>
> We acknowledge that explicit verification via sampling and topological data analysis has high complexity (as discussed in Appendix B.4). However, we emphasize that explicit verification is often unnecessary in practice for the following reasons:
>
> - **Known topological properties:** Many common constraint sets—including convex and star-shaped sets—have well-established topological properties that guarantee ball-homeomorphism without requiring verification.
>
> - **Valid INN condition is sufficient:** More generally, our method can be applied whenever the valid INN condition (Definition 3.2: mapping the center of the unit ball to a feasible point in the constraint set) is satisfied, regardless of whether ball-homeomorphism is explicitly verified. As discussed in Section 4.4, our theoretical guarantees (feasibility and convergence rate) hold under the valid INN condition alone. This makes ball-homeomorphism verification a sufficient but not necessary prerequisite—the valid INN condition provides a more practical and verifiable criterion.
>
> In summary, while verifying ball-homeomorphism is theoretically expensive, practitioners can rely on either known topological properties of standard constraint sets or empirically verify the more tractable valid INN condition to apply our method with theoretical guarantees. We add these discussions in Appendix B.5 **(line 1344-1357)** of the revised manuscript.
>
> -----
> > *Comment 2. The approximation assumption used in the paper is very strong: In Assumption 2, it requires a uniform bounded on both the function and its Jocabian.*
> -----
>
> **Response:**
> We first clarify that, while the approximation errors for the function and its Jacobian are given by different bounds, i.e.,
> $$
> \|\psi-\Phi\|\leq \epsilon_1,\quad \|J_{\psi}-J_{\Phi}\|\leq \epsilon_2,
> $$
> we can define a unified error tolerance by setting $\epsilon_{inn}=\max(\epsilon_1,\epsilon_2)$
> to achieve a uniform bound.
>
> Nevertheless, we acknowledge that Assumption 2 is theoretically strong, and we appreciate the opportunity to clarify (1) why it is necessary for our convergence analysis, and (2) how our algorithmic design helps satisfy it in practice. Assumption 2 consists of two key components:
>
> (i) **Bounded homeomorphism error:**
> $$
> \mathcal{B}(0,1-\epsilon_{\operatorname{inn}}) \subseteq \Phi^{-1}(\mathcal{K}) \subseteq \mathcal{B}(0,1+\epsilon_{\operatorname{inn}}), \quad \|\boldsymbol{\psi}-\Phi\|\leq \epsilon_{\rm inn}.
> $$
> - This condition ensures that $\Phi$ closely approximates the true homeomorphism $\psi$ between the unit ball and the constraint set $\mathcal{K}$. Our INN training framework is specifically designed to achieve this through the loss function in Equation (1), which simultaneously encourages: (a) $\Phi(\mathcal{B}) \subseteq \mathcal{K}$ via the constraint penalty term, and (b) maximal coverage of $\Phi(\mathcal{B})$ within $\mathcal{K}$ via the volume maximization term. For example, we provide validation in Appendix G (**Page 45**) to show how the INN approximates the target constraints of QCQP problems under different input parameters with small errors (**Figs. 6-7**).
>
> (ii) **Bounded Jacobian approximation error:**  $\|J_{\boldsymbol{\psi}}-J_{\Phi}\|\leq \epsilon_{\rm inn}.$
> - This condition is stronger than merely controlling the function value error; it ensures that the derivatives (or local sensitivities) of $\Phi$ and $\psi$ are close. This assumption is indeed needed in our theoretical analysis and is crucial for bounding the approximation error of the KKT solutions in constrained optimization problems. Although this is a strong requirement in theory, it provides an essential upper bound for the worst-case analysis and advances the state-of-the-art results for non-convex constrained optimization.
> - While we cannot directly verify this condition without knowing the ground truth $\psi$, our algorithm design promotes this property through Lipschitz regularization (the spectral norm penalty on $\mathrm{J}_{\Phi}$ in Equation (1)), which prevents $\Phi$ from having large local variations. **Figs. 6-7** in Appendix G (**Page 45**) show that the Lipschitz regularization term decreases consistently during training, indicating that $\Phi$ achieves stable, smooth mappings.
>
> (to be continued...)

---

> ### Author Response · Authors · 2025-11-24
> **Official Response by Authors (Part II)**
>
> (continue with Comment 2)
>
> For high-dimensional problems such as chance-constrained optimization, directly verifying these conditions is indeed non-trivial. However, we emphasize that Assumption 2 serves primarily to enable **worst-case theoretical guarantees**. In practice, we focus on solution quality metrics—specifically, the optimality gap—which demonstrates strong performance compared to industrial solvers in our high-dimensional experiments (Section 5.3).
>
> These explanations have been made below Assumption 2 (**line 313-319**) in the revised manuscript.
>
> -----
> > *Question 1. For JCC/QCQP, could you estimate $\epsilon_{inn}$ from held-out samples and show how good $\epsilon+O(\sqrt{L_H\epsilon_{inn}})$ bound is?*
> -----
>
> **Response:**  Thank you for this insightful question.
>
> - We first clarify that the theoretical bound aims to show that the upper bound of the solution convergence gap is compared to the stationary solution obtained by our methods in the ideal case when the INN is perfectly trained (i.e., $\epsilon_{inn}=0$). In our experiments, we compare against ground-truth solutions obtained by industrial solvers, which provide a more direct and informative performance metric. Empirically, our INN-based PGD consistently achieves very small stationary optimality gaps (less than 3\%) across various benchmarks, including QCQP and other challenging constrained optimization problems.
>
> - We acknowledge that estimating the specific terms in the upper bound provides additional insights into our INN training algorithm. For the illustrative QCQP problems, we provide the following estimates:
>
>     - **$\epsilon_{inn}$ (INN approximation error):** It can be represented as the Hausdorff distance between the true constraint set and the INN-approximated set (zero Hausdorff distance indicates that the INN perfectly maps the ball to the constraint set, achieving the homeomorphism with $\epsilon_{inn}=0$). Using random sampling on boundary points from both sets, as the reviewer suggested, we estimate the average Hausdorff distance across different input parameters to be less than 0.3, as visualized in **Figure 6, Appendix G (**Page 45**)**. This demonstrates that the approximated constraint set closely matches the target set. We have included these discussions from **lines 2385 to 2402** in Appendix G.

---

### Official Review · Reviewer_q4nk · 2025-11-01

**Soundness:** 2
**Presentation:** 2
**Contribution:** 2
**Rating:** 4
**Confidence:** 3

**Summary:**

This paper introduces Hom-PGD+, a method designed for problems with non-convex constraint sets.

The core idea is to use an Invertible Neural Network (INN) to learn a mapping between the complex, non-convex constraint set and a simple unit ball. This transformation allows the algorithm to perform efficient projections onto the unit ball instead of dealing with the original non-convex set.

The proposed method is a first-order, projection-efficient algorithm that achieves an iteration complexity of $\mathcal{O}(\epsilon^{-2}$.

The paper demonstrates the method's effectiveness on problems like non-convex QCQP.

**Strengths:**

1. The idea of using INNs to learn homeomorphisms for reparameterizing constrained optimization problems seems new. It effectively bridges topological concepts with practical machine learning techniques.

2. This paper provides a convergence analysis, establishing an iteration complexity that is competitive with state-of-the-art methods for non-convex constraints. The analysis carefully accounts for the approximation error introduced by the INN.

3. The paper validates its claims on both synthetic (QCQP) and real-world chance-constrained power grid problems.

**Weaknesses:**

1. Assumption 2 appears overly strong, and it is unclear how the QCQP and other constrained optimization problems satisfy it.

2. The performance relies on the accuracy of the learned homeomorphism $\epsilon_{\text{inn}}$; the method may degrade if the INN fails to learn an accurate mapping, particularly for complex constraint sets.

3. Training the INN offline incurs additional computational cost, and verifying that a constraint set is ball-homeomorphic—a prerequisite for the theory—is non-trivial and expensive in high dimensions.

4. The method’s performance may be sensitive to the INN’s depth. A more detailed discussion or guidance on selecting an appropriate INN architecture would enhance its practical applicability.

5. Experiments primarily compare with first-order methods for non-convex constraints. Including a comparison with an industrial solver like IPOPT (for QCQP) or analyzing time-to-solution scaling would provide a clearer view of the method’s competitiveness.

**Questions:**

NA

---

> ### Author Response · Authors · 2025-11-24
> **Official Response by Authors (Part I)**
>
> Dear Reviewer q4nk,
>
> We thank the reviewer for the time and effort dedicated to reviewing our paper and for the positive feedback regarding the novelty of our homeomorphic reparameterization methods. We now address each of your comments and questions in detail below.
>
> > **Comment 1.** *Assumption 2 appears overly strong, and it is unclear how the QCQP and other constrained optimization problems satisfy it.*
>
> **Response:**
> Thank you for this important comment. We acknowledge that Assumption 2 is theoretically strong, and we appreciate the opportunity to clarify (1) why it is necessary for our convergence analysis, (2) how our algorithmic design helps satisfy it in practice, and (3) empirical performance on QCQP and other problems. Assumption 2 consists of two key components:
>
> 1. **Bounded homeomorphism error:**
>    $$\mathcal{B}(0,1-\epsilon_{\operatorname{inn}}) \subseteq \Phi^{-1}(\mathcal{K}) \subseteq \mathcal{B}(0,1+\epsilon_{\operatorname{inn}}), \quad \|\boldsymbol{\psi}-\Phi\|\leq \epsilon_{\rm inn}.$$
>
>    - This condition ensures that $\Phi$ closely approximates the true homeomorphism $\psi$ between the unit ball and the constraint set $\mathcal{K}$. Our INN training framework is specifically designed to achieve this through the loss function in Equation (1), which simultaneously encourages: (a) $\Phi(\mathcal{B}) \subseteq \mathcal{K}$ via the constraint penalty term, and (b) maximal coverage of $\Phi(\mathcal{B})$ within $\mathcal{K}$ via the volume maximization term.
>    - For QCQP problems specifically, we also provide additional validation in Appendix G (**Page 45**) to show how the INN approximates the target constraints under different input parameters with small errors (**Figs. 6-7**).
>
> 2. **Bounded Jacobian approximation error:**  $\|J_\psi-J_\Phi\|\leq \epsilon_{ inn}.$
>
>    - This condition is stronger than merely controlling the function value error; it ensures that the derivatives (or local sensitivities) of $\Phi$ and $\psi$ are close. This assumption is indeed needed in our theoretical analysis and is crucial for bounding the approximation error of the KKT solutions in constrained optimization problems. Although this is a strong requirement in theory, it provides an essential upper bound for the worst-case analysis and advances the state-of-the-art results for non-convex constrained optimization.
>    - While we cannot directly verify this condition without knowing the ground truth $\psi$, our algorithm design promotes this property through Lipschitz regularization (the spectral norm penalty on $\mathrm{J}_{\Phi}$ in Equation (1)), which prevents $\Phi$ from having large local variations. **Figs. 6-7** in Appendix G (**Page 45**) show that the Lipschitz regularization term decreases consistently during training, indicating that $\Phi$ achieves stable, smooth mappings.
>
> 3. **For high-dimensional problems:**
>    For high-dimensional problems such as chance-constrained optimization, directly verifying these conditions is indeed non-trivial. However, we emphasize that Assumption 2 serves primarily to enable **worst-case theoretical guarantees**. In practice, we focus on solution quality metrics—specifically, the optimality gap—which demonstrates strong performance compared to industrial solvers in our high-dimensional experiments (Section 5.3).
>
> These explanations have been made below Assumption 2 (**line 313-319**) in the revised manuscript.
>
> > **Comment 2.** *The performance relies on the accuracy of the learned homeomorphism $\epsilon_{inn}$; the method may degrade if the INN fails to learn an accurate mapping, particularly for complex constraint sets.*
>
> **Response:**
> Thank you for raising this important practical concern. We address it from both theoretical and empirical perspectives:
>
> - **Theoretical Perspective:**
>   In our theory, a large INN approximation error may degrade the performance of our method. However, theoretical analysis typically employs worst-case analysis and provides only an upper bound on the approximation error relative to the true KKT stationary solution.
>
> - **Empirical Perspective:**
>   In practice, the optimality gap—which measures how well solutions satisfy optimality conditions—is the key performance metric. Our experiments demonstrate that the INN-based PGD algorithm consistently achieves very small stationary optimality gaps across various benchmarks, matching or outperforming industrial solvers (e.g., IPOPT and GUROBI) on both synthetic QCQPs and real-world high-dimensional chance-constrained optimization problems.
>
> - **INN Training Procedure:**
>   Our INN training procedure incorporates a well-designed loss function that balances volume maximization, constraint satisfaction, and Lipschitz regularity. This training process promotes that the learned homeomorphism remains accurate even for complex constraint sets, contributing to the robust empirical performance observed in our experiments.

---

> ### Author Response · Authors · 2025-11-24
> **Official Response by Authors (Part II)**
>
> > **Comment 3.** *Training the INN offline incurs additional computational cost, and verifying that a constraint set is ball-homeomorphic—a prerequisite for the theory—is non-trivial and expensive in high dimensions.*
>
> **Response:**  Thank you for raising this important practical consideration. We address both aspects of your concern:
>
> 1. **Offline INN Training Cost:**
>    We acknowledge that training the INN offline adds additional computational cost. However, this cost is incurred only once and can then be amortized across many online problem instances. Furthermore, modern deep learning infrastructure (e.g., PyTorch and GPU) enables efficient INN training (e.g., less than 10 minutes for the high-dimensional chance-constrained problem training). Importantly, once trained, our framework enables solving non-convex constrained optimization with the same $\mathcal{O}(\epsilon^{-2})$ convergence rate as convex optimization and low per-iteration cost. This significantly improves upon the state-of-the-art $\mathcal{O}(\epsilon^{-4})$ (or $\mathcal{O}(\epsilon^{-3})$ under regularity conditions) for general non-convex constrained optimization (see Table 1).
>
> 2. **Ball-Homeomorphism Verification:**
>    We acknowledge that verifying the ball-homeomorphism property incurs high complexity via sampling and conducting topological data analysis (as discussed in Appendix B.4). However, explicit verification is often unnecessary in practice. Many common constraint sets—including convex and star-shaped sets—have known topological properties guaranteeing ball-homeomorphism. More generally, our method can be applied whenever the valid INN condition (**Definition 3.2**: mapping the center of the unit ball to a feasible point in the constraint set) is satisfied, regardless of whether ball-homeomorphism is explicitly verified. As discussed in Section 4.4, our theoretical guarantees (feasibility and convergence rate) hold under the valid INN condition. This makes ball-homeomorphism verification a sufficient but not necessary prerequisite—the valid INN condition provides a more practical and verifiable criterion.
>
> We have added detailed clarification of these points in Appendix B.5 (**Page 25-26**) of the revised manuscript.
>
> > **Comment 4.** *The method’s performance may be sensitive to the INN’s depth. A more detailed discussion or guidance on selecting an appropriate INN architecture would enhance its practical applicability.*
>
> **Response:**
> Thank you for this valuable suggestion. We provide both theoretical insights and empirical guidance on INN architecture selection.
>
> - **Theoretical Tradeoff:**
>   Our method's performance is governed by two key constants from Theorem 1: (i) $\epsilon_{\text{inn}}$, the INN approximation error, and (ii) $L_H$, the smoothness of $H = f \circ \Phi$, which depends on the Lipschitz constant $\mu_{\Phi}$ of the INN. Increasing INN depth generally reduces $\epsilon_{\text{inn}}$ (improving approximation capability) but can theoretically increase $\mu_{\Phi}$ (due to higher network complexity and composition of layers), creating a fundamental tradeoff that affects downstream optimization performance.
>
> - **Mitigation Through Regularization:**
>   To address this tradeoff, our loss function (Equation (1)) incorporates Lipschitz regularization (spectral norm penalty on Jacobians), which constrains $\mu_{\Phi}$ and prevents distortion of the downstream optimization landscape. This regularization is crucial for maintaining favorable optimization properties even with deeper networks.
>
> - **Empirical Guidance:**
>   In Appendix G.2 (**Page 45**), we conduct systematic experiments examining how INN depth affects approximation error, the Lipschitz constant, and downstream optimization performance. Our results show that a 3-layer INN provides a good balance: it achieves low approximation error while maintaining computational efficiency for downstream iterative optimization. Deeper networks provide diminishing returns in approximation quality while increasing computational cost.
>
> Thus, for most applications, we recommend starting with a 3-layer INN architecture and adjusting based on: (1) the complexity of the constraint set (more complex geometries may benefit from additional layers), and (2) the computational budget for training and inference. The Lipschitz regularization should always be included to maintain optimization landscape quality.

---

> ### Author Response · Authors · 2025-11-24
> **Official Response by Authors (Part III)**
>
> > **Comment 5.** *Experiments primarily compare with first-order methods for non-convex constraints. Including a comparison with an industrial solver like IPOPT (for QCQP) or analyzing time-to-solution scaling would provide a clearer view of the method’s competitiveness.*
>
> **Response:**
> Thank you for this valuable suggestion. We clarify that our manuscript does include comparisons with industrial solvers, and we have added further experiments in response to your feedback.
>
> - **Existing Comparisons:**
>   Our original manuscript includes industrial solver benchmarks: (i) IPOPT provides reference solutions for the 2D QCQP illustration, and (ii) Gurobi (a commercial solver with an academic license) benchmarks the large-scale JCC-OPF problem. On the non-convex JCC-OPF, our method achieves solutions within 103 seconds for a 500-bus network—a 10× speedup over Gurobi with only a 3% optimality gap.
>
> - **New QCQP Scaling Experiments (Section 5.2):**
>   In response to your comment, we have conducted comprehensive scaling comparisons with IPOPT on high-dimensional QCQP problems, varying both the number of decision variables $n$ and quadratic constraints $m$ (yielding $\mathcal{O}(n^2 \cdot m)$ problem parameters). Key findings:
>
>   - **Scaling Behavior:** As $m$ increases by two orders of magnitude (from 10 to 1000), IPOPT's per-instance time grows steeply—for $n = 50$, runtime jumps from 3 to 70 seconds. In contrast, Hom-PGD+ exhibits near-constant runtime as $m$ scales and only mild growth with $n$, owing to efficient GPU-accelerated INN computation and batched constraint verification.
>
>   - **Solution Quality:** Hom-PGD+ achieves an average objective gap of 2.9% with zero constraint violations, demonstrating competitive solution quality while maintaining superior computational efficiency.
>
> These results demonstrate that Hom-PGD+ maintains efficiency as problem size grows, while IPOPT's computational cost escalates rapidly, particularly for large $n$ and $m$. We have included these experiments and expanded discussion in the revised manuscript.

---

### Official Review · Reviewer_AhNy · 2025-11-01

**Soundness:** 4
**Presentation:** 4
**Contribution:** 3
**Rating:** 8
**Confidence:** 3

**Summary:**

This paper studies the difficult setting of constrained optimization under non-convex constraints. Noting that several practically relevant constraint classes are homeomorphic to a ball, the authors leverage this to transform the original problem into an equivalent and readily solvable ball-constrained one. Their approach incurs a one-time cost of learning this transformation with an invertible neural network (INN), which is subsequently used for carrying out approximate projections within the classical Projected Gradient Descent scheme. The approach is supported by theory in terms of reformulation equivalence, correspondence of KKT points, and convergence guarantees, as well as extensive numerical experiments.

**Strengths:**

The paper is well-motivated as it proposes a tractable solution for the difficult problem of handling constrained problems with non-convex constraints. The equivalent reformulation into a ball-constrained optimization problem is elegant and highlights a great use case for invertible neural networks. The proposed approach is supported by both theory and extensive experiments, each of which is carried out soundly and with a great amount of detail. The problem and approach description are clear and concise, and supported by explanatory graphics. The related literature is mostly well-covered, to the best of my knowledge.

**Weaknesses:**

1. **Related literature**
	* The idea of using INNs for approximating operators used by classical optimization schemes is not tracked through the literature. Specifically, while the relevant works are cited, the connection is not made explicit. Since this is central to the proposed solution, a discussion is warranted. E.g., the current approach is a bridge between works relying on known unit ball homeomorphisms for convex sets [2] and INN-learned ball-homeomorphisms for arbitrary sets that fulfill sufficient conditions for the existence of such homeomorphisms [1]. In particular, a technical comparison between this work and [1] would be valuable.

2. **Technical assumptions**
	* Lipschitz constants of line 338 are not defined in the main text (deferred to Appendix C2).
	* The technical assumptions on $f_{\theta}$ and $g_{\theta}$ and $\psi_{\theta}$ are also fully deferred to appendix C2. They should be present in the main text, since they are essential to deriving the theoretical results and dictate the applicability of the results.


3. **Minor**
	* Typos: line 222 "the Lipschitz constant", line 1846 "1/L_H"

[1] Liang, Enming, Minghua Chen, and Steven H. Low. "Homeomorphic projection to ensure neural-network solution feasibility for constrained optimization." Journal of Machine Learning Research 25.329 (2024): 1-55.

[2] Chenghao Liu, Enming Liang, and Minghua Chen. Fast projection-free approach (without linear
minimization oracle) for optimization over general compact convex set. Advances in Neural
Information Processing Systems, 2025a.

**Questions:**

* For Fig 5. in the appendix, the last two rows of figures: the log volume quantity exhibits an initial dip followed by an increase. Do the authors know why this happens? Is it due shift in the dominating terms of the loss (as a function of gradient norm)?
* Currently, the technical assumptions on the objective's components are stated as global requirements. Specifically, $f_{\theta}$ is $L_{f,0}$ Lipschitz continuous and $L_f$-smooth .This is quite restrictive, since, for example, the common mean-square type objectives $\frac{1}{2}\\|\cdot\\|^2$ do not satisfy global Lipschitz continuity. Given the compact constraints, can you refine/alleviate these assumptions?

---

> ### Author Response · Authors · 2025-11-24
> **Official Response by Authors (Part I)**
>
> Dear Reviewer AhNy,
>
> Thank you for your time and effort in reviewing our paper, and for your encouraging and insightful review. We are delighted for your positive feedback on our theoretical analysis, extensive experiments, and clear presentation. In the following, we address your questions and concerns one by one.
>
> ----------------------
> > *Comment 1. **Related literature**: The idea of using INNs for approximating operators used by classical optimization schemes is not tracked through the literature. Specifically, while the relevant works are cited, the connection is not made explicit. Since this is central to the proposed solution, a discussion is warranted. E.g., the current approach is a bridge between works relying on known unit ball homeomorphisms for convex sets [2] and INN-learned ball-homeomorphisms for arbitrary sets that fulfill sufficient conditions for the existence of such homeomorphisms [1]. In particular, a technical comparison between this work and [1] would be valuable.*
> ----------------------
>
> **Response:** We thank the reviewer for highlighting the need to explicitly connect our work with existing literature. This work is related to two lines of research on homomorphism and optimization.
>
> - {Homeomorphic reparameterization for optimization}: Methods such as [2] (and related reparameterizations like Hadamard transformations [3,4]) leverage analytically known homeomorphisms specifically for convex sets. We have discussed these works in the Introduction (**lines 77–90**) of the revised manuscript.
>
> - {INN for approximating homeomorphism}: Work [1] demonstrates that INNs can learn homeomorphic mappings that guarantee feasibility under certain conditions. However, [1] focuses on ensuring the feasibility of neural network predictions with respect to constraint sets, rather than solving optimization problems from arbitrary initial points.
>
> Our contribution extends beyond these prior works in several key ways:
>
> - **Algorithmic:** We integrate INN-learned mappings into a projected gradient descent (PGD) framework to solve optimization problems over non-convex, ball-homeomorphic constraint sets. This INN-based reparameterization extends beyond previous works [2,3,4], which are restricted to convex sets. Furthermore, the approach in [1] addresses only the projection subproblem and can be viewed as one step within our overall PGD framework (Alg. 2), whereas we provide an iterative optimization algorithm with convergence guarantees.
>
> - **Theoretical:** We provide a comprehensive landscape (Prop. 4.1) and convergence analysis (Theorem 1) establishing that our framework achieves $O(\epsilon^{-2})$ convergence rates despite the underlying non-convexity, which is a substantial improvement over existing $O(\epsilon^{-3})$ or $O(\epsilon^{-4})$ rates, as explicitly discussed below Theorem 1. Importantly, our analysis also provides explicit bounds that quantify the INN approximation error in the final solution.
>
> We have also added a focused discussion of this positioning in the Introduction (**lines 77–90**) of the revised manuscript to make these connections explicit.
>
> [1] Liang E, Chen M, Low SH. Homeomorphic projection to ensure neural-network solution feasibility for constrained optimization. Journal of Machine Learning Research. 2024;25(329):1-55.
>
> [2] Liu C, Liang E, Chen M. Fast Projection-Free Approach (without Optimization Oracle) for Optimization over Compact Convex Set. InThe Thirty-ninth Annual Conference on Neural Information Processing Systems.
>
> [3] Li Q, McKenzie D, Yin W. From the simplex to the sphere: faster constrained optimization using the Hadamard parametrization. Information and Inference: A Journal of the IMA. 2023 Sep;12(3):1898-937.
>
> [4] Tang T, Toh KC. Optimization over convex polyhedra via Hadamard parametrizations. Mathematical Programming. 2024 Dec 12:1-41.
>
> ----------------------
> > *Comment 2. Technical assumptions.*
> > - Lipschitz constants of line 338 are not defined in the main text (deferred to Appendix C2).
> > - The technical assumptions on $f_\theta$ and $g_{\theta}$ and $\psi_{\theta}$ are also fully deferred to Appendix C2. They should be present in the main text, since they are essential to deriving the theoretical results and dictate the applicability of the results.
> ----------------------
>
> **Response:** We thank the reviewer for raising this important point. We first clarify that the assumptions in our main body indeed include all the needed assumptions to derive our theorem, while we defer the concrete definitions and notations to Appendix C.2. However, as the reviewer pointed out, our assumptions in the main body lack some definitions (e.g., Lipschitz constants of line 338). Therefore, we have modified the structure of the assumption section (**Sec. 4, line 290-320**) in the revised manuscript to improve the flow of our paper. Thanks again for the suggestion.

---

> ### Author Response · Authors · 2025-11-24
> **Official Response by Authors (Part II)**
>
> ----------------------
> > *Minor Comment. Typos: line 222 "the Lipschitz constant", line 1846 "1/L_H"*
> ----------------------
>
> **Response:** We thank the reviewer for pointing out some typos in the manuscript. This typo has been corrected in the revised manuscript (**line 230**). We will carefully review the entire paper and correct every identified error to improve clarity and ensure accuracy throughout the text. We appreciate the detailed feedback and will incorporate these modifications in the final version of the paper.
>
> ----------------------
> > *Question 1. For Fig 5. in the appendix, the last two rows of figures: the log volume quantity exhibits an initial dip followed by an increase. Do the authors know why this happens? Is it due shift in the dominating terms of the loss (as a function of gradient norm)?*
> ----------------------
>
> **Response:** Thank you for this insightful question. Yes, the observed behavior—where log-volume initially dips before increasing—is indeed due to a shift in the dominating terms of the loss function during training, as you suspected. Specifically:
>
> - **Initialization phase:** The INN parameters are randomly initialized (e.g., Gaussian), causing the initial mapping output $\Phi(\mathcal{B})$ to violate the constraint $\Phi(\mathcal{B}) \subseteq \mathcal{K}$. This results in a large constraint penalty term that dominates the total loss (as evident in the second subfigure showing high penalty loss).
>
> - **Shrinking phase:** To reduce constraint violations, the network learns to shrink the mapped region and adjust its position. This shrinking decreases the volume (and thus log-volume drop), while it also reduces the constraint penalty by pushing $\Phi(\mathcal{B})$ fits within $\mathcal{K}$. During this phase, minimizing the penalty term takes priority over maximizing volume.
>
> - **Expansion phase:** Once the constraint is approximately satisfied (indicated by low penalty loss in the second subfigure), the volume maximization term becomes dominant. The network then learns to expand $\Phi(\mathcal{B})$ to occupy as much of $\mathcal{K}$ as possible, ultimately approaching a homeomorphism approximately.
>
> This trade-off between constraint satisfaction and volume maximization is governed by the relative weighting of loss terms, and the transition point corresponds to when the penalty loss drops below a threshold, where volume optimization becomes beneficial. We have added this discussion in Appendix G.1 **(line 2369-2380)**.
>
> ----------------------
> > *Question 2. Currently, the technical assumptions on the objective's components are stated as global requirements. Specifically, $f_\theta$ is $L_{f, 0}$ Lipschitz continuous and $L_f$ smooth. This is quite restrictive, since, for example, the common mean-square type objectives $\frac{1}{2}\|\cdot\|^2$ do not satisfy global Lipschitz continuity. Given the compact constraints, can you refine/alleviate these assumptions?*
> ----------------------
>
> **Response:** Thanks for the insightful question. Yes, we can relax the assumption given the compact constraints. Indeed, we only require $f$ to be Lipschitz continuous on a compact set containing the feasible constrained set $\mathcal{K}$. The following are detailed explanations.
>
> In our convergence analysis of the Hom-PGD$^+$ algorithm, we only require that the composite function $H = f \circ \Phi $
> satisfies:
> (1) $L_H$-smoothness, and  (2) $L_{H, 0}$-Lipschitz continuity on the iterates
> (with both constants depending on the Lipschitz constant of $f$; see Lemma D.1).
>
> Since each iterate $z_k$ is feasible in the ball $\mathcal{B}$, the update
> $
> z_{k+1}^{+} = z_k - \alpha_k \nabla H(z_k)
> $
> remains in a compact set $\mathcal{M}$ (which contains $\mathcal{B}$) for bounded $\alpha_k$ and $\|\nabla H(z)\|$. Thus, it suffices for $H$ to be smooth and Lipschitz continuous over $\mathcal{M}$, meaning that $f$ need only be Lipschitz continuous on the compact set $\Phi(\mathcal{M}) \supset \mathcal{K}$.
>
> These discussions have been added in Appendix C.2 (**line 1478-1487**) and are clarified in the main body (**line 298**) in the revised manuscript.

---

> > ### Comment · Reviewer_AhNy · 2025-11-25
> > **Thank you for your response**
> >
> > I acknowledge the authors' response, as well as the other reviews and respective responses. My questions have been well addressed, and I keep my score.

---

### Author Response · Authors · 2025-12-04
**Author Final Remarks**

Dear Reviewers, Area Chairs, and Program Chairs,

We sincerely thank all the reviewers for their insightful and constructive comments, and thank the area chairs and program chairs for their extra efforts spent on our paper. We are saddened by the unexpected accident that affected our community, and we hope that everyone's efforts will pay off.

We first thank the reviewers for their acknowledgment of the novelty and significance, theoretical and practical contribution, and clear and well-organized structure of our paper:

- **Novelty and Significance**:
   - *Reviewer AhNy*:
> "The paper is well-motivated as it proposes a tractable solution for the difficult problem of handling constrained problems with non-convex constraints." "The related literature is mostly well-covered, to the best of my knowledge."
   - *Reviewer q4nk*:
> ''The idea of using INNs to learn homeomorphisms for reparameterizing constrained optimization problems seems new''

- **Theoretical and Practical Contributions**:
   - *Reviewer AhNy*:
 > "The proposed approach is supported by both theory and extensive experiments, each of which is carried out soundly and with a great amount of detail. "
   - *Reviewer q4nk*:
> "The analysis carefully accounts for the approximation error introduced by the INN." "The paper validates its claims on both synthetic (QCQP) and real-world chance-constrained power grid problems."
   - *Reviewer SbQJ*:
> "The convergence rate mathcing convex‑like first order methods"
   - *Reviewer F6JT*:
> "The experimental results show that the proposed algorithm performs similarly or better than state-of-the-art methods."
   - *Reviewer rwv2*:
> "This reformulation reduces the problem to a ball-constrained one, enabling the use of projection gradient descent to solve the transformed optimization."


- **Clear and Well-Organized Structure.**
    - *Reviewer SbQJ*:
> "The overall writing is well-organized."
    - Reviewer F6JT:
> "The main idea of the paper is interesting and easy to follow."



We have uploaded the revised manuscript, with all major changes clearly marked in **blue**. Based on the reviewers’ feedback, we have made the following main and significant improvements to address the reviewers' main concerns:

- **Theoretical Discussion and Clarification:**

  - We have revised the discussion regarding the global Lipschitz continuity assumption, showing that it can be relaxed under specific conditions in Appendix C.2 (Reviewer AhNy).

  - We have clarified and reinforced the discussion on the reasonableness and necessity of Assumption 2 in Sec. 4 (Reviewers q4nk, SbQJ, F6JT, and rwv2).

- **Discussion on Offline Complexity (Appendix B.5):** We have expanded our discussion on the offline computational cost (Reviewers q4nk and SbQJ).

   - INN training. While the training process needs high complexity, this process only incurs once and once the INN is trained well, our framework can enjoy a fast convergence rate with low cost in each iteration. Moreover, modern deep learning frameworks, such as PyTorch coupled with GPU acceleration, render the training process efficient (e.g., less than 10 minutes for high-dimensional chance-constrained problems).

   - The verification process for determining whether a set is ball-homeomorphic. While verifying the ball-homeomorphism property through sampling and topological data analysis can be computationally expensive, explicit verification is often unnecessary in practice.
   Many common constraint sets—including convex and star-shaped sets—possess known topological properties that naturally guarantee ball-homeomorphism.

- **Additional Experiments:**

  - We added experiments comparing Hom-PGD$^+$ with IPOPT on high-dimensional non-convex QCQP problems (Section 5.2) to demonstrate the scalability of our approach. (Reviewer q4nk)

  - We also evaluated INN training across different network depths (see Appendix G) to identify the configuration that yields the best performance. (Reviewer q4nk)

- **Presentation and Typos:**
  - We have corrected various typographical errors and refined the overall flow of the manuscript to improve clarity.

For all the concerns and questions from reviewers, we respond to them one by one in detail, which can be found below the reviewer's official comments.  We appreciate the constructive feedback offered during the rebuttal process, despite a known technical issue that limited detailed post-rebuttal responses. Notably,

- Reviewer F6JT expressed satisfaction with our revisions and increased the score from 4 to 6.

- Reviewer AhNy confirmed that all concerns had been addressed and maintained a score of 8.

We also recognize that some reviewers did not have the opportunity to provide additional commentary due to time constraints. Overall, these positive updates underscore the substantial improvements made to our manuscript.

---

### Meta-Review · Area_Chair_Q6sc · 2026-01-06

**Summary:**

The approach is well motivated and empirical results promising. Disagreement is whether the contribution is a substantive framework advance or an incremental integration of prior homeomorphic feasibility and projection ideas. The split is that the strongest claims hinge on Assumption 2, which multiple reviewers view as overly strong and difficult to verify in practice, especially the Jacobian-level approximation requirement. The rebuttal improved clarity and added experiments, but teh remaining gap in theory+practice, limited breadth of evaluation, remain concerning.

**Reviewer Concerns:**

Novelty. Several reviewers questioned how much is new beyond prior homeomorphic projection and related reparameterization work, requesting sharper technical differentiation.

Assumptions. Assumption 2 was repeatedly flagged as strong and hard to validate, particularly uniform Jacobian approximation.
Requests to connect theoretical error terms to measurable quantities (or provide practical verification guidance).

Experimental. Desire for stronger scaling + time-to-solution evidence + comparisons to industrial solvers + broader ML-relevant tasks.

**Reviewer Scores:**

No change except one noted by F6JT

---

### Decision · Program_Chairs · 2026-01-26

Reject